# AAV1 is the optimal viral vector for optogenetic experiments in pigeons (*Columba livia*)

Noemi Rook [1✉], John Michael Tuff[1], Sevim Isparta [1,2], Olivia Andrea Masseck [3], Stefan Herlitze[4], Onur Güntürkün[1] & Roland Pusch[1]

Although optogenetics has revolutionized rodent neuroscience, it is still rarely used in other model organisms as the efficiencies of viral gene transfer differ between species and comprehensive viral transduction studies are rare. However, for comparative research, birds offer valuable model organisms as they have excellent visual and cognitive capabilities. Therefore, the following study establishes optogenetics in pigeons on histological, physiological, and behavioral levels. We show that AAV1 is the most efficient viral vector in various brain regions and leads to extensive anterograde and retrograde ChR2 expression when combined with the CAG promoter. Furthermore, transient optical stimulation of ChR2 expressing cells in the entopallium decreases pigeons' contrast sensitivity during a grayscale discrimination task. This finding demonstrates causal evidence for the involvement of the entopallium in contrast perception as well as a proof of principle for optogenetics in pigeons and provides the groundwork for various other methods that rely on viral gene transfer in birds.

[1] Department of Biopsychology, Institute of Cognitive Neuroscience, Faculty of Psychology, Ruhr University Bochum, Universitätsstraße 150, 44801 Bochum, Germany. [2] Department of Genetics, Faculty of Veterinary Medicine, Ankara University, Şht. Ömer Halisdemir Blv, 06110 Ankara, Turkey. [3] University of Bremen, Synthetic Biology, Leobener Straße 5, 28359 Bremen, Germany. [4] Department of General Zoology and Neurobiology, Ruhr University Bochum, Universitätsstraße 150, 44801 Bochum, Germany. ✉email: noemi.rook@rub.de

Birds are valuable model organisms for comparative neuroscientific research as different avian species provide unique research opportunities. While crows have excellent cognitive abilities that are on par with primates[1–3], zebra finches and other songbirds are widely studied as a model for language[4–7]. Furthermore, pigeons have outstanding visual capabilities[8], navigational skills[9,10], and represent classic animal models for research on learning and memory[11]. Birds are capable of those behaviors although their brains are organized radically different than those of mammals[12]. While the mammalian neocortex is organized in six layers, the pallium of birds is structured in a nuclear fashion[12]. There is, however, cumulative evidence suggesting that, although the avian and mammalian brains differ on the macroscopic level, the local circuitry within their sensory systems is highly comparable indicating conserved principles in sensory systems organization[13]. Finding those invariant properties can help to establish circuit–function relationships that highlight general principles of the brain. Thus, comparative research is indispensable to understand how brain functions emerge from structure[14].

Unfortunately, most of what we know about the function of the avian pallium, especially for the sensory system, comes from purely correlative methods[15,16] or lesion studies that lack spatial and temporal precision[17–21]. However, in order to study the function of neuronal networks, methods that are able to control neuronal activity precisely are mandatory. This ambitious goal was first achieved with optogenetics, allowing researchers to activate or silence specific networks with high temporal and spatial resolution through the integration of artificial light-sensitive ion channels into the cell membrane[22,23]. Optogenetics brought a revolution to rodent research[24] and has been established in other species such as primates[25], zebra finches[7], and ferrets[26] over the last years. However, the functional implementation of optogenetics in other species has been challenging[27]. While several studies have been able to show the effects of electrical microstimulation during decision-making or perception in primates[28–30], studies using optogenetic stimulation have sometimes failed to find behavioral effects, despite reporting physiological changes[25,31–33]. One explanation that has been provided for the absence of behavioral effects is insufficient viral efficiency resulting in low amounts of protein expression[25,27]. As one key component in optogenetics is the expression of light-sensitive ion channels that are typically transferred into the brain via viral vectors, the efficiency of those constructs has to be carefully investigated prior to the application of optogenetics in vivo. Viral vector efficiency can vary considerably between brain areas and species highlighting the need for viral transfection studies in various model organisms[34–39]. Especially in birds, viral transfection has proven to be difficult[40], possibly due to properties of the immune system[41,42]. Although optogenetics has been already used in some areas of the zebra finch song system[7,43,44], the efficiency of different viral constructs was not compared within these and other brain areas such as the visual system. In this study, we compared the efficiency of six viral constructs in their ability to transduce neurons in the pigeon forebrain and found that AAV1 is the optimal viral construct for optogenetic experiments in birds. As it has been complicated to induce behavioral effects with optogenetics in primates due to insufficient protein expression[25,27], we furthermore confirmed that stimulation of channelrhodopsin (ChR2) leads to physiological as well as behavioral effects in pigeons. In our study, we have focused on the visual system, as birds are highly visual animals and recent studies have indicated that characteristic properties of sensory systems, such as a columnar and laminar organization, are conserved between birds and mammals[13]. We targeted the entopallium, which is the most important primary visual area in the pigeon telencephalon and which has been associated with discrimination

of form, pattern, color, motion, and luminance[17,18,21]. We employed a grayscale visual discrimination task and found that optogenetic stimulation within this structure resulted in impaired contrast perception indicated by decreased discrimination accuracy. With this study, we provide a proof of principle for optogenetics in pigeons as well as further insights into the function of the entopallium.

## Results

**Comparative transduction analysis of adeno-associated viral vector serotypes 1, 5, and 9 in the avian forebrain.** In a first step, we wanted to determine the most efficient adeno-associated viral vector (AAV) for optogenetic experiments in the visual system of pigeons. For the viral transfection study, we compared the efficiency of AAV serotype 2 pseudotyped with serotype 1 (here referred to as AAV1), pseudotyped with serotype 5 (here referred to as AAV5), and pseudotyped with AAV9 (here referred to as AAV9). All serotypes were combined with either the human synapsin 1 gene (hSyn) promoter or the chicken beta-actin (CAG) promoter and were injected into the entopallium of the pigeon brain (Fig. 1a, b). Each construct was injected into at least five separate hemispheres of three pigeons (for more information see Table 1). After 6 weeks of transfection, pigeons were sacrificed and immunohistochemical stainings against ChR2 were performed in all brain slices containing the entopallium. The counterstaining was performed to allow for an equal comparison between the serotypes, as serotypes with the hSyn promoter were tagged with eYFP, whereas serotypes with the CAG promoter were tagged with mCherry. Moreover, the amount of transgene expression can be underestimated when analyzing native fluorescence, as the signal increases with counterstainings (see Supplementary Fig. 1 and Method section for more detail). The efficiency of all six constructs was assessed based on the number of ChR2 expressing somata (Fig. 1a) and the transfected area of ChR2 expressing somata, dendrites, and axons in relation to the size of the entopallium (Fig. 1b).

We found that the construct had a significant effect on the number of ChR2 expressing cells (one-way ANOVA with Welch correction $F_{(4,9.511)} = 14.949$, $p < 0.001$, Fig. 1c). While there was no difference in the number of ChR2 expressing cells between injections of AAV1-hSyn-ChR2 and AAV1-CAG-ChR2 (Bonferroni corrected pairwise comparisons, $p = 1.00$, Fig. 1c, e, h), both constructs were significantly more efficient than all other serotypes including AAV5-hSyn-ChR2 ($p \leq 0.001$, Fig. 1c, f), AAV5-CAG-ChR2 (AAV1-hSyn-ChR2: $p = 0.003$, AAV1-CAG-ChR2: $p = 0.025$, Fig. 1c, i) and AAV9-CAG-ChR2 ($p \leq 0.001$, Fig. 1c, j, for mean values and SEM see Table 1). Furthermore, the construct had a significant effect on the percentage of ChR2 expressing area within the entopallium (one-way ANOVA with Welch correction, $F_{(4,8.515)} = 12.791$, $p = 0.001$, Fig. 1d). There was no significant difference in the percentage of the transduced area between injections of AAV1-hSyn-ChR2 and AAV1-CAG-ChR2 (Bonferroni corrected pairwise comparisons, $p = 0.402$, Fig. 1d, e, h). However, AAV1-CAG-ChR2 resulted in a significantly greater area expressing ChR2 than all other serotypes including AAV5-hSyn-ChR2 ($p < 0.001$, Fig. 1d, f), AAV5-CAG-ChR2 ($p = 0.001$, Fig. 1d, i) and AAV9-CAG-ChR2 ($p = 0.001$, Fig. 1d, j, for mean values and SEM see Table 1). Moreover, the ChR2 expressing area was significantly greater for AAV1-hSyn compared to AAV5-hSyn ($p = 0.039$, Fig. 1d–f). Furthermore, transduction efficiencies of the serotypes followed a similar pattern when the ChR2 expressing area was compared to the size of the entopallium only in slices with transduction (see Supplementary Fig. 2). The expression pattern following injections of AAV9 differed from all the other serotypes, as for AAV9-hSyn-ChR2 only two ChR2 expressing

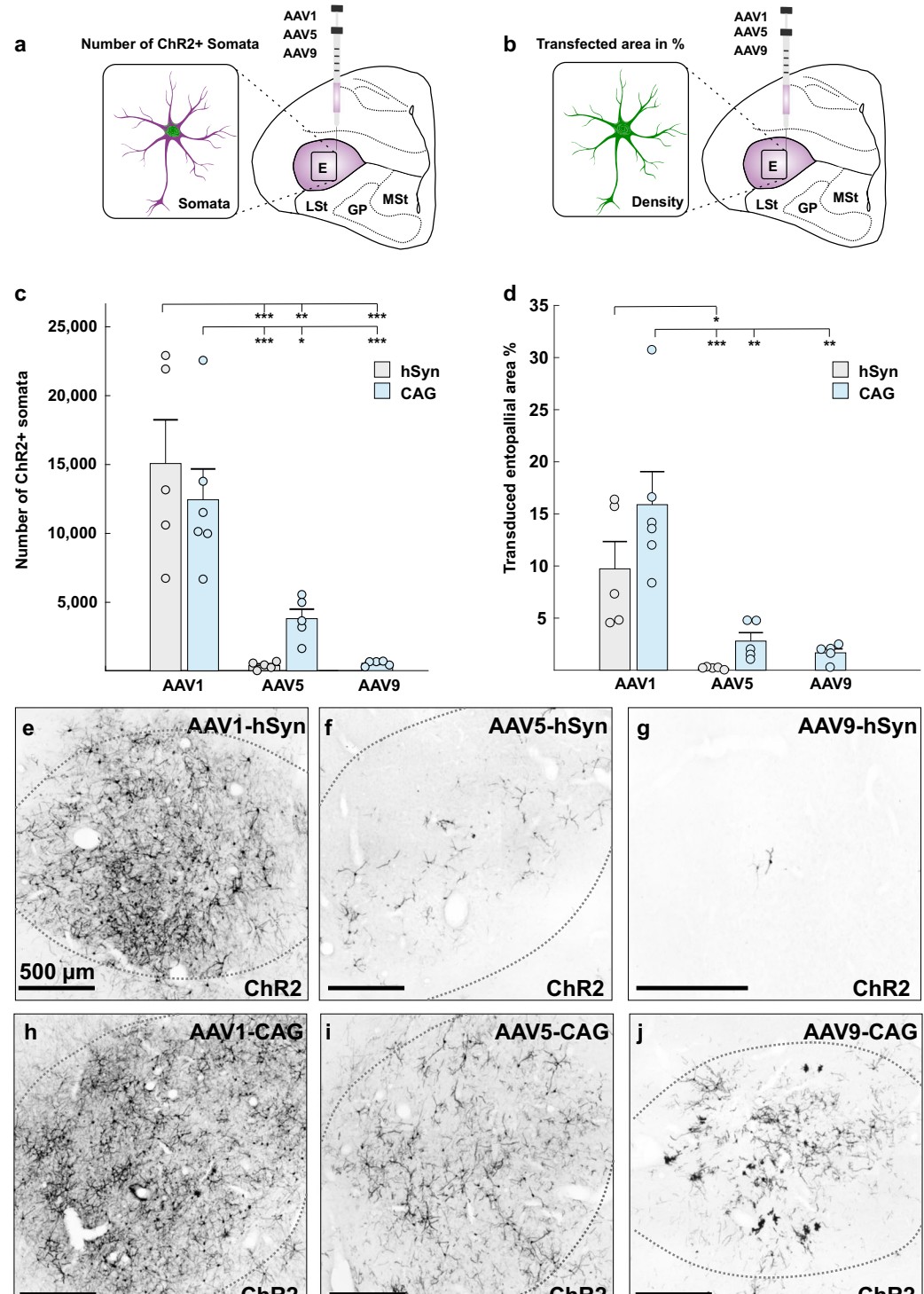

**Fig. 1 Comparative transduction analysis of AAV1, AAV5, and AAV9 in combination with the hSyn and CAG promoters. a** Schematic illustration of the injection area and analysis type. For the first analysis, all somata were counted that displayed ChR2 expression. **b** Schematic illustration of the injection area and the analysis type. For the second analysis, the area of ChR2 expressing somata, dendrites, and axons was measured and compared to the total area of the entopallium. **c** Quantitative comparison of all tested constructs in their ability to drive transgene expression in somata of the entopallium. AAV1-hSyn-ChR2, as well as AAV1-CAG-ChR2, were significantly more efficient than all other tested constructs. **d** Percentage of ChR2 expressing entopallial area for all tested constructs. AAV1-CAG was significantly more efficient than all other tested constructs. All AAVs with hSyn promoter are depicted in gray and all AAVs with CAG promoter are depicted in blue. **e-j** Qualitative pictures of ChR2 expression following injections of **e** AAV1-hSyn-ChR2 ($n = 5$), **f** AAV5-hSyn-ChR2 ($n = 5$), **g** AAV9-hSyn-ChR2 ($n = 5$), **h** AAV1-CAG-ChR2 ($n = 6$), **i** AAV5-CAG-ChR2 ($n = 5$), and **j** AAV9-CAG-ChR2 ($n = 5$). All scale bars represent 500 μm. Error bars represent the standard error of the mean (SEM) and dots represent the raw data, ***$p < 0.001$, **$p < 0.01$, *$p < 0.05$. Abbreviations: AAV adeno-associated viral vector, hSyn human synapsin 1 gene promoter, CAG chicken beta-actin promoter.

**Table 1 Comparative transduction analysis of AAV1, AAV5, and AAV9 in combination with the hSyn and CAG promoter.**

| Serotype | Number of ChR2 expressing cells | Transduced entopallial area in % |
|---|---|---|
| AAV1-CAG | 12405 ± 2230 SEM, $n = 6$ | 15.89 ± 3.17 SEM, $n = 6$ |
| AAV1-hSyn | 15028 ± 3170 SEM, $n = 5$ | 9.73 ± 2.62 SEM, $n = 5$ |
| AAV5-CAG | 3782 ± 690 SEM, $n = 5$ | 2.79 ± 0.81 SEM, $n = 5$ |
| AAV5-hSyn | 406 ± 125 SEM, $n = 5$ | 0.18 ± 0.05 SEM, $n = 5$ |
| AAV9-CAG | 574 ± 83 SEM, $n = 5$ | 1.67 ± 0.38 SEM, $n = 5$ |

For all serotypes, the mean number of ChR2 expressing cells and the mean ChR2 expressing entopallial area in % was assessed for at least five injections into separate hemispheres of at least three pigeons.

cells were found in one of five cases (Fig. 1g), while AAV9-CAG-ChR2 led to reliable ChR2 expression in all five cases but also to neurotoxicity (Fig. 1j, will be discussed in detail later).

While AAV1-CAG-ChR2 and AAV1-hSyn-ChR2 did not differ in their efficiency to drive ChR2 expression within the entopallium, they differed in other properties such as anterograde and retrograde expression of ChR2 in target and input structures of the entopallium. We found that AAV1-CAG-ChR2 injections into the entopallium (Fig. 2a) resulted in extensive ChR2 expression in fibers projecting to target structures such as the ventrolateral mesopallium (MVL, Fig. 2b, d) and nidopallium intermedium (NI, Fig. 2b, d). In contrast to this, only little ChR2

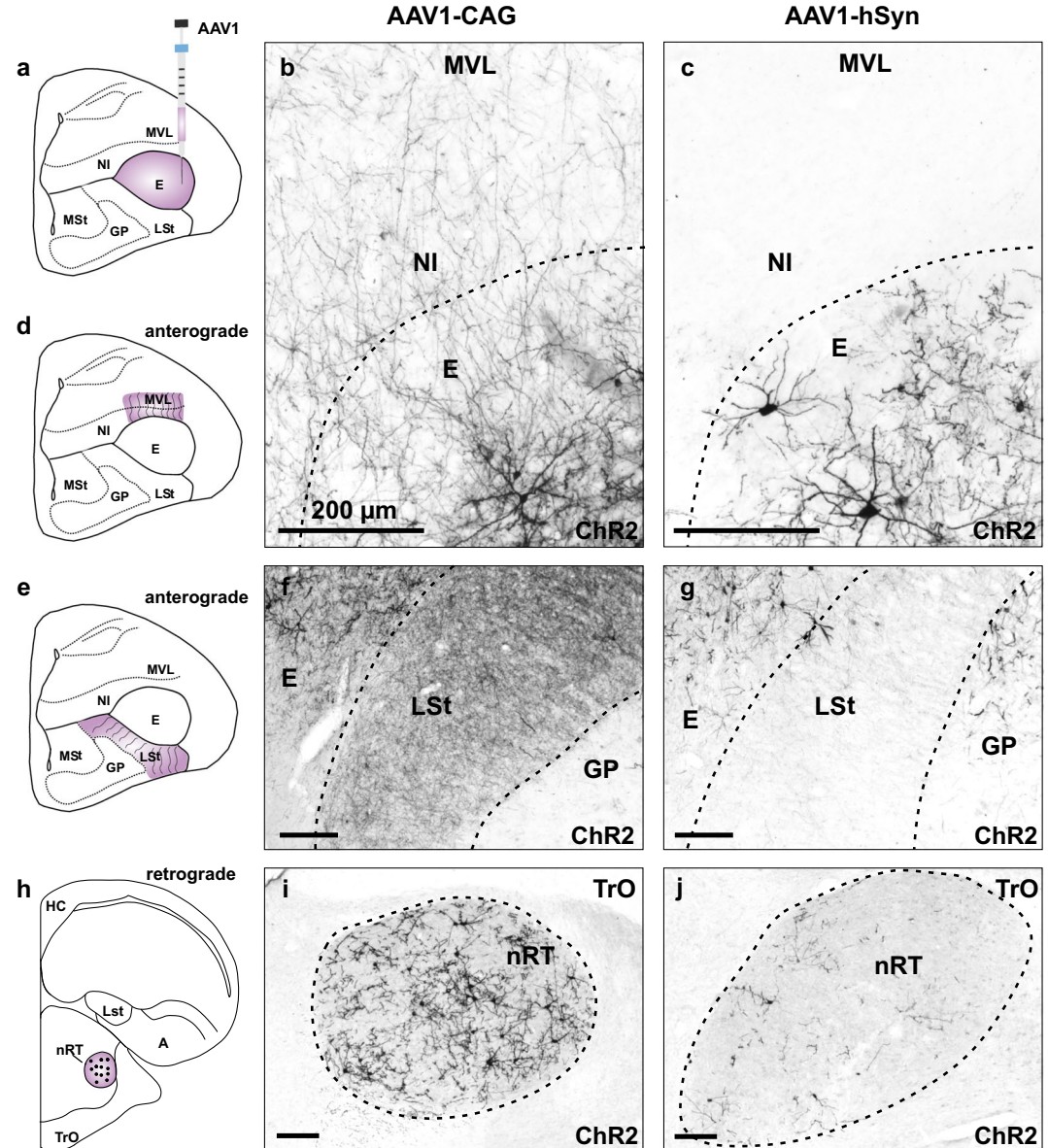

**Fig. 2 Anterograde and retrograde ChR2 expression in target and input structures of the entopallium. a** Schematic illustration of the injection area in the entopallium. **b** Anterograde ChR2 expression in fibers projecting to the NI/MVL following injections of AAV1-CAG-ChR2. **c** Little anterograde ChR2 expression following injections of AAV1-hSyn-ChR2. **d** Schematic illustration of anterograde labeling in fibers projecting to the NI/MVL. **e** Schematic illustration of anterograde labeling in fibers projecting to the striatum. **f** Anterograde ChR2 expression in fibers projecting to the striatum following injections of AAV1-CAG-ChR2. **g** Little anterograde ChR2 expression in the striatum following injections of AAV1-hSyn-ChR2. **h** Schematic illustration of ChR2 expression in the nucleus rotundus, which is the main input region of the entopallium. **i** Extensive retrograde ChR2 expression in the nucleus rotundus following injections of AAV1-CAG-ChR2 into the entopallium. **j** Scarce retrograde ChR2 expression in the nucleus rotundus following injections of AAV1-hSyn-ChR2. All scale bars represent 200 µm. Note that images **f**, **g**, **i**, **j** have been rotated 90° to the left. Abbreviations: Arco arcopallium, E entopallium, GP globus pallidus, LSt lateral striatum, MSt medial striatum, MVL ventrolateral mesopallium, NI nidopallium intermedium, nRT nucleus rotundus, TrO tractus opticus.

expression could be detected in NI and MVL following injections of AAV1-hSyn-ChR2 (Fig. 2c, Supplementary Table 1, Supplementary Figs. 3 and 4). A similar pattern of expression could be seen in the striatum (Fig. 2e), which is another target structure of the entopallium. We found extensive ChR2 expression in fibers projecting to the striatum following injections of AAV1-CAG-ChR2 into the entopallium (Fig. 2f), which was weaker for AAV1-hSyn-ChR2 injections (Fig. 2g, Supplementary Table 1, Supplementary Figs. 5 and 6). The main input region to the entopallium is the diencephalic nucleus rotundus (Fig. 2h). Injections of both AAV1-hSyn-ChR2 and AAV1-CAG-ChR2 into the entopallium resulted in ChR2 expression in neurons in the nucleus rotundus after 6 weeks of expression time (Supplementary Table 1, Supplementary Figs. 7 and 8). However, retrograde ChR2 expression was more extensive after AAV1-CAG-ChR2 injections (Fig. 2i) than after AAV1-hSyn-ChR2 injections (Fig. 2j), especially after longer expression times of 6 months (Fig. 2i, j, Supplementary Table 1, Supplementary Figs. 7 and 8).

The efficiency of AAV1-CAG-ChR2 was furthermore investigated with single injections of 5 µl each in other regions of the avian pallium to assess its brain-wide usefulness for optogenetic experiments in birds. AAV1-CAG-ChR2 was able to drive ChR2 expression in the hippocampus (Fig. 3a, b), the nidopallium caudolaterale (Fig. 3a, c), the entopallium (Fig. 3d, e), the globus

pallidus (Fig. 3d, f), the hyperpallium apicale (Fig. 3g, h), and the medial striatum (Fig. 3g, i).

Since the CAG promoter can drive transgene expression in all cell types, the extent to which ChR2 expressing cells were also co-localized with a neuronal marker was investigated. Therefore, we performed combined immunohistochemical stainings against ChR2 and NeuN to visualize neurons (Fig. 4a–d). We found that AAV1-CAG-ChR2 led to significantly more transgene expression in neurons than in glial cells (neurons: 92.19% ± 1.99 SEM, $n = 5$; glial cells: 7.81% ± 1.99 SEM, $n = 5$; $Z = -2.023$, $p = 0.043$, Fig. 4a–d). This quantification could not be performed for AAV9-CAG-ChR2 injections, as this serotype resulted in a severe reduction of NeuN expression within the ChR2 expressing area (AAV1-CAG: 1871 NeuN+ cells per mm$^2$ ± 41 SEM, $n = 6$; AAV9-CAG: 1102 NeuN+ cells per mm$^2$ ± 77 SEM, $n = 4$; $Z = -2.558$, $p = 0.01$, Fig. 4e–h), suggesting neurotoxicity of this serotype. To further investigate the possible neurotoxicity of this serotype, we performed combined stainings against ChR2 and glial fibrillary acidic protein (GFAP) for AAV9-CAG-ChR2 and AAV1-CAG-ChR2, as GFAP is a marker for astrocyte activation after stress or injury to the brain[45,46]. We found that injections of AAV1-CAG-ChR2 led to extensive ChR2 expression in the injection area, while GFAP expression was low and occurred mainly around blood vessels (ChR2 expression: 8.5% ± 1.6 SEM,

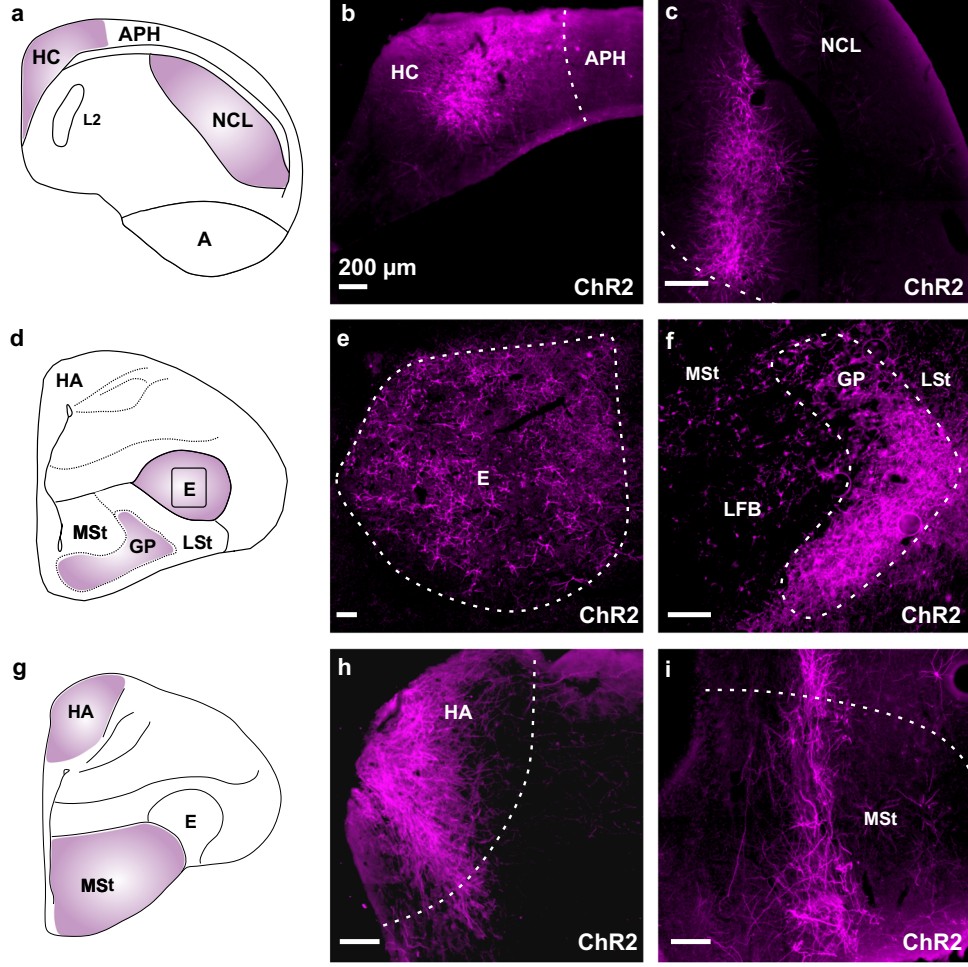

**Fig. 3 AAV1-CAG is efficient in driving transgene expression in various regions of the pigeon forebrain. a** Schematic illustration of the hippocampus (HC) and nidopallium caudolaterale (NCL). **b** ChR2 expression in HC. **c** ChR2 expression in NCL. **d** Schematic illustration of the entopallium (E) and globus pallidus (GP). **e** ChR2 expression in E. **f** ChR2 expression in GP. **g** Schematic illustration of the hyperpallium apicale (HA) and the medial striatum (MSt). **h** ChR2 expression in HA. **i** ChR2 expression in MSt. All scale bars represent 200 µm. Abbreviations: APH area parahippocampalis, E entopallium, GP globus pallidus, HA hyperpallium apicale, HC hippocampus, LFB lateral forebrain bundle, LSt lateral striatum, MSt medial striatum.

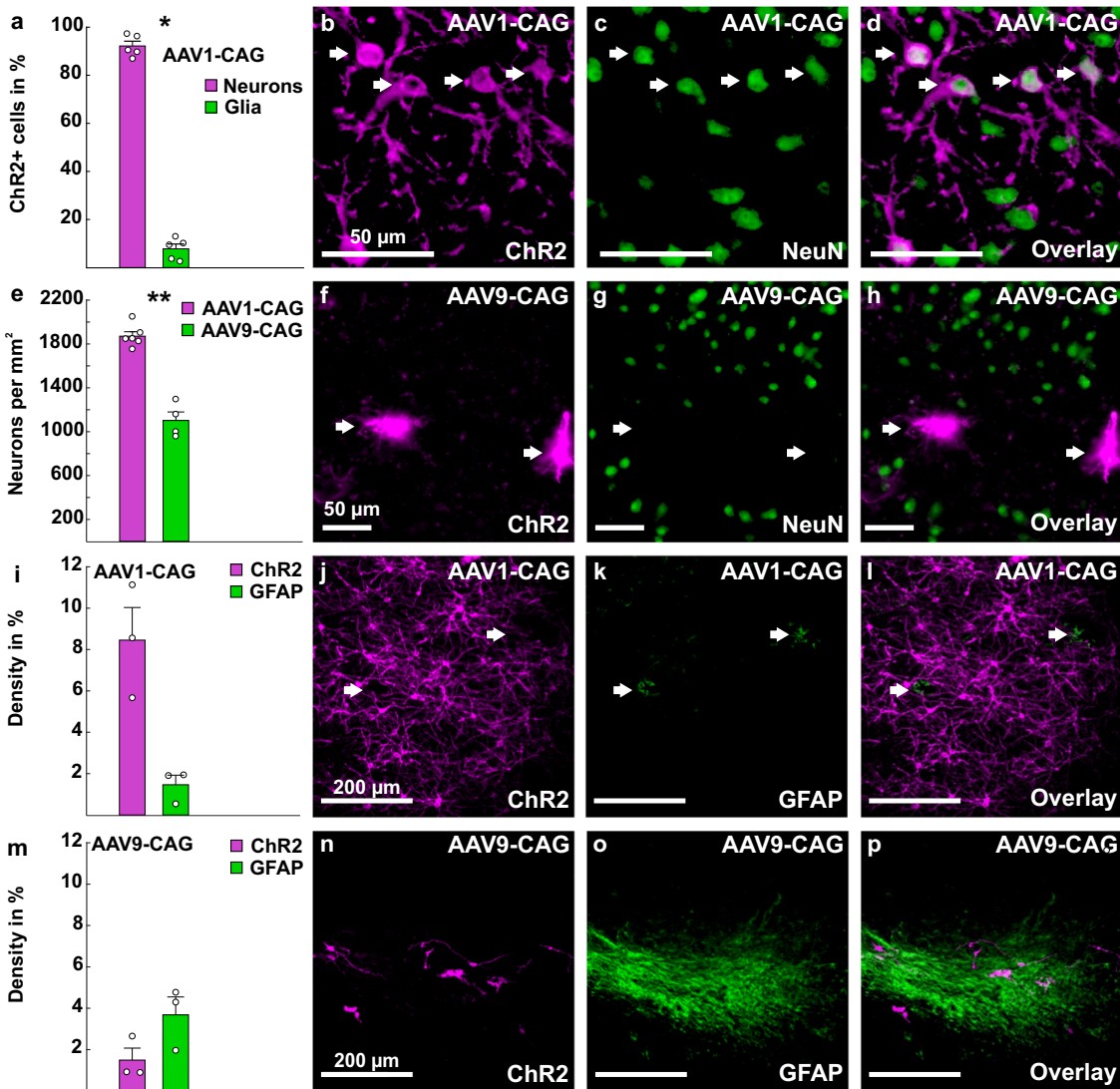

**Fig. 4 Cellular tropism of AAV1-CAG and AAV9-CAG. a–d** AAV1-CAG-ChR2 leads to ChR2 expression in significantly more neurons (pink bar) than glial cells (green bar, $n = 5$ injections). **b** ChR2 expression after injections of AAV1-CAG. **c** NeuN expression in the corresponding injection site. **d** Overlay of ChR2 and NeuN expression. **e–h** AAV9-CAG (green bar, $n = 4$ injections) leads to a significant reduction of NeuN in the injection site compared to AAV1-CAG (pink bar, $n = 5$ injections). **f** ChR2 expression following injections of AAV9-CAG. **g** NeuN in the corresponding injection site. **h** Overlay of ChR2 and NeuN expression indicating that AAV9-CAG injections result in reduced NeuN expression in the injection site. **i–l** AAV1-CAG leads to extensive ChR2 expression (pink bar), but only to weak GFAP expression (green bar, $n = 3$ injections). **j** ChR2 expression following injections of AAV1-CAG-ChR2. **k** GFAP expression occurs mainly around blood vessels. **l** Overlay of ChR2 and GFAP expression. **m** AAV9-CAG injections lead to weak ChR2 expression (pink bar), but increased GFAP expression (green bar, $n = 3$ injections) in the injection site. **n** ChR2 expression following injections of AAV9-CAG-ChR2. **o** GFAP expression within the corresponding injection site. **p** Overlay of ChR2 and GFAP expression. All scale bars are specified within the microscopic images. Error bars represent the standard error of the mean (SEM), dots represent the raw data. $**p < 0.01$, $*p < 0.05$.

$n = 3$; GFAP expression: $1.5\% \pm 0.5$ SEM, $n = 3$; Fig. 4i–l, Supplementary Fig. 9). In contrast to this, injections of AAV9-CAG led to weak ChR2 expression, while GFAP expression was strong and occurred throughout the injection site (ChR2 expression: $1.5\% \pm 0.6$ SEM, $n = 3$; GFAP expression: $3.7\% \pm 0.9$ SEM, $n = 3$; Fig. 4m–p, Supplementary Fig. 9). This supports the idea of neurotoxicity for AAV9-CAG-ChR2.

**Physiology of ChR2 expressing cells during optical stimulation investigated with in vivo electrophysiology and immediate early gene expression.** The physiology of ChR2 was assessed in two experiments. In the first experiment, pigeons were anesthetized, and extracellular single-unit recordings and optical stimulation were performed simultaneously within the entopallium of

two pigeons in four hemispheres. The goal of these specific experiments was to preselect cells that were responsive to light and assess their characteristics. Therefore, a constantly repeated light pulse of 1 s duration was presented to evoke spikes during the advancement of the electrode. When a responsive cell was encountered, we used a sequence of blue light pulses (465 nm) of different durations (1, 10, 100, 200, and 500 ms; see Supplementary Table 2) for optical stimulation and repeated this sequence (sweeps, Fig. 5a(i)) three times. In total, the neuronal responses of nine cells were recorded during optical stimulation (Fig. 5, Supplementary Figs. 10–17). In all preselected cells that were recorded, a significant number of action potentials could be evoked by optical stimulation and all cells showed a significant response when the duration of the stimulation was a least 10 ms (one-sided Wilcoxon rank-sum test, all $p$ values < 0.05, for details, see

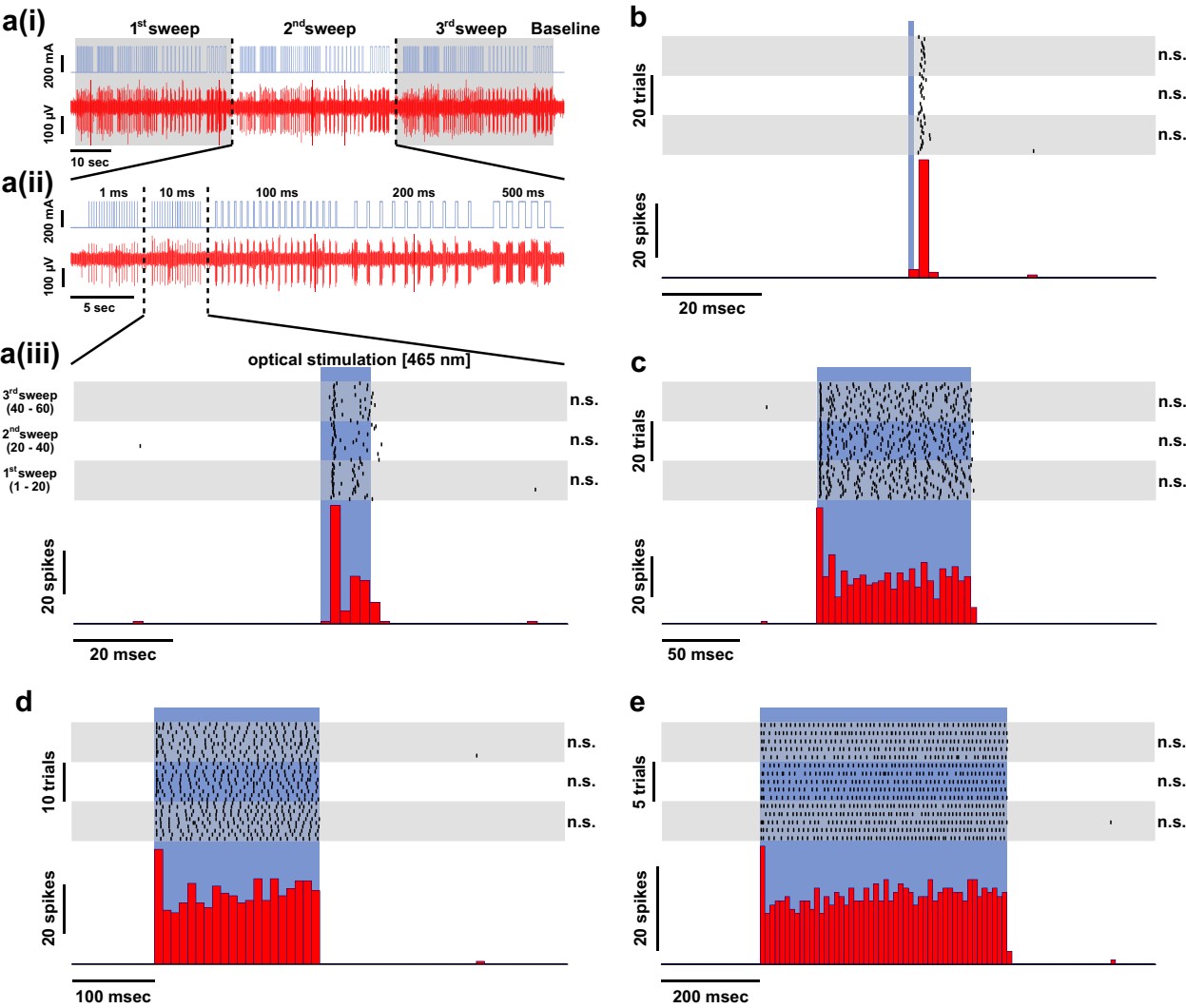

**Fig. 5 Single-cell responses upon optical stimulation (Cell 4). a(i)** Stimulation protocol (upper trace) and the resulting evoked cellular responses (lower trace). The stimulation protocol consisted of repeated light pulses of different durations (1, 10, 100, 200, and 500 ms) and was repeated three times (sweep 1–3, indicated by the gray/white background). The baseline-firing rate of each neuron was assessed before/after the light stimulation protocol. **a(ii)** A single sweep (i.e., sweep 2) of the optical stimulation protocol. The 10 ms optical stimulation is highlighted and further analyzed in **(a(iii))**. Raster plot (upper part) and peri-stimulus time histogram (PSTH; lower part) of the cell response. The raster plot characterizes the cellular response aligned with the onset of the optical stimulation (465 nm at 300 mA; blue shaded area). Each line represents one optical stimulation. Each dot within that line represents an evoked action potential. For each repetition of the stimulation, a new line is added to the plot. The gray/white background of the raster plot indicates the blocks of repetition (sweep 1–3). To assess the variability of the evoked responses, the sweeps were statistically compared. In the case of this cell, no statistical differences were found (indicated by the abbreviation n.s. next to the raster plots). The PSTH represents the summed responses within a certain time window (bin). In case of the 10 ms stimulus presentation, the bin width is 2 ms. **b** Raster plot (upper trace) and PSTH (lower trace) for the 1 ms stimulus duration (bin width: 2 ms). **c** Raster plot (upper trace) and PSTH (lower trace) for the 100 ms stimulus duration (bin width: 4 ms). **d** Raster plot (upper trace) and PSTH (lower trace) for the 200 ms stimulus duration (bin width: 10 ms). **e** Raster plot (upper trace) and PSTH (lower trace) for the 500 ms stimulus duration (bin width: 10 ms).

Supplementary Table 3). To assess the variability of the evoked neuronal responses, we compared the spikes evoked during these sweeps using a nonparametric analysis of variance (Kruskal–Wallis test). If we found significant differences between the sweeps a Bonferroni-corrected multiple comparison test was conducted. In only two out of the nine cells, significant differences between sweeps in some conditions were detected (see Supplementary Table 3). The differences were found in stimulation trials of longer duration (>100 ms). For stimulation durations below 100 ms no significant differences could be detected (the significance is indicated in Supplementary Table 3 and in the single-cell raster plots). Overall, the evoked responses were robust.

The recorded cells differed in their overall response properties (Supplementary Figs. 10–18). We found cells that responded throughout the entire optical stimulation period with a constant amount of spikes, albeit showing a pronounced peak of activation at the onset of light stimulation (Fig. 5, Supplementary Figs. 15 and 17, Supplementary Fig. 18 cells 4, 7, and 9). Further, we found cells that weakened their responses over the course of prolonged stimulation (Supplementary Figs. 10–12, Supplementary Fig. 18 cells 1–3). Another response pattern that was found showed a sharp peak only during the onset of the stimulus (Supplementary Figs. 13, 14, and 16, Supplementary Fig. 18 cells 5, 6, and 8). After the electrophysiological experiments were

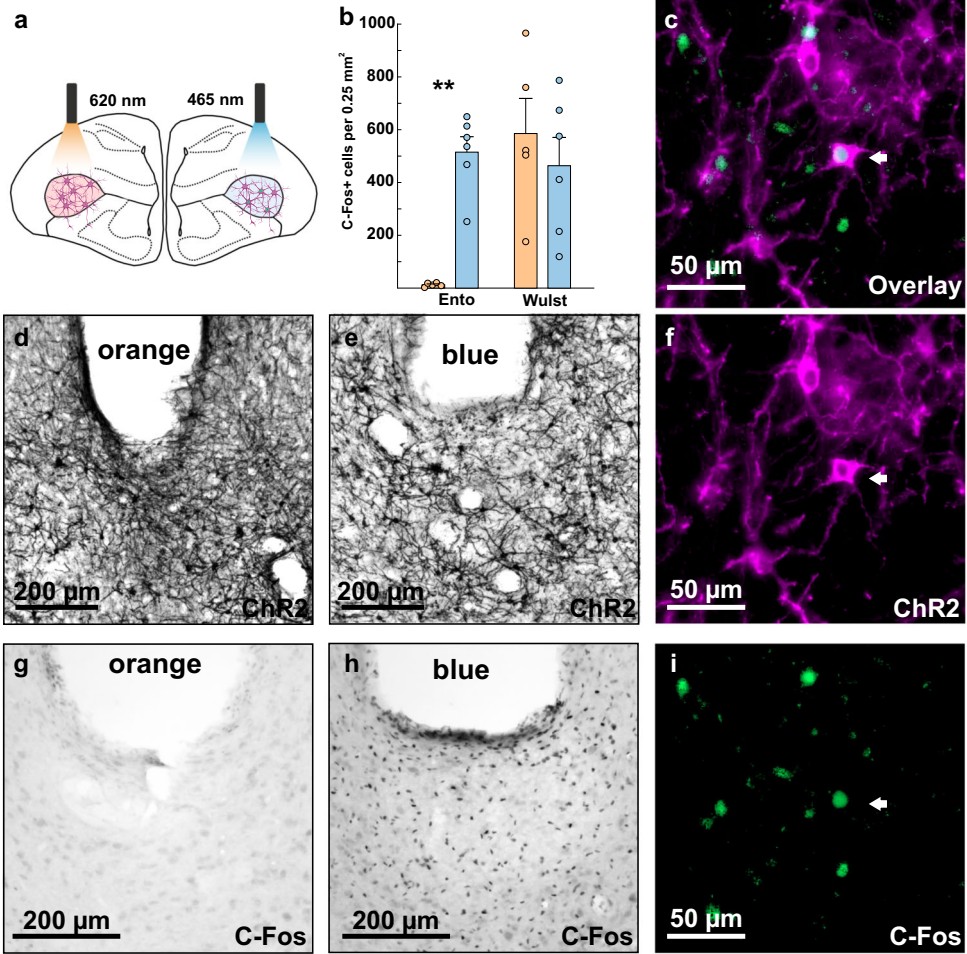

**Fig. 6 Immediate early gene expression after stimulation of ChR2 expressing cells with blue light (465 nm). a** Schematic drawing illustrating the experimental procedure. In three animals, two sites of one hemisphere were stimulated with orange light (620 nm), while two sites in the other hemisphere were stimulated with blue light (465 nm). **b** Quantification of c-Fos expression following orange and blue light stimulation in the stimulated entopallium and a non-stimulated control area in the visual wulst. Significantly more c-Fos expression was observed in the blue light stimulated hemisphere ($n = 6$ sites, blue) compared to the orange stimulated hemisphere ($n = 5$ sites, orange). However, in the non-stimulated visual wulst c-Fos activation was comparable. **c** Overlay of ChR2 expression with c-Fos expression (fluorescence) indicating that blue light stimulation resulted in cellular activity in ChR2 expressing cells. **d** ChR2 expression in the orange stimulated hemisphere. **e, f** ChR2 expression in the blue light stimulated hemisphere. **g** Little c-Fos expression occurred in the orange stimulated hemisphere. **h, i** Extensive c-Fos expression occurred in the blue light stimulated hemisphere. Scale bars are specified within the images. Error bars represent the standard error of the mean (SEM), dots represent the raw data. **$p < 0.01$.

finished, histology was performed to check for ChR2 expression in the entopallium (Supplementary Fig. 19).

We decided to use a pulsed optical stimulation protocol of 40 Hz in our behavioral experiments, as some cells exclusively showed a sharp onset peak and pulses of 10 ms durations reliably evoked spikes in all recorded cells (pulse duration: 15 ms; inter-pulse interval: 10 ms). The physiological validity of the applied protocol was further verified in an additional experiment. Here, we investigated the functionality of ChR2 with immediate-early gene expression in awake pigeons (Fig. 6). Following a sensory deprivation phase of one hour, pigeons were stimulated for a period of 30 min with alternating intervals of 5 min 40 Hz stimulation and 5 min no stimulation with orange light in one hemisphere and blue light in the other hemisphere (Fig. 6a). After that, the pigeons were sensory deprived for a further 60 min to allow for adequate c-Fos expression and subsequently transcardially perfused with PFA. Sensory deprivation before and after the experiment was performed to reduce stimulation unrelated c-Fos expression. Subsequently, stainings against the immediate early gene c-Fos

were performed. The cellular activation was assessed at two stimulation sites in each hemisphere of three pigeons within the stimulated entopallium and in a control area within the unstimulated visual wulst to make sure that both hemispheres show comparable levels of c-Fos expression in general. Blue light stimulation resulted in increased c-Fos expression beneath the cannula within the entopallium (Fig. 6b, c, f, i) compared to orange light stimulation (blue light stimulation: 515 cells ± 59 SEM, $n = 6$; orange light stimulation: 12 cells ± 3 SEM, $n = 5$; $Z = 2.739$, $p = 0.004$, Fig. 6b, g, h), although both hemispheres showed reliable ChR2 expression (Fig. 6d–f). In the control area within the visual wulst, there was no difference between orange and blue light stimulation, indicating that staining intensities were similar between the two hemispheres (blue light stimulation: 464 cells ± 107 SEM, $n = 6$; orange light stimulation: 585 cells ± 133 SEM, $n = 5$, $Z = -0.548$, $p = 0.662$, Fig. 6b).

**Transient activation of ChR2 expressing cells in the entopallium reduces contrast sensitivity indicated by impaired**

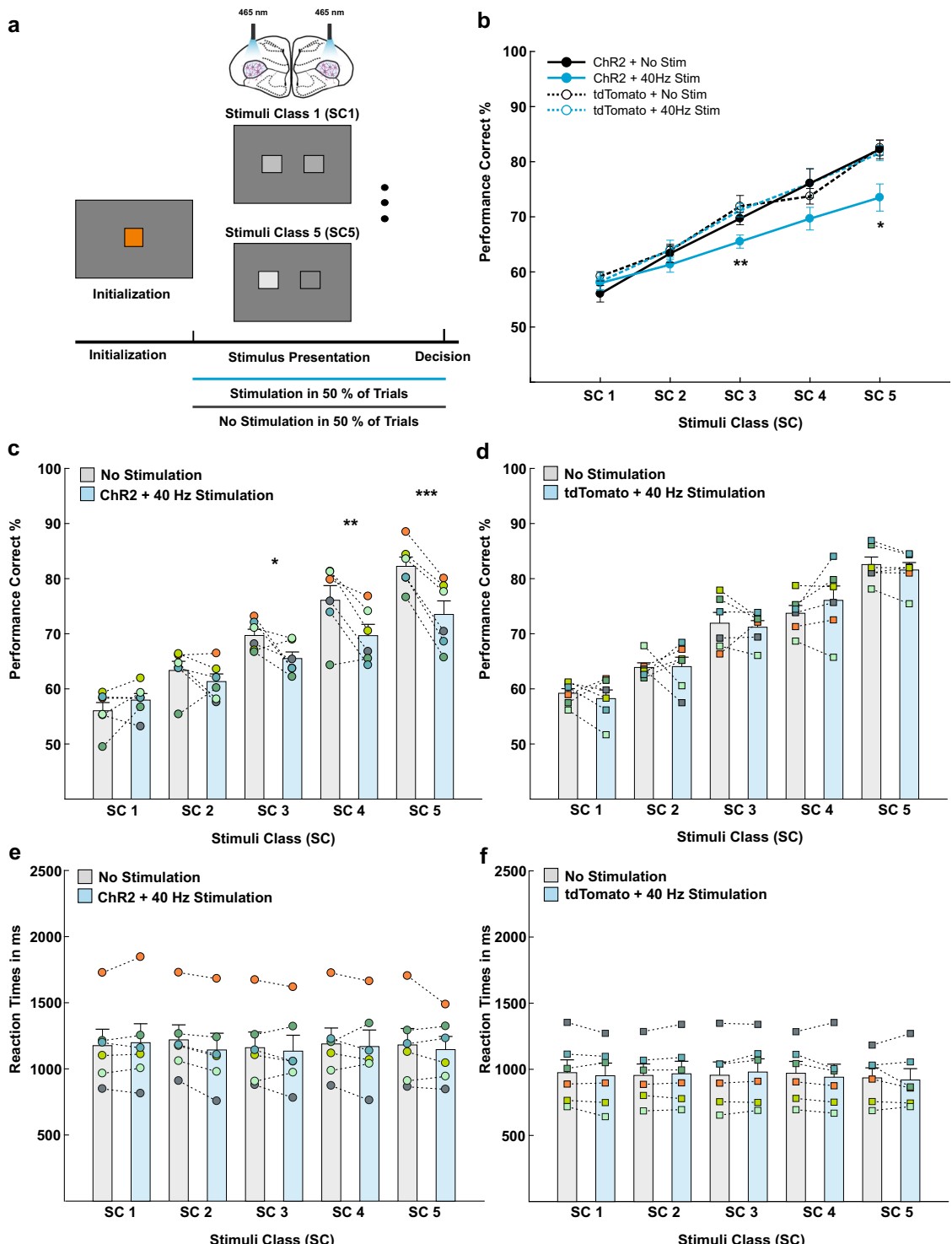

**performance in a grayscale visual discrimination task**. For all behavioral experiments, AAV1-CAG was used to deliver the ion channel ChR2 or the fluorescent protein tdTomato into neurons of the entopallium. To investigate the behavioral effect of optogenetic stimulation, an experimental group expressing ChR2 and a control group expressing tdTomato were bilaterally stimulated in the entopallium during the whole stimulus presentation phase or until the pigeon responded. Stimulation took place in 50% of the trials of a forced-choice visual discrimination task. The remaining 50% of the trials were void of optical stimulation (Fig. 7a).

In every trial of a session, pigeons were confronted with two grayscale pictures that varied in their luminance. The grayscale picture pairs could belong to five different stimulus classes (SC), depending on their luminance difference. SC1 consisted of grayscale pictures with the lowest luminance difference (mean luminance difference 5.81 cd/m$^2$, Supplementary Fig. 20). Thus, these stimuli were similar to each other and therefore hard to discriminate. In contrast, grayscale picture pairs of the SC5 showed the highest luminance difference (mean luminance difference 30.39 cd/m$^2$, Supplementary Fig. 20). Consequently, these stimuli were easier to discriminate (see more details in the

**Fig. 7 Transient 40 Hz activation of ChR2 expressing cells in the avian entopallium reduces contrast sensitivity. a** Schematic illustration of the experimental procedure. Pigeons were conditioned to discriminate grayscales of different stimulus classes (SC). SC1 consisted of grayscale pictures that were difficult to discriminate and SC5 consisted of grayscale pictures that were easy to discriminate. Pigeons were bilaterally stimulated in half of the trials in a given session and stimulation took place during the whole stimulus presentation phase or until the animal responded. **b** Visual discrimination performance of the control ($n = 6$) and experimental group ($n = 6$) displayed in one graph. Experimental group data can be seen in more detail in panel (**c**) and control group data can be seen in more detail in panel (**d**). The control and experimental group had comparable performances and a significant drop in performance could only be seen for optogenetic stimulation in the experimental group in SC3 and SC5. **c** Visual discrimination performance of the experimental group expressing ChR2 in the entopallium. Optogenetic stimulation reduced contrast sensitivity as indicated by a significant reduction of discrimination accuracy for stimuli in SC3, SC4, and SC5, but unimpaired discrimination performance in SC2 and SC1. **d** Visual discrimination performance of the control group expressing tdTomato in the entopallium. Optogenetic stimulation had no effect on contrast sensitivity as discrimination performance was comparable between stimulated and unstimulated trials in all stimulus classes. **e, f** Reaction times for stimulated and unstimulated trials of the experimental ($n = 6$) and control group ($n = 6$). There was no difference in reaction times between the different stimuli classes or between stimulated and unstimulated trials for both groups. **c–f** Mean performances of all sessions are plotted for all pigeons in individual colors. Stimulated and unstimulated performances within each pigeon have been connected with lines. Error bars represent the standard error of the mean (SEM), \*\*\*$p < 0.001$, \*\*$p < 0.01$, \*$p < 0.05$.

| **Table 2 Effects of optogenetic stimulation on grayscale visual discrimination.** | | | | | |
|---|---|---|---|---|---|
| | **SC1** | **SC2** | **SC3** | **SC4** | **SC5** |
| All groups | 57.86 ± 0.77 | 63.15 ± 0.73 | 69.57 ± 0.81 | 73.88 ± 1.45 | 79.97 ± 1.22 |
| *Exp. group* | | | | | |
| 40 Hz stimulation | 57.97 ± 1.19 | 61.32 ± 1.38 | 65.50 ± 1.19 | 69.66 ± 2.05 | 73.50 ± 2.46 |
| No stimulation | 56.03 ± 1.49 | 63.35 ± 1.66 | 69.69 ± 1.13 | 76.08 ± 2.66 | 82.23 ± 1.69 |
| *p* Value | 0.205 | 0.334 | 0.028* | 0.006** | <0.001*** |
| *Control group* | | | | | |
| 40 Hz stimulation | 58.23 ± 1.57 | 64.04 ± 1.71 | 71.19 ± 1.20 | 76.06 ± 2.61 | 81.57 ± 1.35 |
| No stimulation | 59.20 ± 0.84 | 63.88 ± 0.84 | 71.91 ± 1.96 | 73.72 ± 1.41 | 82.55 ± 1.37 |
| *p* Value | 0.513 | 0.937 | 0.665 | 0.228 | 0.270 |

Mean performance in % and SEM for all five stimulus classes for the experimental and control groups. Both groups consisted of six animals each that were repeatedly tested in five sessions. \*\*\*$p < 0.001$, \*\*$p < 0.01$, \*$p < 0.05$.

method section). The pigeons were conditioned to peck on the darker image of the given comparison and tested in sessions consisting of 600 trials until five sessions meeting the behavioral criteria were gathered. All sessions in which pigeons participated in at least 50% of the trials and their performance reached 75% in one SC were analyzed. A repeated-measures ANOVA with the within-subject factors 40 Hz stimulation (stimulated and non-stimulated trials), session (behavioral sessions 1–5) and SC 1–5 and the between-subject factor group (experimental group and control group) was calculated.

For the grayscale discrimination, animals belonging to the experimental ($n = 6$) and the control group ($n = 6$) showed a similar overall behavioral performance (no main effect of the between-subject factor group $F_{(1,10)} = 2.475$, $p = 0.147$, $\eta_p^2 = 0.198$). Further, the choice performance of all five behavioral sessions of each group was comparable and not confounded with learning effects (no main effect of session $F_{(4,40)} = 0.802$, $p = 0.531$, $\eta_p^2 = 0.074$), or long-term effects of the stimulation (no interaction between stimulation and session $F_{(4,40)} = 0.626$, $p = 0.647$, $\eta_p^2 = 0.059$). However, the discrimination performance for the five different SC differed significantly for all tested groups (main effect of SC $F_{(4,40)} = 181.015$, $p < 0.001$, $\eta_p^2 = 0.948$) and gradually increased from SC1 to SC5 (all Bonferroni corrected comparisons $p < 0.007$, for mean values and SEM see Table 2).

Importantly, optical stimulation resulted in different effects for each tested group (interaction of the between-subject factor group and the within-subject factor 40 Hz stimulation, $F_{(1,10)} = 8.703$, $p = 0.015$, $\eta_p^2 = 0.465$). We found no impact of optical stimulation on the behavioral performance of the animals in the control group expressing the fluorescent protein tdTomato (no stim: 70.25% ± 1.23 SEM, stim: 70.22% ± 1.37 SEM, $p = 0.970$; Bonferroni corrected pairwise comparisons) indicating that blue light itself had no impact on visual discrimination performance.

However, optical stimulation of the animals in the experimental group expressing ChR2 reduced contrast sensitivity indicated by significantly impaired discrimination accuracy (no stim: 69.48% ± 1.23 SEM, stim: 65.59% ± 1.39 SEM, $p = 0.002$; Bonferroni corrected pairwise comparisons).

Optical stimulation differentially impaired the behavioral performance in the different SC (interaction between the factors 40 Hz stimulation, SC and group $F_{(4,40)} = 5.184$, $p = 0.002$, $\eta_p^2 = 0.341$, Fig. 7b–d, Supplementary Fig. 21). In the experimental group, 40 Hz stimulation impaired the discrimination accuracy in SC5 ($p < 0.001$, Fig. 7c), SC4 ($p = 0.006$, Fig. 7c), and SC3 ($p = 0.028$, Fig. 7c). However, stimulation had no effect on the discrimination accuracy in SC2 ($p = 0.334$, Fig. 7c) and SC1 (Fig. 7c, $p = 0.205$, Bonferroni corrected pairwise comparisons; see Table 2 for mean values and SEM). The stimulation effect was robust and occurred in at least one SC in every test session (Supplementary Fig. 21). In contrast to the experimental group, there was no effect of the optical stimulation in any SC in the control group (all $p$ values > 0.227; Fig. 7d, Supplementary Fig. 21, see Table 2 for mean values and SEM). Furthermore, Bonferroni corrected pairwise comparisons revealed that the performance of the experimental and control group differed significantly for stimulated trials in SC3 ($p = 0.007$, Fig. 7b) and SC5 ($p = 0.017$, Fig. 7b). For SC4 there was a trend toward significance ($p = 0.082$, Fig. 7b).

In the next step, we determined whether the drop in performance in the experimental group was associated with an increase in reaction times that could have reflected attentional deficits. Therefore, we measured the reaction times in all SC and calculated a repeated-measures ANOVA with the same factors as described above.

We found that there was no main effect of the session ($F_{(3,30)} = 0.738$, $p = 0.538$, $\eta_p^2 = 0.069$), 40 Hz stimulation ($F_{(1,10)} = $

2.292, $p = 0.161$, $\eta_p^2 = 0.186$) and SC ($F_{(4,40)} = 1.439$, $p = 0.239$, $\eta_p^2 = 0.126$) on the reaction times. Furthermore, there was no interaction between 40 Hz stimulation and SC ($F_{(4,40)} = 0.702$, $p = 0.595$, $\eta_p^2 = 0.066$, Fig. 7e, f, Supplementary Fig. 22) indicating that reaction times were comparable between stimulated and unstimulated trials as well as between all SC in the experimental and control groups (Fig. 7e, f). Furthermore, the groups did not differ significantly in their overall reaction times (no main effect of group $F_{(1,10)} = 2.067$, $p = 0.181$, $\eta_p^2 = 0.171$, no interaction of stimulation, SC and group $F_{(4,40)} = 2.139$, $p = 0.094$, $\eta_p^2 = 0.176$, Supplementary Fig. 22). A different pulse protocol with 20 Hz stimulation was tested in five pigeons and was also able to impair behavioral performance for stimuli in SC5 (see Supplementary Fig. 23). At the end of experimental testing, for both the experimental and control group, coordinates of the cannula tips were assessed and these were mainly located in the medial to anterior entopallium (see Supplementary Fig. 24).

## Discussion

In the current study, we investigated the efficiency of three AAVs (AAV1, AAV5, and AAV9) in combination with two different promoter systems (hSyn and CAG) in their ability to drive ChR2 expression in neurons within the entopallium, the first visual input structure of the avian forebrain. This comparison was performed to determine the optimal viral construct for optogenetic experiments in the visual system of birds. We found that AAV1 was the most efficient viral vector regardless of the promoter system as this vector transduced the greatest number of cells and showed the highest transfection density compared to the other AAV constructs. When AAV1 was used in combination with the CAG promoter, we observed extensive anterograde as well as retrograde labeling that was weaker when AAV1 was combined with the hSyn promoter. Since several studies have reported difficulties in producing behavioral effects with optogenetic stimulation in primates, we furthermore wanted to confirm that stimulation of ChR2 could produce physiological as well as behavioral effects in pigeons. The physiological effect of ChR2 was verified with combined optical stimulation and electrophysiological recordings as well as with immediate-early gene expression. The behavioral effect was investigated in a grayscale visual discrimination task. We could show that contrast sensitivity decreased when neural activity was temporarily increased in the entopallium. This finding provides causal evidence for the involvement of the entopallium in contrast perception as well as a proof of principle for optogenetics in pigeons.

The finding that AAV1 was the most efficient construct in transducing cells in the entopallium of pigeons is in line with several studies showing that this construct is also highly efficient in primates[36], cats[39], mice[38], and rats[34,37]. Nevertheless, in primates[25,36] and rodents[34] AAV5 seems to be even more efficient indicating differences in viral transfection between species, as AAV5 transduced only few cells in pigeons and has been shown to be completely ineffective in cats[39]. While AAV9 is widely used in rodents, especially because of its capacity to cross the blood–brain barrier[47–49], and has also been used in zebra finches[7,44], our study indicates that AAV9 combined with the hSyn promoter was completely inefficient in driving transgene expression in cells of the entopallium in pigeons. In contrast to this, AAV9 combined with the CAG promoter led to moderate transgene expression in the entopallium, but also to a severe reduction of NeuN and activation of astrocytic cells indicating neurotoxicity of this serotype. The interpretation that the cell loss occurred as a consequence of an inflammatory response is corroborated by other transfection studies showing that AAV9 encoding nonself proteins can result in cell-mediated immune

responses in rodents as well as in primates[45,46]. Those studies concluded that AAV9 is able to transduce antigen-presenting cells (APC) and can trigger a cell-mediated immune response depending on the immunogenicity as well as the expression level of the transgene[45]. Our finding that the inflammatory response only occurred when AAV9 was used in combination with the CAG promoter, but not when used with the hSyn promoter, supports this idea. This is the case as the CAG promoter drives strong transgene expression in various cell types, whereas the hSyn promoter has been shown to be neuron-specific[25,34,49–51], making it unlikely that transgene expression occurred in APC. Thus, it is still possible that AAV9 combined with other promoters provides a useful optogenetic tool. Indeed, several studies conducted in zebra finches use scAAV9 in combination with neurexin or CMV promoter in areas of the zebra finch song system[7,44]. However, our results cannot easily be compared to these studies, as they used custom-built self-complementary AAVs (scAAVs), which differed not only in the integrated promoters but also in other aspects, as scAAVs do not require second strand synthesis but are immediately ready for replication and transcription.

For our purposes, AAV1-CAG was the most efficient construct and was therefore tested in other areas of the pigeon brain to investigate its brain-wide usefulness for optogenetic experiments. We found that AAV1-CAG was effective in driving transgene expression in cells of the visual wulst, the entopallium, the globus pallidus, the hippocampus, the striatum, and the nidopallium caudolaterale. Importantly, ChR2 expression following injections of AAV1 combined with the unspecific CAG promoter occurred mainly in neurons indicating that AAV1 has a natural tropism for this cell type. That AAV1 transfects mainly neurons has already been reported in other studies[38] supporting the idea that this serotype is a useful tool for optogenetic experiments.

In birds, AAV1 has not been used in the visual system so far. However, AAV1 has recently been used in zebra finches in the song system to study how optogenetic stimulation of ventral tegmental area (VTA) axon terminals can guide learned changes in song[43]. This study reported anterograde labeling in Area X following injections of AAV1-CAG into the VTA[43]. This is in line with our finding that injections of AAV1-CAG result in extensive anterograde ChR2 expression in well-described target structures of the entopallium such as the NI, MVL, and the striatum[52]. In addition, our study observed extensive retrograde ChR2 expression in input structures of the entopallium such as the nucleus rotundus[53]. The projection from the diencephalic nucleus rotundus into the telencephalic entopallium is well known and ideal to test the retrograde properties of a viral vector as those two structures are located in different parts of the brain[54]. Thus, it is impossible that the retrogradely labeled cells that have been observed in the nucleus rotundus are the result of injection leakage from the entopallium. The finding that AAV1 in combination with the CAG promoter leads to retrograde ChR2 expression is in line with transduction studies performed in mice and rats that have reported similar properties for this viral vector[55,56]. Retrograde properties of AAV1 in birds have, to the best of our knowledge, so far not been reported. However, in the song system of zebra finches, AAV9 has been used to transduce cells retrogradely[44]. Our findings indicate that AAV1 in combination with the CAG promoter provides a useful alternative for those who depend on commercially available products. Identifying anterograde as well as retrograde properties of AAVs in birds is crucial since this is a way of gaining specificity, which can be used for detailed circuit analyses.

We found that 20 Hz as well as 40 Hz blue light optical stimulation of ChR2 expressing neurons within the entopallium

resulted in impaired grayscale visual discrimination, while the same manipulations in control birds expressing tdTomato did not result in any behavioral effects. This indicates that the blue light stimulation itself had no effect on visual discrimination accuracy and that the discrimination deficit in the experimental group can be traced back to the changed physiology of cells within the entopallium. Furthermore, these findings suggest that the entopallium is involved in contrast perception/visual discrimination of luminance, which is well in line with lesion studies that have been performed in the entopallium and other areas of the avian collothalamic pathway[17,19,20,57]. For example, lesions of the entopallium have been shown to reduce the ability to categorize stimuli into bright and dim[19] or to discriminate between pictures of varying patterns and luminance[20]. However, in most cases, the lesions were not confined to the entopallium, complicating the attribution of the observed perceptual and behavioral deficits to the functionality of the entopallium alone. Using optogenetics, a stronger claim for the role of the entopallium in contrast perception can now be made, as we selectively increased the firing rates of entopallial cells. This procedure has been shown to be a suitable approach for disrupting behaviors that rely on heterogeneous as well as time-varying population coding[7,58,59]. Visual processing depends on such adaptive coding mechanisms since neurons within the sensory system vary their responses dynamically according to the input[60]. In visual discrimination tasks, where one stimulus needs to be selected over the other, cells in higher visual areas code for this discrimination by selectively increasing and decreasing their firing rates for the selected and nonselected stimuli, respectively. The difference in firing rates of these cells represents the discriminative information, which is greater for simple compared to complicated discriminations[61]. Since the optogenetic manipulation within the entopallium was not cell type-specific, the discriminative information of these cells was probably disrupted by elevating the firing rates of all cells, thereby reducing their contrasts. This reduced contrast in neural representations might have led to impaired discrimination accuracy. Thus, for cell type unspecific optogenetics during visual discrimination, excitatory optogenetic tools might offer advantages over inhibitory optogenetic tools, as excitation creates a new signal that can disturb the population dynamics more intensely than inhibition, which might result in floor effects[62]. Although excitatory optogenetic tools offer the above-mentioned advantages and might yield comparable effects to temporal lesions[63], it needs to be noted that these tools might have knock-on effects on connected circuit areas of the entopallium such as the NFL, NI, and MVL[64]. Thus, to further substantiate the role of contrast perception to the entopallium, similar experiments using inhibitory tools or pharmacology could be performed. Moreover, future optogenetic studies could investigate the effect of optogenetic stimulation on other entopallial functions such as motion, pattern, and color discrimination. In these tasks, functional segregation has been described as lesions within the anterior entopallium affect the pattern, color, and form discrimination, whereas lesions of the posterior entopallium impair motion processing[18,21]. Based on these findings and comparable topographic projection patterns within the avian and mammalian tectofugal pathways, it has been proposed that the motion-sensitive posterior entopallium is comparable to area MT, whereas the color/form/pattern sensitive anterior entopallium is comparable to V2, V3, and IT in mammals[21]. As brightness sensitive neurons of the nucleus rotundus primarily project to the anterior entopallium, brightness processing has also been linked to this subdivision[21]. Although not showing a functional segregation, our study confirms that the anterior entopallium is involved in brightness perception as our neuronal manipulation was primarily focussed on this region.

The finding that this area governs brightness perception furthermore suggests a functional similarity of the entopallium to mammalian V1[65].

With our finding, we furthermore provided, to the best of our knowledge, the first proof of principle for the functionality of optogenetics in pigeons. Since the functional implementation of optogenetics in other species, especially on a behavioral level, has been challenging[24,27], the establishment of this method in a novel species is interesting in itself. While attempts to drive behavior in primates with optical stimulation of the motor cortex have failed[25], the production of saccadic eye movements with stimulation of V1 was successful[66]. Based on this, it has been suggested that, especially for unspecific optogenetic stimulation of both inhibitory and excitatory cell types, manipulations within sensory areas are more effective in driving behavior than manipulations within motor structures[66]. This is the case as unspecific stimulation is not able to produce finely tuned action plans that are necessary to drive motor behavior[58], whereas unspecific stimulation of visual areas can produce artificial percepts that attract attention, therefore, changing behavior[66]. The idea of artificial percepts and their attentional effects might also apply to our finding that stimulation of the primary visual area in the pigeon brain impaired contrast perception in a grayscale discrimination task. However, as the reaction times in this task did not differ between stimulated and unstimulated trials it is unlikely that our findings are simply the results of averted attention but rather of deficient contrast coding within the entopallium.

However, the fact that unspecific stimulation is not able to produce finely tuned action plans that are necessary to drive motor behavior[58] also indicates the importance of cell type or projection specific approaches when investigating behaviors that rely on tightly regulated mechanisms. In order to investigate the function of the columnar and laminar organization of the bird brain that has been proposed by several studies[13,67,68], projection specific approaches are needed. With projection specific optogenetics, the role of specific visual circuits between layers can be investigated to establish circuit–function relationships that highlight general principles of brain organization. With our finding that AAV1 in combination with the CAG promoter is able to drive ChR2 expression anterogradely as well as retrogradely, we provided a useful tool for future studies that can use this serotype to investigate whether the conserved principle of a columnar and laminar organization in avian sensory areas also translates to similar functions. With the viral comparison study, we furthermore provided the groundwork for various other methods that rely on viral gene transfer, such as DREADDS[24], genetic ablations[69], calcium imaging[69], or local genetic knock-in/out studies[5], that can from now on be applied in the pigeon brain and can vastly improve comparative research. Overall, we created the foundations for a mechanistic understanding of the avian pallium and demonstrated conserved principles of avian and mammalian visual systems encouraging the use of avian model organisms for comparative and vision research in the future.

## Methods

**Experimental subjects**. For this study, $N = 35$ adult homing pigeons (*Columba livia*) of undetermined sex were obtained from local breeders. The pigeons were between 1 and 4 years of age. They were individually caged and placed on a 12-h light–dark cycle. During the time period of training and testing, the birds were maintained at approximately 85% of their free-feeding weight. All experiments were performed according to the principles regarding the care and use of animals adopted by the German Animal Welfare Law for the prevention of cruelty to animals as suggested by the European Communities Council Directive of November 24, 1986 (86/609/EEC) and were approved by the animal ethics committee of the Landesamt für Natur, Umwelt und Verbraucherschutz NRW, Germany. All efforts were made to minimize the number of animals used and to minimize their suffering.

**Viral vector injections/cannula implantations**. The anesthesia was initiated with a 7:3 mixture of Ketamine (Ketavet 100 mg/ml, Zoetis GmbH, Berlin Germany) and Xylazine (20 mg/ml Rompun, Bayer Vital GmbH, Leverkusen Germany). Pigeons received intramuscular injections of 0.075 ml for each 100 g bodyweight. This translates to 52.5 mg Ketamine per kg body weight and 4.5 mg Xylazine per kg body weight. Following that, the anesthesia was sustained with a consistent flow of Isoflurane (Forane 100%, Abbott GmbH & Co. KG, Wiesbaden, Germany). Prior to the surgical procedure, the feathers on top of the head covering the target area were cut. As soon as the pigeons no longer showed any pain reflexes, they were positioned in a stereotactic apparatus. At first, the skin covering the head was incised to expose the cranial bone. Then, craniotomies were performed to uncover the brain tissue. Craniotomies were performed at different locations depending on the target structure (entopallium: A + 9.5, L ± 5.5, DV −4.5; hippocampus: A + 6.0, L ± 1.0, DV −1; wulst: A + 11, L ± 1.0, DV −1.5; globus pallidus: A + 9.0, L ± 4.5, DV −6; NCL: A + 5.5, L ± 7.0, DV −2.5; medial striatum: A + 11.0, L ± 1.5, DV −6.0[54]). After the removal of the dura mater injections were made at the specific locations. For the viral transfection study 5 µl of each viral vector (AAV1.hSyn.hChR2 (H134R)-eYFP, AAV1.CAG.hChR2(H134R)-mCherry.WPRE.SV40, AAV5.hSyn. hChR2(H134R)-eYFP, AAV5.CAG.hChR2(H134R)-mCherry.WPRE.SV40, AAV9. hSyn.hChR2(H134R)-eYFP, AAV9.CAG.hChR2(H134R)-mCherry.WPRE.SV40 (all vectors were obtained from Addgene, Watertown, USA, all titer were ≥1 × $10^{13}$ vg/ml)) were pressure injected into at least five separate hemispheres of three pigeons. Therefore, 5 µl were pipetted on parafilm and drawn into a glass pipette with a 25 µm tip. The total volume was distributed over a range of 500 µm dorsal/ ventral. For the behavioral and electrophysiological experiments, 10 µl of AAV1. CAG.hChR2(H134R)-mCherry.WPRE.SV40 (Addgene, Watertown, USA, all titer ≥ 1 × $10^{13}$ vg/ml) was injected bilaterally into the entopallium (5 µl at A + 9.5, L ± 5.5, DV −4.5 and 5 µl at A + 10.5, L ± 6.0, DV −4.5). The control pigeons received injections of AAV1-CAG-tdTomato (Addgene, Watertown, USA) instead. The pigeons used in behavioral experiments were furthermore bilaterally implanted with dual fiberoptic cannula (DFC_200/245-0.53_12mm_GS0.7_FLT, Doric Lenses Inc, Quebec, Canada). Therefore, stainless-steel screws (Small Parts, Logansport, USA) were fixed to the skull to serve as anchors for the dental cement. After that, the cannulas were inserted vertically into the entopallium (A + 9.5, L ± 5.5, DV −4.5). Finally, the cannula was attached to the head with dental cement. After the surgery was completed, pigeons received analgesics for three consecutive days (0.5 ml 10 mg/ml Rimadyl, Pfizer GmbH, Münster, Germany). The recovery period lasted 14 days, in which the pigeons had access to food and water ad libitum. After the recovery period, the pigeons continued training to get accustomed to the patch chords.

**Perfusion**. Animals in the viral transfection study were perfused 6 weeks after the injections to ensure that ChR2 expression was already stable (see Supplementary Fig. 25), whereas pigeons from the behavioral paradigm were perfused after all experimental tests were completed. We found that ChR2 expression was stable up to 12 months (see Supplementary Fig. 26). After pigeons had been deeply anesthetized with Equithesin (0.45 ml per 100 g body weight), the transcardial perfusion was initiated with 0.9% sodium chloride (NaCl) and followed by cold (4 °C) 4% paraformaldehyde (PFA) in 0.12 M phosphate buffer (PB; pH 7.4). After the blood had been successfully exchanged with PFA, the brain was removed from the skull and stored in a postfix solution (4% PFA with 30% sucrose) at 4 °C for the next 2 h. After that, all brains were stored in a 30% sucrose solution in phosphate-buffered saline (PBS; pH 7.4) for at least 24 h to remove water for cryoprotection. Before slicing, brains were embedded in 15% gelatin/30% sucrose and stored in postfix for 24 h. Finally, brains were cut in a coronal plane in 40 µm-thickness using a freezing microtome (Leica, Wetzlar, Germany).

**Immunohistochemistry**. All stainings were performed with free-floating sections and every tenth slice was used for the immunohistochemistry. The counterstaining was performed to allow for an equal comparison between the serotypes, as serotypes with the hSyn promoter were tagged with eYFP, whereas serotypes with the CAG promoter were tagged with mCherry. Moreover, the amount of transgene expression can be underestimated when analyzing native fluorescence, as the signal increases with counterstainings (see Supplementary Fig. 1).

For the 3,3 diaminobenzidine (DAB) reaction, slices were first rinsed (3 × 10 min in PBS) and then incubated in 0.3% hydrogen peroxide ($H_2O_2$) in distilled water for 30 min to block endogenous peroxidases. After that, slices were rinsed (3 × 10 min) and then transferred into 10% normal horse serum (NHS; Vector Laboratories-Vectastain Elite ABC kit) in PBS with 0.3% Triton-X-100 (PBST) for 30 min to block unspecific binding sites. For the comparative viral transduction analysis and for the assessment of anterograde and retrograde transport, slices were incubated with a monoclonal mouse anti-ChR2 antibody (1:1000 in PBST, PROGEN Biotechnik GmbH, Heidelberg, Germany) at 4 °C overnight. For the immediate early gene experiment, slices were incubated with a polyclonal rabbit anti-c-Fos antibody (1:500 in PBST, Santa Cruz Biotechnology, Texas, USA) at 4 °C overnight. The following day, all slices were rinsed in PBS (3 × 10 min) and then incubated at room temperature with the corresponding secondary biotinylated antibody (anti-mouse antibody 1:500 in PBST, anti-rabbit antibody 1:200 in PBST; Vector Laboratories-Vectastain Elite ABC kit) for 1 h. Subsequently, slices were rinsed (3 × 10 min in PBS) and then transferred into an avidin–biotin-peroxidase

complex (Vector Laboratories-Vectastain Elite ABC kit; 1:100 in PBST). Following further rinsing (3 × 10 min), slices were incubated in DAB solution. The solution comprised 5 ml distilled water, 100 µl of DAB stock solution, 84 µl of a nickel solution, and 84 µl of buffer stock solution (Vector Laboratories, DAB Substrate Kit SK-4100). All slices were incubated in 12-well plates, with each well containing 1 ml of this working solution. After adding 6 µl $H_2O_2$ to each well the reaction started and was stopped after 2 min by transferring the slices into 12-well plates containing PBS. Subsequently, slices were rinsed (2 × 5 min in PBS) and then mounted on gelatin-coated slides. Finally, slices were dehydrated in alcohol and coverslipped with depex (Fluka, Munich, Germany)[70].

For the fluorescent stainings, slices were first rinsed (3 × 10 min in PBS) and then incubated in 10% NHS (Vector Laboratories-Vectastain Elite ABC kit) to block unspecific binding sites. After blocking and rinsing (3 × 10 min), the slices for the single ChR2 stainings were incubated with a monoclonal mouse anti-ChR2 antibody (1:1000 in PBST, PROGEN Biotechnik GmbH, Heidelberg, Germany). For the combined ChR2 and NeuN stainings, slices were incubated in a mixture of a monoclonal mouse anti-ChR2 antibody (1:1000 in PBST, PROGEN Biotechnik GmbH, Heidelberg, Germany) and a polyclonal rabbit anti-NeuN antibody (1:500 in PBST, EMD Millipore, Darmstadt, Germany). For the combined ChR2 and c-Fos stainings, slices were incubated in a mixture of a monoclonal mouse anti-ChR2 antibody (1:1000 in PBST, PROGEN Biotechnik GmbH, Heidelberg, Germany) and a polyclonal rabbit anti-c-Fos (1:500 in PBST, Santa Cruz Biotechnology, TX, USA). For the combined ChR2 and GFAP stainings, slices were incubated in a mixture of a monoclonal mouse anti-ChR2 antibody (1:1000 in PBST, PROGEN Biotechnik GmbH, Heidelberg, Germany) and a monoclonal rat anti-GFAP (1:500 in PBST, Invitrogen, Darmstadt, Germany). All primary antibodies were incubated overnight at 4 °C. The next day, after rinsing (3 × 10 min), the slices for the single ChR2 stainings were incubated with a goat anti-mouse antibody AlexaFluor594 (1:500 in PBST, Invitrogen, Darmstadt, Germany). The slices for the combined ChR2/NeuN and ChR2/c-Fos stainings were incubated in a mixture of a goat anti-mouse AlexaFluor594 antibody (1:500 in PBST, Invitrogen, Darmstadt, Germany) and a goat anti-rabbit AlexaFluor488 antibody (1:200 in PBST, Invitrogen, Darmstadt, Germany). The slices for the combined ChR2 and GFAP stainings were incubated in a mixture of a goat anti-mouse AlexaFluor594 antibody (1:500 in PBST, Invitrogen, Darmstadt, Germany) and a goat anti-rat AlexaFluor488 antibody (1:200 in PBST, Invitrogen, Darmstadt, Germany). All secondary antibodies were incubated for 1 h at room temperature. After further rinsing (3 × 10 min), the slices were mounted onto glass slides (Superfrost® Plus, Thermo Scientific) in PBS, and embedded with DAPI Fluoromount-G® (SouthernBiotech, Birmingham, USA), with minimal light exposure to preserve as much fluorescence as possible. Primary antibody specificity was either evaluated with western blots, based on previously published literature or investigated with immunohistochemical stainings in negative control samples. All stainings were performed twice with separate brain series to qualitatively confirm the reproducibility of the stainings.

**Microscopic analysis**. For the quantitative analysis of ChR2 expression, all slices that contained ChR2 expressing cells were imaged bilaterally with 200× magnification using a ZEISS AXIO Imager.M1 with a camera (AxioCam MRm ZEISS 60N-C 2/3"0.63×). All sections containing the entopallium between A + 8.0 and A + 11.25 of one brain series (every tenth slice) were analyzed. For the quantification of somata, every ChR2 expressing soma within the whole entopallium was highlighted in ZEN 2.3 lite and later automatically counted by the program. For all viral vectors, this analysis was performed in one-tenth of the brain sections and the number of all ChR2 expressing soma was summed up for further statistical analysis.

Furthermore, the density of the stained area was determined with ZEN 2.3 pro Image analysis tool wizard. For this, the whole entopallium was selected as an analysis frame and parameters for segmentation were adjusted (smooth: none, sharpen: none, minimum area 1 pixel, threshold: Otsu threshold, separate: none). Following this, the size of all stained particles within the analysis frame was automatically measured in µm². In the same sections, the size of the entopallium was measured in µm² by delineating the borders. The same analysis was performed in images with 100× magnification to determine the density of ChR2 and GFAP expression within the injection site for all cases used in the tropism experiment. For the quantification of NeuN signals in the neuron loss analysis upper and lower thresholds were also adjusted with Otsu's method. However, parameters for segmentation were adjusted and set to (smooth: none, sharpen: delineate, threshold: 0, size: 6, minimum area: 96 pixels, separate: morphology, count: 20). Likewise, the c-Fos signal was quantified in the entopallium and wulst, with the only difference being that the minimum area was adjusted to 24 pixels. For the quantification of anterograde and retrograde transport, the target area was delineated and intensity values were determined. For anterograde transport cases were classified into + little, ++ moderate and +++ extensive (+ signal was <6% darker than the background, ++ signals was 6–12% darker than the background, +++ signal was >12% darker than the background). For retrograde transport cases were classified into + little, ++ moderate and +++ extensive (+ signal was <2% darker than the background, ++ signals was 2–4% darker than the background, +++ signal was >4% darker than the background). All quantifications were performed blinded with the experimenter not being aware of the serotype investigated.

**Electrophysiology**. For extracellular recordings, insulated tungsten wire electrodes with an impedance of ~4 MΩ were used (Product No. #26-05-3; FHC, Bowdoinham, ME, USA). At the tip of each recording electrode, an optical fiber for light stimulation (Doric lenses Inc, Quebec, Canada) was glued. The electrodes were mounted on a micromanipulator (Narishige, MO-8, Tokyo, Japan) and advanced into the brain tissue. During the advancement of the electrode, a constantly repeated light pulse of 1 s duration was presented to evoke spikes. When a responsive cell was encountered, a fixed stimulation protocol was applied. The protocol consisted of repeated light pulses of variable duration (1, 10, 100, 200, and 500 ms) and was repeated three times (sweeps, Fig. 5a(i) and Supplementary Table 2). The resulting neuronal activity was amplified (10.000×) and band-pass filtered (300 Hz–3 kHz) using a differential amplifier for extracellular potentials (DAM 80, World Precision Instruments, Sarasota, FL, USA). In addition, an active notch filter was used to eliminate line noise (HumBug, Quest Scientific, North Vancouver, Canada). Single-cell responses were digitized at a rate of 83 kHz using an analog-digital converter (Cambridge Electronic Design, Micro 1401 mkII). Spikes were sorted offline and analyzed using Spike 2 software (Cambridge Electronic Design, V.8) and Matlab (The Mathworks, Natick, MA, USA). For optical stimulation, we employed LED modules (Plexon Compact modules, 465 nm, Plexon, TX, USA) driven by a control unit (PlexBright 4 Channel Optogenetic Controller, Plexon, TX, USA) using Radiant Software (Plexon, TX, USA). The time course of the light stimulation was synchronously digitized at a rate of 41 kHz for the precise analysis of the evoked cellular responses (Cambridge Electronic Design, Micro 1401 mkII). To compare the number of spikes evoked by optical stimulation with the baseline activity of each cell, a one-sided Wilcoxon rank-sum test was performed in all conditions tested. To assess the variability of the evoked neuronal responses, we compared the spikes evoked during single sweeps using a non-parametric analysis of variance (Kruskal–Wallis test). If we found significant differences between the sweeps Bonferroni-corrected multiple comparison tests were conducted.

**Skinner boxes**. All training and testing were conducted in two operant chambers (32 cm (w) × 34 cm (d) × 32 cm (h)). The operant chambers were equipped with cameras for monitoring purposes, two white house lights on either side on the ceiling, and a touch screen monitor in the front panel. Furthermore, a feeder was positioned in the front panel directly beneath the touch screen where the birds received mixed grains as a food reward for correct responses. A light above the feeder indicated when food was available and served as a second reinforcer. All behavioral paradigms for this experiment were programmed in MATLAB with help of the Biopsychology Toolbox[71].

**Stimuli**. The stimulus set that was used in this study consisted of an orange initialization stimulus and grayscale pairs with varying luminance differences that were divided into five different SC (SC1 mean luminance difference 5.81 cd/m²; SC2 mean luminance difference 12.56 cd/m²; SC3 mean luminance difference 17.96 cd/m²; SC4 mean luminance difference 23.91 cd/m²; SC5 mean luminance difference 30.39 cd/m², see Supplementary Fig. 20). A stimulus pair of one SC was always displayed at the center of the touchscreen computer monitor. The image with the lower luminance of the grayscale pair was assigned to the left or right monitor position in a randomized order. An example SC1 stimulus pair and an example SC5 stimulus pair is depicted in Fig. 7a. All stimuli measured 3.5 cm × 3.5 cm and the grayscale pairs were displayed 5 cm left and right of the center of the screen.

**Grayscale visual discrimination task**. The first training phase was an autoshaping procedure, where the birds learned to peck stimuli from all SC as well as the orange initialization stimulus. After the pigeons had successfully learned to peck all stimuli, they were trained in a grayscale visual discrimination task. During this phase, stimulus pairs with a high luminance difference (SC5) were presented and the pigeon had to learn to peck on the stimulus with the lower luminance. Once the discrimination performance reached 75% pigeons were trained in the final paradigm that contained stimuli of all five SC. Once pigeons showed a stable performance (at least 50% initializations and 75% correct discrimination performance in one SC) they underwent surgery. After 14 days and full recovery, pigeons were trained in the final paradigm again to get accustomed to the patch chords. Behavioral testing was started when 6 weeks since the surgery had elapsed and behavioral performance was back to the criteria. The 6 weeks between surgery and behavioral testing allowed for stable transgene expression.

Every test session contained 600 trials in total. An individual trial started with the illumination of an orange initialization stimulus in the center of the front panel together with a tone indicating the beginning of the trial (Fig. 7a). After pecking the orange stimulus, a grayscale pair of one of the five SC was displayed on the touch screen monitor for 8 s or until a pecking response was detected (Fig. 7a). Pigeons were conditioned to peck on the darker of the two grayscale images. When responding correctly, pigeons received a food reward for 2 s. When responding incorrectly pigeons were punished by turning off the house lights for 2 s. At the end of each trial, a 2 s intertrial interval was employed before the next trial started with the presentation of the orange initialization stimulus and a tone. Pigeons were

stimulated with blue light (465 nm, Plexon) in a randomized order during the stimulus presentation phase in half of the trials (Fig. 7a). Thus, stimulation was stopped at the end of the stimulus presentation (either after 8 s or when pigeons responded to the grayscale images). Furthermore, in every session, an equal number of stimuli belonging to each SC was displayed in a randomized order. Pigeons were assigned to the experimental group (expressing ChR2) and to the control group (expressing tdTomato) in a randomized order. The behavioral performance of both experimental and control pigeons was recorded in MATLAB and was thus not subject to observation. Furthermore, the experimenter that trained and tested the pigeons was not aware of the group allocation.

**Optogenetic manipulations**. A 20 Hz (pulse duration: 15 ms; inter-pulse interval: 35 ms) and 40 Hz (pulse duration: 15 ms; inter-pulse interval: 10 ms) light stimulation protocols were produced with the Radiant software (Plexon, TX, USA). For the behavioral experiments, the entopallium was illuminated with blue light (blue LED, 465 nm, Plexon). For the c-Fos experiment, the entopallium was illuminated with blue light (blue LED, 465 nm, Plexon) in one hemisphere and orange light (orange LED, 620 nm, Plexon) in the other hemisphere for a period of 30 min with alternating intervals of 5 min 40 Hz stimulation and 5 min no stimulation. The LED was mounted on a rotary joint placed in a fixed location above the behavior chamber. Lightweight optical patch chords (BFP(2)_200/220/LMWJ-0.53_1.5m_LC-GS0.7, Doric Lenses Inc., Quebec, Canada) were used to connect the dual fiber optic cannulas with the light source. The light output was controlled with the PlexBright4-channel controller (Plexon, TX, USA). Before cannulas were used for behavioral or electrophysiological experiments, the light output at the cannula tip was verified using a power meter. Before each experiment, the light output of the patch chord was also checked (range of light intensity cannula: 3.5–4.5 mW, range of light intensity at patch chord: 4.5–6 mW, Thorlabs, Newton, USA).

**Statistics and reproducibility**. For the viral transfection study, the sample size was determined based on previously published viral transfection studies that have been published in other species such as rats[34], mice[38], primates[36], and cats[39]. For all viral vectors, a minimum of five injections was performed in separate hemispheres of at least three pigeons. The total amount of ChR2 expressing cells was determined for every injection in one brain series (one-tenth of all brain sections) and then multiplied by ten to estimate the actual amount of ChR2 expressing cells. For the analysis of the transduced area, the total ChR2 expressing area within the entopallium was determined in the same brain series. Furthermore, the size of the entopallium was measured in all sections of one series. The transduced area was then put into relation with the size of the whole entopallium and with the size of the entopallium in sections with ChR2 expression. All serotypes were included in this analysis except for AAV9-hSyn-ChR2, as this serotype did not result in reliable ChR2 expression. We used the Shapiro–Wilk test to test for normal distribution of the data and the Levene's test to test for the homogeneity of the variance. A one-way ANOVA with Welch correction was calculated to investigate the effect of serotype on the number of ChR2 expressing cells and on the ChR2 expressing area. Post hoc tests were Bonferroni corrected. For the NeuN overlay experiments, ChR2 expressing soma co-localized with NeuN, as well as ChR2 cells without co-localization, were counted in one series per injection. The percentage of cells with and without co-localization was determined and compared with a Wilcoxon signed-rank test. For the neuron loss experiment, NeuN signal was quantified in the injection site of AAV9-CAG and AAV1-CAG injections of one brain series per injection. Mean neuron numbers of both serotypes were compared with a Mann–Whitney $U$ test. For the c-Fos experiment, c-Fos signals were quantified at two sites in two brain areas in the blue and orange light stimulated hemispheres in three animals. One stimulated site in an orange stimulated hemisphere could not be reconstructed leading to $n = 6$ for the blue light stimulation and $n = 5$ for the orange light stimulation. Mean c-Fos signals for the blue and orange stimulated hemispheres within the entopallium and within the wulst were compared with a Mann–Whitney $U$ test.

For the behavioral analysis, six control pigeons and six experimental pigeons were tested in five sessions. The sample size was determined on previously published entopallium lesion studies that investigated visual discrimination in birds[17–21]. For the behavioral data analysis, five sessions that matched the criterion of at least 50% trial participation and 75% correct answers in at least one SC were considered. The mean reaction times and the mean performance in stimulated and unstimulated trials for all SC were extracted using MATLAB. We used the Shapiro–Wilk test to test for normal distribution of the data and the Levene's test to test for the homogeneity of the variance. A repeated-measures ANOVA with the within-subject factors session, stimulation, and SC and the between-subject factor group was calculated to investigate the effects on reaction times (in four sessions) and performance (in five sessions). Reproducibility of the behavioral findings was verified by comparing the performance of all tested sessions. Post hoc tests were Bonferroni corrected. Alpha was set at 0.05 for all analyses. All statistical analyses were performed with the software IBM SPSS Statistics (v. 20).

**Reporting summary**. Further information on research design is available in the Nature Research Reporting Summary linked to this article.

## Data availability

The data that support the findings of this study are included in the Article and its Supplementary Information or available from the corresponding author upon request. The source data of the main text figures is available in Supplementary Data 1.

## Code availability

All codes were written in Matlab using Biopsychology Toolbox (available at http://biopsytoolbox.sourceforge.net/). Custom codes will be made available upon request. Please contact the corresponding author.

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

## Acknowledgements

Funded by the Deutsche Forschungsgemeinschaft (DFG, German Research Foundation) —project number 316803389—SFB 1280, projects A01 (O.G and R.P.) and A07 (S.H.), as well as project number 122679504—SFB 874, project B12 (O.M.). We furthermore want to thank Sara Letzner and Sebastian Obst for their help with pilot studies on viral transfection.

## Author contributions

N.R. and R.P. conceived and analyzed the experiments. N.R. conducted viral injections, cannula implantations, and perfusions. N.R., J.M.T., and S.I. performed the histology and microscopic analysis. N.R. and R.P. performed the electrophysiological recordings. N.R., J.M.T., and S.I. performed behavioral training and testing. N.R. and J.M.T. programmed the behavioral paradigm and analysis codes. O.M. and S.H. provided advice and initial technical support and equipment. O.G. and R.P. acquired the funding. N.R. wrote the original draft of the paper. R.P. edited the paper. All authors reviewed the paper.

## Funding

## Competing interests

The authors declare no competing interests.
