## [Peer Review File · Communications Biology]

Reviewers' comments:

Reviewer #1 and #2 (PI/Trainee evaluation; Remarks to the Author):

While optogenetics is now widely used for neuroscience in genetic models like rodents, flies and zebrafish, it is still rarely used in other organisms where efficient viral transduction is more challenging to achieve. This study establishes viral transfer in the pigeon, a common model animal in avian neuroscience, and provides proof of principle for the use of optogenetics to manipulate cellular and behavioral activity through electrophysiology and a contrast perception task. I expect this work to be influential in the field of avian neuroscience as other labs work to adapt these tools for the manipulation of cells, circuits and behavior in the avian nervous system.

Major concerns:

Overall this is a fairly comprehensive study of the efficiency of viral transduction in that a suite of serotypes and a pair of promoters are methodically analyzed to determine characteristics that might best suit future studies. My concerns are suggestions or questions that would make this strong contribution even stronger by providing additional information.

The first concern is lack of reporting sample sizes and quantification of qualitative data. Specifically, I recommend adding the following:

- 1) State how many birds were used in figure 1 (cell/area counting components).
- 2) Provide quantitative information about the level of anterograde and retrograde transport of the channels (Figure 2).
- 3) State how many birds were included in figure 4 (the tropism experiments).
- 4) Provide a quantification of the GFAP signal in figure 5.
- 5) Provide a population analysis for the nine cells in figure 6 (electrophysiology) if possible. Were these all from the same animal?
- 6) State how many birds were used in figure 7 (IEGE).
- 7) Also for figure 7, quantify the change in C-Fos levels between hemispheres.
- 8) Provide the sample size for the individual birds in the contrast perception task (Figure 8).

The second concern pertains to the nature of the data in figure 6. It is very compelling to see that entopallial cells were modulated in response to light. However, it is not explained if these cells are visually modulated in the absence of the optrode stimulus. If these cells were first determined to be visually modulated and then demonstrated to also be optogenetically modulated, then evidence for both types of modulation should be shown. If the assessment of visual modulation was performed but cannot now be quantified, then that process should at least be described. Adding evidence that these are visually mediated cells would explicitly connect the visual response properties of the cells and the optrode response to the behavioral outputs you test later.

The third concern is that the entopallium is expected to have functional and anatomical topography, at least to my understanding. Given this, it was surprising that reporting of locations for recording, stimulating, and tracing was not more explicit. It would be helpful to know locations for the following elements of the manuscript:

- 1) Provide information about recording sites used to generate the data in figure 6. Ideally, this figure would include a panel for a map of the entopallium with all of the recording sites. A supplementary figure with the histology would also be useful. Together, these images would allow the reader to see

if the three populations of cells are in different regions. I recognize that the data might not be available for all of this wish list, but presumably some version of coordinate data can be included. 2) Similarly, it would be helpful to know locations for the canula implant sites in figure 8. After the behavioral tests, was histology performed to see exactly where the canula were implanted within the entopallium? Again, if histology images are available, these should be included in supplementary figures, ideally for each bird.

Adding this information would allow readers to consider your data in the context of reported anatomical and functional topography of the region. If this is possible and proves interesting then it is also recommended that these associations are covered in the discussion.

Minor concerns:

Viral transduction: “morphological abnormalities” maybe worth mentioning possible neurotoxicity right away? Could you provide additional justification for your technique of differentiating neurons from glia? Why didn’t you test an AAV2 serotype as well, as this is commonly used in the zebra finch song system?

Electrophysiology: It might be nice to report what type of recording you’re doing in the manuscript as well as the optical power range. Would like to know the Hz of the LED light pulses, not just duration, since this could impact the biophysics of how channels open and close in the cells—unless it was always just a single pulse? Did you generate a dose-response curve of changes in optical power levels or just pick one level?

IEGE: Additional justification for why you chose the 40Hz (15ms on, 10ms off) stimulus protocol would be helpful. After the pigeons were stimulated by the LED for 30 minutes, they were sensory deprived for 60 minutes afterwards—would you lose some of your IEGE signal by then? Why not perfuse them immediately after the LED stimulus?

Behavior: Would be helpful to report the bilateral implantation in the manuscript, and how long would the 40Hz stimulus last? What does gray-scale different 10 vs. 50 mean? Were the luminance stimulus checked with a light meter? It would also be nice to see the resulting psychometric curve, perhaps in a supplemental figure. Is it necessary to report the groups you used in your ANOVA in the manuscript rather than in the methods section?

Discussion: I would be interested to hear any ideas the authors may have as to why they found 3 “classes” of responses during the electrophysiology as well as why the birds were more effected by the LED stimulus during the more difficult contrast perception tasks as opposed to being equally effected at all difficulty levels.

Methods: During injections, did you always inject in a single track or did you have to do multiple tracks in the same area to fill the entire region? How did you inject your constructs and how did you know the exact volume injected? Did you calibrate your fiber optic canula before implanted to be sure of the optical power output? What is the high end of your power output? Reporting a range would be more helpful than just above 3.5mW

Line 647: I found the language confusing—I did not realize immediately that “randomized stimulation” meant the LED light, not the discrimination stimulus.

Figure 2: d, e, h, k: would be nice to have label of retro vs. antero, e: does this mean no expression in MVL at 6mo because it is no longer highlighted?, e, f, g: mirroring of schematic and image a bit confusing—a label of LSt in the schematic would help orient viewers

Figure 4/5 could be combined into a single figure exploring cellular tropism.

Figure 8e: significance is from experimental group (Chr2 + 40Hz) compared to which control?

Reviewer #3 (Remarks to the Author):

In my opinion this is a fine work reporting excellent news for the avian brain community. The paper is clearly written, and nicely illustrated. Results are convincing, and very promising. Methods, including statistical analysis, look sound and are adequately described. Discussion is also well written, and covers the main aspects of the work presented. I have only minor concerns I will like to point out:

The number of individuals forming the control and experimental groups in the behavioral experiment it is not said. Neither it is said the rate of success of the transfections. How many animals were transfected with the AAV1-CAG construct? How many of these attempts were successful? I think a table containing such info may be very valuable to assess the reliability of the method. Legend of figure 7 describe panel 7g as "illustrating that no c-Fos expression occurred in the orange stimulated hemisphere". However, picture 7g indeed show some dark dots that looks very much alike c-Fos expression. I suggest rephrasing such description.

Reviewer #4 (Remarks to the Author):

Rook et al., provide the first example of optogenetic investigations in pigeons and evaluate a set of serotype and promoter combinations that are currently readily available to the scientific community - making for a valuable resource across labs. They quantify and highlight the usefulness of the AAV1-CAG construct for Chr2 expression in the pigeon brain, particularly, but not limited to, the entopallium. They go on to directly demonstrate the functionality of Chr2 expression to induce activity with light after transfection in the entopallium. Although the electrophysiological data presented is convincing, some additional controls and/or information about the recordings would be useful. Finally, they demonstrate the ability to directly manipulate behaviour by disrupting activity in the entopallium with light stimulation of Chr2 in a context discrimination task. The results are novel in the respect that optogenetic manipulations have not been performed before in pigeons. Although this is not the first demonstration in birds, large differences can exist across avian species and given the import of pigeons as a model, this will be of interest to others in the field - particularly saving time/effort to optimize across various available AAV constructs. In general the paper is well written and includes high quality figures.

General comments:

1. Previous work in finches has been done with the CMV promoter (e.g. Roberts et al., 2012). Is there a specific reason why hSyn and CAG were selected for these studies? It would seem to me that

having a CMV comparison would be beneficial. In the least, the choice of promoters should be justified (e.g. lines 111) and comparison with CMV discussed (e.g. line 357).

2. Can the authors explain the choice to perform statistics across sessions and not animals For Figure 8 and related analysis. Specifically, since the injection site (and subsequent potential for a on/off stim Chr2 effect) for each animal does not change over sessions if there is no effect of learning across sessions, as stated.

3. The choice to evaluate the functional success of the optogenetic stimulation with a task that the entopallium has not previously been determined to play a role in is somewhat confusing. Perhaps demonstrating both a novel role for the nucleus and evaluating the effects of optogenetic stimulation in the same nucleus is somewhat difficult to interpret. Can the authors comment on how they can assign the role of context discrimination directly to the entopallium when there are surely knock-on effects of the stimulation in connected circuit regions. There are some references to lesion studies in the discussion, but these should be elaborated if they want to strengthen the claim that they have established a new role for entopallium in context discrimination. Alternatively, I am not certain this is fundamental to the novelty of the paper and could also be toned down until further confirmatory experiments evaluating more circuit elements/blocking activity are performed.

4. Nevertheless, the results that context discrimination is affected in some way by the optogenetic manipulation are convincing, although it is vital they present the data to be compared stim vs no stim directly following for each session/animal to evaluate the precise effect size of the stimulation itself, not just averaged across groups (as noted in detail below).

5. In general, timelines of various experiments/expression could be clarified throughout. E.g.: Line 112: 6 weeks seems like a long time for AAV expression? Yet, experiments after 6 months would suggest expression continues to increase. What are the practical implications of this? How does this compare to AAV expression in other species? Line 498: the authors chose 6 weeks because this is deemed to be 'stable', but data from 6 months (anterograde and retrograde) would suggest otherwise(?). Could the authors please clarify/reconcile this. Additionally, what was the precise timeline of expression in pigeons used for behavioural tests. The general outline is indicated in the methods, but precise details are missing for the specific data analyzed.

Specific comments:

1. Line 100: please include references for early lesion studies of note.

2. Line 110: '...different AAV serotypes...'

3. Line 202: a very brief indication here of the electrophysiological recording method would be useful. (i.e. extracellular single-unit recordings).

4. Line 203-204: It should be made clear that the goal of these specific experiments is to pre-select cells that are responsive to the light and assess their characteristics – not to do an unbiased assessment of what proportion of cells in the entopallium are then responsive to light after injection, for instance. How were cells chosen for recording is only indicated in the methods (this should be briefly mentioned in the results for clarity) as it includes pre-screening for optically-evoked responses. Therefore, to say that 'In all recorded cells, a significant number of action potentials could be evoked by optical stimulation' is a bit misleading since these were very specifically selected for it would seem. Also this should be addressed in some way, as Figure 4 shows many NeuN+ and Chr2- cells, so implications of low-transfection rates should be discussed in light of behavioural manipulations – as is a caveat in all optogenetic experiments. Do the authors have any data from cells that were not responsive to light, or any indication of how hard it is to find these responsive cells? Also, I did not see a place where it mentioned how many pigeons these cells were from. This should be included.

5. Line 220: I am not sure I understand the leap from Figure 6 results and the preceding two

paragraphs to: 'therefore we decided to use a pulsed optical stimulation protocol of 40 Hz in our behavioral experiments (pulse duration: 15 ms; inter-pulse interval: 10 ms).' Just because 10ms was effective I assume? And what about the inter-pulse interval? Please clarify this.

6. Line 252: Can you give some specifics about 'repeatedly tested' and the sessions that were analyzed. How many sessions over how many days for learning for this experiment, range and variability, length of sessions, etc. I see from the methods that this was a very long training procedure, it would be useful to know the variability across learning for the pigeons and if this correlated with any measure in the experiments.

7. Line 255: How were the 5 sessions chosen for analysis? The first 5 sessions to meet criteria? The last 5 sessions? Were there more than 5 to choose from for each pigeon? If so how was this done? In relation to this, are the results comparable if all sessions reaching criteria are used (i.e. how reliable are the results across all sessions)? If the pigeon is performing at 50% in SC5 in a session, does the light have any additional effect? One may hypothesize that it would not if it was specifically behaviourally related, as in SC1.

8. Related to above: While Figures 8c-f are informative, it will be important to show the results by animal, directly comparing each stimulus class ChR2 stim and no stim conditions – to see the within animal (& within session) effect of the optogenetics individually, rather than just the mean effect of the group, i.e. are the best performers in SC5 no stim also the best performers in SC5 stim. This is not a given and should be demonstrated. Is the effect size (absolute decrease in performance) consistent or variable across sessions/animals. Equally, does this effect size correlate to any property of the injection (e.g. size of injection site or density of ChR2 labelling in entopallium, etc).

9. How many times were stimuli classes presented in a session and were they randomized or did they proceed from SC1 to SC5 or vice versa?

10. It is interesting that there is no effect of stimulus class on reaction time even though they are performing at near chance levels for SC1. The 'deciding' phase is not prolonged even if the task is difficult. Is this consistent with previous literature?

11. There is no comparison between the reaction time of the ChR2 injected group and the control group. The control group seems to have faster reaction times in general. Is this significant? If so, could the authors explain. This may be an important control to indicate that the ChR2 is not toxic or interfering with other elements of processing in the task.

12. Line 307: I think it would be prudent to note that the 20Hz protocol (please also specify stim time and inter-stim interval here) had comparable results only in the SC5 category – although the trend is clear, this stimulation protocol did not seem to be sufficient to elicit significant differences in SC3 and SC4.

13. Line 479: typo in second virus, presumable AAV5.hsyn...

14. Line 482: how were injections made? Hamilton syringe (gauge)? Glass pipette (tip diameter)?

15. Line 484: change 'pigeons' here to 'experiments'; related, line 487: 'The pigeons used in behavioural experiments...'

16. Line 494: states recovery period was 14 days but link to timeline in behaviour is not made until further and virus expression was much longer than 14 days. It may be useful to specify recovery until what here.

17. Line 512: The hSyn promoter virus was eYFP though correct? I assume part of the point of the DAB immune was also to create equal comparisons between these two AAVs (i.e. also different fluorophores)– if so, I would state this explicitly. Similarly though, was this eYFP signal sufficiently bright on its own? Also, Figure 3 (mCherry) looks quite bright - was this sometimes variable then?

18. Line 531/533: In my experience, 'cells wells' are generally referred to as XX well plates (e.g. 24 well plates).

19. Lines 552/554, etc. Do the authors mean AlexaFluor 594?

20. Line 573: it would be useful to know the units of 'value'. Are these in um, pixels, arbitrary, etc.?

21. Line 577: states that analysis for area was performed in 1/10th of the brain sections and then summed, but was the total number of sections kept constant? Was there a specified start and end point or was this determined by the injections size alone? (or atlas sections?) Taking every 10th section could theoretically give you a variable number of total sections to sum. Also, every 10th section seems quite low – this would be analysis only every 400um. Were there any differences in the rostral-caudal extent of labelling from injections that might indicate the injection size itself differed (which would clearly affect the amount of resulting labelling)?

22. Line 610: typo 'end'

Comments related to figures:

Figure 1:

1. D) To get a better idea of the density it would be helpful to know the total area of the entopallium for the measured sections (or simply to report density rather than summed area, which may be variable depending on the rostral-caudal extent of the injection itself).

2. C-d) there seems to be a large variability in number of labelled soma (5-25000 range) – what are the implications for behaviour here? Can the authors speculate on the source of this variability (how is injection volume controlled, etc).

Figure 2:

3. Are e to f & g oriented in the same way? F and g appear to be flipped mediolaterally to e? It is hard to tell if this is also the case for b and c.

4. G) There appears to be no labelling in LSt, but some in GP for hSyn?

Figure 4:

5. It would be helpful to add to b) a graph of NeuN+/ChR2- cells (% of NeuN cells labelled in entopallium – i.e. transfection rate)

Figure 6:

6. With only 9 cells, could the authors provide (at least as supplement) a figure showing responses in all these cells. This would be useful information to compare the evoked response patters across this population. This would also help to determine if dividing the 9 cells into three 'classes' is really appropriate? Are these three classes clear in a sample of only 9 cells, or are responses heterogeneous and on a continuum? (see also comment for Supp Table 1 below)

7. How robust are these responses across stim periods? Some measure of variability should be presented.

8. Reporting the spontaneous firing rate (in the off phase) would be helpful as it seems very low from what can be seen and this would be an important check of the overall health of the recorded/transfected neurons. Is it comparable to previous literature (i.e. ChR2 negative cells in entopallium)? Also, a stronger link could be made here in relation to the choice to use excitatory rather than inhibitory opsins.

9. Additionally, is there any data from these cells for just visually-evoked responses to light (presented to the eye)? To demonstrate normal physiological responses?

10. Similarly, are there controls for these recordings showing that the cells do not respond to 620nm stimulation light?

Figure 7:

11. There seem to be many cells that are C-Fos positive but ChR2 negative – yet virtually no C-Fos expression in (g) (presumably reflecting a very low spontaneous activity rate?). In fact, from the presented image, the double labelling seems to be in the minority. Can the authors comment on this and potentially quantify and present the proportion of double labelling here.

Figure 8:

12. Although the mCherry only AAV control is convincing, it would be ideal to have a 620nm stimulation control as well for a ChR2 injected animal, even in small numbers as proof of principle. If the authors do not have this data, I would make a stronger but directly explicit argument in the text based on the mCherry controls that the stimulation light itself was not affecting the visual perception of the pigeons during the task. IF available, a video of the task during stimulation may indicate how bright the stimulation light itself is during the task etc.

Supp Table 1:

13. It would be useful to see this table expanded with mean firing rates (baseline and stim +/- sem) for all cells, in addition to p values.

14. What about the 20ms and 500ms durations? This data is missing (related, also line 602: why would these two stimulation categories not be analysed?)

Reviewer #5 (Remarks to the Author):

This manuscript offers a proof of principal for the use of optogenetics in pigeons by examining the efficacy of several potential viral vectors. By examining the effectiveness of transfection at the injection site, anterograde and retrograde transport, they conclude that the AAV1-CAG is the construct of choice, whereas others are less effective, no effective, or even neurotoxic. Their conclusions are further supported with electrophysiological recordings from transfected neurons in the entopallium (E). Although their sample size is very (small), they show that all neurons can be effectively activated with pulses of light. Finally, in a very well controlled experiment they show that optogenetic activation of E can impair visual intensity discrimination in a behavioural task. Generally speaking this manuscript is well motivated, thorough, clearly written and well organized. With a few exceptions noted below, the data is subject to rigorous quantification and statistical analysis. This proof of principle is an important contribution as the pigeon is historically an important model for studying learning, memory and visual processing. These fields with benefit from the application of optogenetics. In addition, the demonstration that E is involved in intensity discrimination adds to our knowledge about visual processing in E. However, I believe the manuscripts would benefit from with consideration of and E lesion study that was not cited by the authors.

1) The study not cited is by Nguyen et al. (2004), Journal of Neuroscience 24:4962-4970. In the discussion of the submitted manuscript the authors state that the previous studies of the entopallium is involved in pattern and motion processing and that the optogenetic stimulation results are in line with previous lesion studies of entopallium (lines 395-401). To the best of my knowledge, Nguyen et al. is the only study to show that lesion studies of the entopallium impair motion detection. Moreover, Nguyen et al. showed a clear double dissociation whereby the lesions to the caudal and rostral entopallium resulted, respectively, in motion and spatial vision deficits. (This finding was highly consistent with the known topographic projections from rotundas). This study could clearly inform the results of the present study, as it is likely that the optogenetic stimulation of the rostral entopallium is what impaired intensity discrimination. Moreover, their conclusion that their results lend support to the idea that the avian tectofugal pathway is similar to the mammalian geniculostriate pathway (lines 401-403 and 103-104) must be taken with caution.

The pitfalls of such a conclusion is discussed in detail by Nguyen et al., who note that studies of the mammalian tectofugal pathway show that it is also involved in colour and form vision, but these studies tend to be overlooked.

2) In general there is a concern about how labelled neurons were counted and it was not clear in the methods and associated text. At first I thought it was an exhaustive count of all labelled cells, but at times it said that every tenth section was counted. If there is a lot of labelling, the authors should be following the principals of unbiased stereology, and clearly document these in the methods. Could a section be added to the methods clearly indicating the procedures for counting cells?

3) Was the labelling distributed evenly throughout the rostro-caudal extent of the entopallium in each case? Was the extent of spread outside the entopallium quantified?

4) lines 179-180 – It is noted that there was a severe reduction of NeuN expression with AAV9 within the expression area. How was this quantified? To what areas were the comparison made? Is there any data in this regard?

5) lines 210-215 – It is stated that there were 3 classes of cells. Please indicate how many of the 9 total cells were in each class.

6) lines 227-231 – It is indicated that optogenetic stimulation resulted in c-fos activation only in the blue-light hemisphere. Surely there must have been some basal level in the unstimulated hemisphere (and see fig 7g?). The number of c-fos-positive cells should be quantified in each E, and a alternate non-stimulated site in the telencephalon. I am also a little concerned with the section shown in 7h. The pale colour where the c-fos-positive cells are is concerning. Is this tissue damage? Is this simply c-fos activation due to injury?

7) line 318 – replace “birds” with “pigeons”

8) line 465-466 – dosage of ketamine and xylazine should be given in mg/kg.

9) line 482 - how were the constructs injected? Hamilton syringes? Over what time course?

10) line 610 – replace “end” with “and”

11) line 667 – replace “with” with “by”

General Reply:

We would like to express our gratitude to the editors and the five anonymous reviewers for their thorough and positive evaluation of our manuscript. We believe the reviewers' comments and suggestions have significantly improved the quality and transparency of the manuscript.

The revised version now comprises additional information on the general role of the entopallium in visual processing and especially in the studied visual behavior. In addition, we added methodological information. Further, we have incorporated all additional quantifications and revised the respective text passages and figures.

In this response letter, we have consecutively numbered all remarks. We have addressed all issues point-by-point and highlighted all changes made in the manuscript using red fonts. To ensure a straightforward evaluation we also included the updated figures in the response letter as well as in the revised manuscript. Sometimes, remarks or questions occurred more than once to the same part of the manuscript. In these cases, we answered them extensively in one comment and later refer to this to avoid redundancies.

Reviewers' comments:**Reviewer #1 and #2 (PI/Trainee evaluation; Remarks to the Author):**

While optogenetics is now widely used for neuroscience in genetic models like rodents, flies and zebrafish, it is still rarely used in other organisms where efficient viral transduction is more challenging to achieve. This study establishes viral transfer in the pigeon, a common model animal in avian neuroscience, and provides proof of principle for the use of optogenetics to manipulate cellular and behavioral activity through electrophysiology and a contrast perception task. I expect this work to be influential in the field of avian neuroscience as other labs work to adapt these tools for the manipulation of cells, circuits and behavior in the avian nervous system.

Major concerns:

Overall this is a fairly comprehensive study of the efficiency of viral transduction in that a suite of serotypes and a pair of promoters are methodically analyzed to determine characteristics that might best suit future studies. My concerns are suggestions or questions that would make this strong contribution even stronger by providing additional information.

The first concern is lack of reporting sample sizes and quantification of qualitative data. Specifically, I recommend adding the following:

Comment 1:

State how many birds were used in figure 1 (cell/area counting components).

Answer 1:

Every serotype was injected into at least $n = 5$ hemispheres in at least three animals. Table 1 informed the reader about the number of injections made for each serotype. We have now included the information about the number of pigeons in the results and in the method section.

New sentence in the results:

“Each construct was injected into at least five separate hemispheres of three pigeons (for more information see table 1)”

New sentence in the methods:

“For the viral transfection study 5 µl of each viral vector were pressure injected into at least five separate hemispheres of three pigeons.”

Performing the analysis with the number of injections/hemispheres instead of animals is often done in viral transfection studies to reduce the amount of animals sacrificed.

<https://pubmed.ncbi.nlm.nih.gov/25240284/>

<https://pubmed.ncbi.nlm.nih.gov/26839901/>

<https://pubmed.ncbi.nlm.nih.gov/14502216/>

Comment 2

Provide quantitative information about the level of anterograde and retrograde transport of the channels (Figure 2).

Answer 2:

We thank the reviewer for this helpful suggestion. We have now included supplementary table 1, in which the amount of anterograde and retrograde transgene expression is quantified for all cases. This quantification was performed by measuring intensity values. The examples for all cases can be found in supplementary figures 3-8. We classified the cases into + little, ++ moderate and +++ extensive anterograde and retrograde transgene expression based on the intensity values.

This form of classification is often performed in tracing studies for anterograde and retrograde transport:

<https://pubmed.ncbi.nlm.nih.gov/15221956/>

The new section in the methods reads as follows:

For the quantification of anterograde and retrograde transport, the target area was delineated and intensity values were determined. For anterograde transport cases were classified into + little, ++ moderate and +++ extensive (+ signal was < 6 % darker than the background, ++ signals was 6 – 12 % darker than the background, +++ signal was > 12 % darker than the background). For retrograde transport cases were classified into + little, ++ moderate and +++ extensive (+ signal was < 2 % darker than the background, ++ signals was 2 – 4 % darker than the background, +++ signal was > 4 % darker than the background).

Furthermore, we now refer to supplementary table 1 and supplementary figures 3-8 in the result section about anterograde and retrograde transport.

The new table and figures look as follows:

Supplementary table 1. Classification of all anterograde and retrograde cases for AAV1-CAG and AAV1-hSyn in NI/MVL, striatum and nucleus rotundus. Cases were classified into + little, ++ moderate and +++ extensive based on the intensity of transgene expression (see method section for more details).

Case	Serotype	Time	Anterograde NI/MVL	Anterograde Striatum	Retrograde nRT	Supp. Figure
Case 1	AAV1-CAG	> 2 months	+++	+++	+++	S. Fig. 3-5 a
Case 2	AAV1-CAG	> 2 months	+++	+++	+++	S. Fig. 3-5 b
Case 3	AAV1-CAG	> 2 months	+++	+++	+++	S. Fig. 3-5 e
Case 4	AAV1-CAG	> 2 months	+++	+++	+++	S. Fig. 3-5 f
Case 5	AAV1-CAG	> 2 months	++	+	+++	S. Fig. 3-5 i
Case 6	AAV1-CAG	> 2 months	++	+++	++	S. Fig. 3-5 j
Case 7	AAV1-CAG	> 2 months	+++	+++	+++	S. Fig. 3-5 m
Case 8	AAV1-CAG	> 2 months	+++	+++	+++	S. Fig. 3-5 n
Case 9	AAV1-CAG	6 weeks	++	+	+++	S. Fig. 3-5 c
Case 10	AAV1-CAG	6 weeks	+	+	++	S. Fig. 3-5 d
Case 11	AAV1-CAG	6 weeks	++	+	+	S. Fig. 3-5 g
Case 12	AAV1-CAG	6 weeks	+	++	+	S. Fig. 3-5 h
Case 13	AAV1-CAG	6 weeks	++	++	n.a.	S. Fig. 3-5 k
Case 14	AAV1-CAG	6 weeks	++	+	n.a.	S. Fig. 3-5 l
Case 15	AAV1-hSyn	> 2 months	+	+	+	S. Fig. 6-8 a
Case 16	AAV1-hSyn	> 2 months	+	+	+	S. Fig. 6-8 b
Case 17	AAV1-hSyn	> 2 months	+	+	+	S. Fig. 6-8 e
Case 18	AAV1-hSyn	> 2 months	+	+	+	S. Fig. 6-8 f
Case 19	AAV1-hSyn	> 2 months	+	+	+	S. Fig. 6-8 c
Case 20	AAV1-hSyn	6 weeks	+	+	+	S. Fig. 6-8 d
Case 21	AAV1-hSyn	6 weeks	+	+	++	S. Fig. 6-8 g
Case 22	AAV1-hSyn	6 weeks	+	++	++	S. Fig. 6-8 h
Case 23	AAV1-hSyn	6 weeks	+	+	+	S. Fig. 6-8 j
Case 24	AAV1-hSyn	6 weeks	+	+	+	S. Fig. 6-8 k

2 months

6 weeks

Supplementary Figure 3. Anterograde labeling in NI/MVL following injections of AAV1-CAG-ChR2 into the entopallium. Sections were stained against ChR2 and the signal was visualized with a DAB staining procedure. Cases 1 - 8 had longer expression times (> 2 months) and cases 9-14 were perfused 6 weeks after the injections. All scale bars depict 100 μm .

2 months

6 weeks

Supplementary Figure 4. Anterograde labeling in striatum following injections of AAV1-CAG-ChR2 into the entopallium. Sections were stained against ChR2 and the signal was visualized with a DAB staining procedure. Cases 1 - 8 had longer expression times (> 2 months) and cases 9-14 were perfused 6 weeks after the injections. All scale bars depict 100 μm.

2 months

6 weeks

Supplementary Figure 5. Retrograde labeling in the nucleus rotundus following injections of AAV1-CAG-ChR2 into the entopallium. Sections were stained against ChR2 and the signal was visualized with a DAB staining procedure. Cases 1 - 8 had longer expression times (> 2 months) and cases 9-12 were perfused 6 weeks after the injections. All scale bars depict 100 μ m.

2 months

6 weeks

Supplementary Figure 6. Anterograde labeling in NI/MVL following injections of AAV1-hSyn-ChR2 into the entopallium. Sections were stained against ChR2 and the signal was visualized with a DAB staining procedure. Cases 15 - 18 had longer expression times (> 2 months) and cases 19-24 were perfused 6 weeks after the injections. All scale bars depict 100 μm.

2 months

6 weeks

Supplementary Figure 7. Anterograde labeling in striatum following injections of AAV1-hSyn-ChR2 into the entopallium. Sections were stained against ChR2 and the signal was visualized with a DAB staining procedure. Cases 15 - 18 had longer expression times (> 2 months) and cases 19-24 were perfused 6 weeks after the injections. All scale bars depict 100 μm.

2 months

6 weeks

Supplementary Figure 8. Retrograde labeling in the nucleus rotundus following injections of AAV1-hSyn-ChR2 into the entopallium. Sections were stained against ChR2 and the signal was visualized with a DAB staining procedure. Cases 15 - 18 had longer expression times (> 2 months) and cases 19-24 were perfused 6 weeks after the injections. All scale bars depict 100 μ m.

Comment 3:

State how many birds were included in figure 4 (the tropism experiments).

Comment 4:

Provide a quantification of the GFAP signal in figure 5.

Answer comment 3/4:

We have included the information about the number of cases that were analyzed for the tropism experiments (NeuN stainings) in the figure description and hope that it gets clearer based on the data points that are now plotted into the new figure 4. Furthermore, we have rearranged Figures 4 and 5 based on your suggestion and have merged them into one figure that now also includes a quantification of the GFAP signal for AAV1 and AAV9.

The new result section reads as follows:

“We found that injections of AAV1-CAG-ChR2 led to extensive ChR2 expression in the injection area, while GFAP expression was low and occurred mainly around blood vessels (ChR2 expression: $8.5 \% \pm 1.6 \text{ SEM}$, $n = 3$; GFAP expression: $1.5 \% \pm 0.5 \text{ SEM}$, $n = 3$; Fig. 4 i-l, Supplementary Fig. 9). In contrast to this, injections of AAV9-CAG led to weak ChR2 expression, while GFAP expression was strong and occurred throughout the injection site (ChR2 expression: $1.5 \% \pm 0.6 \text{ SEM}$, $n = 3$; GFAP expression: $3.7 \% \pm 0.9 \text{ SEM}$, $n = 3$; Fig. 4 m-p, Supplementary Fig. 9).”

The new figure 4 looks as follows:

Figure 4. Cellular tropism of AAV1-CAG and AAV9-CAG. (a-d) AAV1-CAG-ChR2 leads to ChR2 expression in significantly more neurons (pink bar) than glial cells (green bar, n = 5 injections). **(b)** ChR2 expression after injections of AAV1-CAG. **(c)** NeuN expression in the corresponding injection site. **(d)** Overlay of ChR2 and NeuN expression. **(e-h)** AAV9-CAG (green bar, n = 4 injections) leads to a significant reduction of NeuN in the injection site compared to AAV1-CAG (pink bar, n = 5 injections). **(f)** ChR2 expression following injections of AAV9-CAG. **(g)** NeuN in the corresponding injection site. **(h)** Overlay of ChR2 and NeuN expression indicating that AAV9-CAG injections result in reduced NeuN expression in the injection site. **(i-l)** AAV1-CAG leads to extensive ChR2 expression (pink bar), but only to weak GFAP expression (green bar, n = 3 injections). **(j)** ChR2 expression following injections of AAV1-CAG-ChR2. **(k)** GFAP expression occurs mainly around blood vessels **(l)** Overlay of ChR2 and GFAP expression. **(m)** AAV9-CAG injections lead to weak ChR2 expression (pink bar), but increased GFAP expression (green bar, n = 3 injections) in the injection site **(n)** ChR2 expression following injections of AAV9-CAG-ChR2. **(o)** GFAP expression within the corresponding injection site. **(p)** Overlay of ChR2 and GFAP expression. All scale bars are specified within the microscopic images. Error bars represent the standard error of the mean (SEM), dots represent the raw data. **p < .01, *p < .05.

Comment 5

Provide a population analysis for the nine cells in figure 6 (electrophysiology) if possible. Were these all from the same animal?

Answer 5

The recorded cells originate from two animals (four hemispheres). They were recorded around the coordinates 9.5 ± 0.5 AP (see also comment 10). Reviewer 4 in comment 67 also raised the question about a population analysis to clarify if it is justified to report three distinct classes of cells. To compare the response profiles of all recorded cells, we included normalized responses to the light stimulation for the respective stimulation durations. We agree with both reviewers, that a statement of cell classes is not warranted based on the small amount of recorded cells. The experiments were geared to show the overall functioning of the channels and not to assess the physiological properties of single cells. Besides the normalized responses, in line with Reviewer 4 we decided to include raster plots of all recorded cells (see Supplementary Fig. 10 – 18, see comment 67) to illustrate the individual responses. We toned down our statement on distinct cell classes and describe the individual cell responses instead. We complement the description with a detailed table of characteristic values to clarify the individual responses (Supplementary table 2).

The new result section reads as follows:

The physiology of ChR2 was assessed in two experiments. In the first experiment, pigeons were anesthetized, and extracellular single unit recordings and optical stimulation were performed simultaneously within the entopallium of two pigeons in 4 hemispheres. The goal of these specific experiments was to preselect cells that were responsive to light and assess their characteristics. Therefore, a constantly repeated light pulse of 1 s duration was presented to evoke spikes during the advancement of the electrode. When a responsive cell was encountered, we used a sequence of blue light pulses (465 nm) of different durations (1 ms, 10 ms, 100 ms, 200 ms and 500 ms; see supplementary table 3) for optical stimulation and repeated this sequence (sweeps, Fig. 5 a1) three times. In total, neuronal responses of nine cells were recorded during optical stimulation. In all preselected cells that were recorded, a significant number of action potentials could be evoked by optical stimulation and all cells showed a significant response when the duration of the stimulation was a least 10 ms (one-sided Wilcoxon rank sum test, all p-values < 0.05, for details see Supplementary Table 2). To assess the variability of the evoked neuronal responses, we compared the spikes evoked during these sweeps using a non-parametric analysis of variance (Kruskal-Wallis-Test). If we found

significant differences between the sweeps a Bonferroni-corrected multiple comparison test was conducted. In only two out of the nine cells, significant differences between sweeps in some conditions were detected (see supplementary table 2). The differences were found in stimulation trials of longer duration (>100 ms). For stimulation durations below 100 ms no significant differences could be detected. The significance is indicated in supplementary table 2 and in the single cell raster plots. Overall, the evoked responses were robust.

The recorded cells differed in their overall response properties. We found cells that responded throughout the entire optical stimulation period with a constant amount of spikes, albeit showing a pronounced peak of activation at the onset of light stimulation (Fig. 5, Supplementary Fig. 15, 17; Supplementary Fig. 18 cells 4, 7 and 9). Further we found cells that weakened their responses over the course of prolonged stimulation (Supplementary Fig. 10, 11 and 12; Supplementary Fig. 18 cells 1, 2 and 3). Another response pattern found, showed a sharp peak only during the onset of the stimulus (Supplementary Fig. 13, 14 and 16, Supplementary Fig. 18 cells 5, 6 and 8). After the electrophysiological experiments were finished, histology was performed to check for ChR2 expression in the entopallium (Supplementary Fig. 19)

The new main text figure looks as follows:

Figure 5. Single cell responses upon optical stimulation (Cell 4). (a1) Stimulation protocol (upper trace) and the resulting evoked cellular responses (lower trace). The stimulation protocol consisted of repeated light pulses of different durations (1 ms, 10 ms, 100 ms, 200 ms and 500 ms) and was repeated three times (sweep 1-3, indicated by the gray/white background). The baseline-firing rate of each neuron was assessed before/after the light stimulation protocol. (a2) A single sweep (i.e. sweep 2) of the optical stimulation protocol. The 10 ms optical stimulation is highlighted and further analyzed in (a3). Raster plot (upper part) and Peri-stimulus time histogram (PSTH; lower part) of the cell response. The raster plot characterizes the cellular response aligned with the onset of the optical stimulation (465 nm @ 300 mA; blue shaded area). Each line represents one optical stimulation. Each dot within that line represents an evoked action potential. For each repetition of the stimulation, a new line is added to the plot. The gray/white background of the raster plot indicates the blocks of repetition (sweep 1-3). To assess the variability of the evoked responses, the sweeps were statistically compared. In the case of this cell, no statistical differences were found (indicated by the abbreviation n.s. next to the raster plots). The PSTH represents the summed responses within a certain time window (bin). In case of the 10 ms stimulus presentation the bin width is 2 ms. (b) Raster plot (upper trace) and PSTH (lower trace) for the 1 ms stimulus duration (bin width: 2 ms). (c) Raster plot (upper trace) and PSTH (lower trace) for the 100 ms stimulus duration (bin width: 4 ms). (d) Raster plot (upper trace) and PSTH (lower trace) for the 200 ms stimulus duration (bin width: 10 ms). (e) Raster plot (upper trace) and PSTH (lower trace) for the 500 ms stimulus duration (bin width: 10 ms).

The new supplementary Fig. look as follows:

Supplementary figure 10: Single cell responses upon optical stimulation (Cell 3). **(a1)** Stimulation protocol (upper trace) and the resulting evoked cellular responses (lower trace). The stimulation protocol consisted of repeated light pulses of different durations (1 ms, 10 ms, 100 ms, 200 ms and 500 ms) and was repeated three times (sweep 1-3, indicated by the gray/white background). The baseline-firing rate of each neuron was assessed before/after the light stimulation protocol. **(a2)** A single sweep (i.e. sweep 2) of the optical stimulation protocol. The 10 ms optical stimulation is highlighted and further analyzed in **(a3)**. Raster plot (upper part) and Peri-stimulus time histogram (PSTH; lower part) of the cell response. The raster plot characterizes the cellular response aligned with the onset of the optical stimulation (465 nm @ 300 mA; blue shaded area). Each line represents one optical stimulation. Each dot within that line represents an evoked action potential. For each repetition of the stimulation, a new line is added to the plot. The gray/white background of the raster plot indicates the blocks of repetition (sweep 1-3). To assess the variability of the evoked responses, the sweeps were statistically compared. In the case of this cell, no statistical differences were found (indicated by the abbreviation n.s. next to the raster plots). The PSTH represents the summed responses within a certain time window (bin). In case of the 10 ms stimulus presentation the bin width is 2 ms. **(b)** Raster plot (upper trace) and PSTH (lower trace) for the 1 ms stimulus duration (bin width: 2 ms). **(c)** Raster plot (upper trace) and PSTH (lower trace) for the 100 ms stimulus duration (bin width: 4 ms). **(d)** Raster plot (upper trace) and PSTH (lower trace) for the 200 ms stimulus duration (bin width: 10 ms). **(e)** Raster plot (upper trace) and PSTH (lower trace) for the 500 ms stimulus duration (bin width: 10 ms).

Supplementary figure 11: Single cell responses upon optical stimulation (Cell 1). (a) Raster plot (upper trace) and PSTH (lower trace) for the 1 ms stimulus duration (bin width: 2 ms). (b) Raster plot (upper trace) and PSTH (lower trace) for the 10 ms stimulus duration (bin width: 2 ms). (c) Raster plot (upper trace) and PSTH (lower trace) for the 100 ms stimulus duration (bin width: 4 ms). (d) Raster plot (upper trace) and PSTH (lower trace) for the 200 ms stimulus duration (bin width: 10 ms). (e) Raster plot (upper trace) and PSTH (lower trace) for the 500 ms stimulus duration (bin width: 10 ms).

Supplementary figure 12: Single cell responses upon optical stimulation (Cell 2). (a) Raster plot (upper trace) and PSTH (lower trace) for the 1 ms stimulus duration (bin width: 2 ms). (b) Raster plot (upper trace) and PSTH (lower trace) for the 10 ms stimulus duration (bin width: 2 ms). (c) Raster plot (upper trace) and PSTH (lower trace) for the 100 ms stimulus duration (bin width: 4 ms). (d) Raster plot (upper trace) and PSTH (lower trace) for the 200 ms stimulus duration (bin width: 10 ms). (e) Raster plot (upper trace) and PSTH (lower trace) for the 500 ms stimulus duration (bin width: 10 ms).

Supplementary figure 13: Single cell responses upon optical stimulation (Cell 5). (a) Raster plot (upper trace) and PSTH (lower trace) for the 1 ms stimulus duration (bin width: 2 ms). (b) Raster plot (upper trace) and PSTH (lower trace) for the 10 ms stimulus duration (bin width: 2 ms). (c) Raster plot (upper trace) and PSTH (lower trace) for the 100 ms stimulus duration (bin width: 4 ms). (d) Raster plot (upper trace) and PSTH (lower trace) for the 200 ms stimulus duration (bin width: 10 ms). (e) Raster plot (upper trace) and PSTH (lower trace) for the 500 ms stimulus duration (bin width: 10 ms).

Supplementary figure 14: Single cell responses upon optical stimulation (Cell 6). (a) Raster plot (upper trace) and PSTH (lower trace) for the 1 ms stimulus duration (bin width: 2 ms). (b) Raster plot (upper trace) and PSTH (lower trace) for the 10 ms stimulus duration (bin width: 2 ms). (c) Raster plot (upper trace) and PSTH (lower trace) for the 100 ms stimulus duration (bin width: 4 ms). (d) Raster plot (upper trace) and PSTH (lower trace) for the 200 ms stimulus duration (bin width: 10 ms). (e) Raster plot (upper trace) and PSTH (lower trace) for the 500 ms stimulus duration (bin width: 10 ms).

Supplementary figure 15: Single cell responses upon optical stimulation (Cell 7). (a) Raster plot (upper trace) and PSTH (lower trace) for the 1 ms stimulus duration (bin width: 2 ms). (b) Raster plot (upper trace) and PSTH (lower trace) for the 10 ms stimulus duration (bin width: 2 ms). (c) Raster plot (upper trace) and PSTH (lower trace) for the 100 ms stimulus duration (bin width: 4 ms). (d) Raster plot (upper trace) and PSTH (lower trace) for the 200 ms stimulus duration (bin width: 10 ms). (e) Raster plot (upper trace) and PSTH (lower trace) for the 500 ms stimulus duration (bin width: 10 ms).

Supplementary figure 16: Single cell responses upon optical stimulation (Cell 8). (a) Raster plot (upper trace) and PSTH (lower trace) for the 1 ms stimulus duration (bin width: 2 ms). (b) Raster plot (upper trace) and PSTH (lower trace) for the 10 ms stimulus duration (bin width: 2 ms). (c) Raster plot (upper trace) and PSTH (lower trace) for the 100 ms stimulus duration (bin width: 4 ms). (d) Raster plot (upper trace) and PSTH (lower trace) for the 200 ms stimulus duration (bin width: 10 ms). (e) Raster plot (upper trace) and PSTH (lower trace) for the 500 ms stimulus duration (bin width: 10 ms).

Supplementary figure 17: Single cell responses upon optical stimulation (Cell 9). (a) Raster plot (upper trace) and PSTH (lower trace) for the 1 ms stimulus duration (bin width: 2 ms). (b) Raster plot (upper trace) and PSTH (lower trace) for the 10 ms stimulus duration (bin width: 2 ms). (c) Raster plot (upper trace) and PSTH (lower trace) for the 100 ms stimulus duration (bin width: 4 ms). Please note the difference in the firing rates between the first and third sweep. The firing rates of the individual sweeps are given in Supplementary table 2. (d) Raster plot (upper trace) and PSTH (lower trace) for the 200 ms stimulus duration (bin width: 10 ms). Please note the difference in the firing rates between the first and third sweep. The firing rates of the individual sweeps are given in Supplementary table 2. (e) Raster plot (upper trace) and PSTH (lower trace) for the 500 ms stimulus duration (bin width: 10 ms). Please note the difference in the firing rates between the first and third sweep as well as the second and third sweep. The firing rates of the individual sweeps are given in Supplementary table 2.

Supplementary figure 18: Absolute and relative evoked firing rates under light stimulation. (a) Average evoked firing rates during different stimulation durations. Please note that for the 1 ms and the 10 ms stimulation duration a 20 ms time window after stimulation onset was analyzed. In the table below the figure, values for each individual cell are given. **(b)** Normalized evoked firing rates.

The new supplementary table looks as follows:

Supplementary table 2: Summary of cell responses upon optical stimulation. For each cell, the spontaneous activity is given in the first block of the table. Following blocks summarize the cellular responses upon visual stimulation for each stimulus duration. Each sweep of stimulation is given separately. The labeling of the sweeps (gray/white background) corresponds to the figures in the single cell depiction. As a measure of variability, firing rates per bin of each individual sweep were statistically compared. If there were significant differences in the firing rate per bin between consecutive sweeps, values are printed in bold and marked with an asterisk. The last row of each block indicates if the optical stimulation resulted in a significant cell response compared to the non-stimulated periods within the time window of the respective Raster/PSTH time window. Please note that for the 1 ms and 10 ms stimulation duration a 20 ms time window after stimulation onset was analyzed.

	Cell 1	Cell 2	Cell 3	Cell 4	Cell 5	Cell 6	Cell 7	Cell 8	Cell 9
Spontaneous activity	2.00	3.82	5.07	0	0	1.50	1.90	1.36	3.88
Standard deviation	1.79	2.14	4.20	0	0	1.51	1.60	2.27	3.22
1 ms stim. duration									
Mean AP/bin sweep 1	0.60	0.20	0.40	1.80	2.40	0.40	0	0	1.10
Standard deviation	0.32	0.42	0.70	4.37	3.72	0.84	0	0	2.85
Mean AP/bin sweep 2	0.40	0.10	0.40	1.50	2.30	0.10	0.10	0	1.30
Standard deviation	0.70	0.32	0.70	4.40	3.37	0.32	0.32	0	3.77
Mean AP/bin sweep 3	0.10	0	0.30	1.60	3.10	0.10	0.20	0	1.20
Standard deviation	0.32	0	0.95	5.06	3.51	0.32	0.42	0	2.78
Mean AP/bin all sweeps	1	0.3	1.1	4.9	7.8	0.6	0.3	0	3.6
Standard deviation	1.56	0.67	1.73	13.78	9.83	1.08	0.67	0	9.34
Evoked firing rate [Hz]	9.17	2.5	9.17	40.83	130	5	5	0	30
p-value	0.14	0.66	0.38	0.04	0.003	0.55	0.54	-----	0.35
10 ms stim. duration									
Mean AP/bin sweep 1	2.90	1.40	1.60	4.10	4.00	3.90	1.20	2.10	5.60
Standard deviation	3.00	1.96	2.32	6.92	4.40	4.95	1.32	3.73	7.49
Mean AP/bin sweep 2	3.90	1.20	1.10	3.50	4.00	4.40	1.40	1.90	5.40
Standard deviation	4.36	1.87	1.52	5.50	4.35	5.36	1.78	3.57	7.03
Mean AP/bin sweep 3	2.20	1.20	1.50	3.90	3.80	4.60	1.20	1.90	5.50
Standard deviation	2.70	1.87	1.90	5.72	4.73	6.24	1.69	3.18	7.89
Mean AP/bin all sweeps	9	3.8	4.2	11.5	11.8	12.9	3.8	5.9	16.5
Standard deviation	8.22	4.59	5.03	17.4	12.98	14.96	4.26	10.02	22.31
Evoked firing rate [Hz]	75	31.67	35	95.83	196.67	107.5	63.33	98.33	137.5
p-value	< 0.001	0.01	0.02	0.003	0.007	< 0.001	0.005	0.02	0.003
100 ms stim. duration									
Mean AP/bin sweep 1	16.48	2.44	2.56	8.48	3.68	4.72	1.68	1.08	11.48 *
Standard deviation	2.82	1.53	1.56	3.57	4.53	6.33	1.46	2.41	3.94
Mean AP/bin sweep 2	17.08	2.48	3.04	7.72	4.60	5.05	1.40	0.68	9.00
Standard deviation	3.00	1.69	1.62	2.97	4.32	6.60	1.38	2.04	4.44
Mean AP/bin sweep 3	16.84	2.44	2.88	7.64	4.80	4.44	1.60	0.56	8.52 *
Standard deviation	3.94	1.56	2.05	3.47	4.80	6.16	1.47	2.06	5.16
Mean AP/bin all sweeps	50.4	7.36	8.48	23.84	13.08	14.2	4.68	2.28	29
Standard deviation	6.68	3.34	3.47	8.47	12.81	18.57	3.64	5.83	12.21
Evoked firing rate [Hz]	210	30.67	35.33	99.33	109	59.17	39	19.33	120.83
p-value	< 0.001	< 0.001	< 0.001	< 0.001	< 0.001	< 0.001	< 0.001	< 0.007	< 0.001
200 ms stim. duration									
Mean AP/bin sweep 1	23.35	2.50	2.85	10.40	3.10	3.50	1.90	0.60	10.10 *
Standard deviation	3.07	1.54	1.81	1.96	4.99	6.19	1.59	2.09	5.47
Mean AP/bin sweep 2	23.25	3.00	2.95	9.85	2.75	2.80	1.25 *	0.55	10.00
Standard deviation	4.06	1.97	1.39	2.25	4.64	5.47	1.65	2.01	6.15
Mean AP/bin sweep 3	23.15	2.60	2.80	9.20	3.60	2.70	2.35 *	0.35	13.25 *
Standard deviation	3.34	1.05	1.74	2.46	4.98	5.29	1.23	1.35	3.80
Mean AP/bin all sweeps	69.75	8.1	8.6	29.5	9.45	9	5.5	1.5	33.35
Standard deviation	8.75	3.19	3.78	6.02	14.18	16.64	3.83	5.41	14.34
Evoked firing rate [Hz]	232.5	27	28.67	98.17	63	30	36.67	10	111.17
p-value	< 0.001	< 0.001	< 0.001	< 0.001	< 0.001	0.046	< 0.001	0.94	< 0.001
500 ms stim. duration									
Mean AP/bin sweep 1	10.04	0.82	1.14	5.74	0.78	0.64	1.54	0.26	6.25 *
Standard deviation	3.75	0.83	1.05	1.07	2.13	1.89	0.91	0.96	2.00
Mean AP/bin sweep 2	9.16	0.98	1.08	5.40	1.10	0.66	1.26 *	0.24	6.65 *
Standard deviation	4.06	1.00	0.78	1.39	2.26	1.80	0.85	0.92	1.91
Mean AP/bin sweep 3	8.98	1.02	1.18	5.28	1.00	0.44	1.90 *	0.18	7.75 *
Standard deviation	3.88	0.98	0.90	1.21	2.20	1.43	1.11	0.75	2.21
Mean AP/bin all sweeps	28.18	2.82	3.4	16.42	2.88	1.74	4.7	0.68	21.04
Standard deviation	10.88	1.84	1.8	2.67	6.43	4.97	1.97	2.54	4.22
Evoked firing rate [Hz]	187.86	18.8	22.67	109.47	32	11.6	52.22	7.55	140.4
p-value	< 0.001	< 0.001	< 0.001	< 0.001	< 0.001	0.01	< 0.001	0.005	< 0.001

Comment 6

State how many birds were used in figure 7 (IEGE).

Answer 6

We investigated the c-Fos expression in three animals. However, as we used dual fiber optic cannulas (see picture attached), with a pitch of 0.7 mm, we quantified the IEG expression at two stimulation sites per hemisphere (two orange stimulated, two blue stimulated). For one animal one cannula tip was not detectable leading to n = 6 blue stimulated and n = 5 orange stimulated sites.

<http://doriclenses.com/life-sciences/fiber-optic-cannulas/825-dual-fiber-optic-cannulas.html>

Comment 7

Also for figure 7, quantify the change in C-Fos levels between hemispheres.

Answer 7

We have included a quantification of the c-Fos data, which is now displayed in figure 6.

The new result section reads as follows:

The cellular activation was assessed at two stimulation sites in each hemisphere of three pigeons within the stimulated entopallium and in a control area within the unstimulated visual wulst. Blue light stimulation resulted in increased c-Fos expression in ChR2 expression cells within the entopallium (Fig 6 c, f, i) compared to orange light stimulation (blue light stimulation: 514 cells \pm 59 SEM, n = 6; orange light stimulation: 12 cells \pm 3 SEM, n = 5; Z = 2.739, p = .004, Fig. 6 b, g, h), although both hemispheres showed reliable ChR2 expression (Fig. 6 d, e, f). In the control area within the visual wulst, there was no difference between red and blue light stimulation, indicating that staining intensities were similar between the two hemispheres (blue light stimulation: 464 cells \pm 107 SEM, n = 6; orange light stimulation: 585 cells \pm 133 SEM, n = 5, Z = -.548, p = .662, Fig. 6 b).

The new figure looks as follows:

Figure 6. Immediate early gene expression after stimulation of ChR2 with blue light (460 nm). (a) Schematic drawing illustrating the experimental procedure. In three animals, two sites of one hemisphere were stimulated with orange light (620 nm), while two sites in the other hemisphere were stimulated with blue light (460 nm). (b) Quantification of c-Fos expression following orange and blue light stimulation in the stimulated entopallium and a non-stimulated control area in the visual wulst. Significantly more c-Fos expression was observed in the blue light stimulated hemisphere (n = 6 sites, blue) compared to the orange stimulated hemisphere (n = 5 sites, orange). However, in the non-stimulated visual wulst c-Fos activation was comparable. (c) Overlay of ChR2 expression with c-Fos expression (fluorescence) indicating that blue light stimulation resulted in cellular activity in ChR2 expressing cells. (d) ChR2 expression in the orange stimulated hemisphere. (e, f) ChR2 expression in the blue light stimulated hemisphere. (g) Little c-Fos expression occurred in the orange stimulated hemisphere. (h, i) Extensive c-Fos expression occurred in the blue light stimulated hemisphere. Scale bars are specified within the images. Error bars represent the standard error of the mean (SEM), dots represent the raw data. **p < .01

Comment 8

Provide the sample size for the individual birds in the contrast perception task (Figure 8).

Answer 8

The contrast perception task was performed in n = 6 experimental pigeons and n = 6 control pigeons. In our old manuscript version this information was provided in the table description of table 2 but might have been easily overlooked. We have changed the old figure 8 (now figure 7) to better illustrate the number of animals used. The new figure can be found in comment 34.

Comment 9

The second concern pertains to the nature of the data in figure 6. It is very compelling to see that entopallial cells were modulated in response to light. However, it is not explained if these cells are visually modulated in the absence of the optrode stimulus. If these cells were first determined to be visually modulated and then demonstrated to also be optogenetically modulated, then evidence for both types of modulation should be shown. If the assessment of visual modulation was performed but cannot now be quantified, then that process should at least be described. Adding evidence that these are visually mediated cells would explicitly connect the visual response properties of the cells and the optrode response to the behavioral outputs you test later.

Answer 9

The concern on the visual responsiveness of entopallial cells was also raised by reviewer 4 comment 70. Unfortunately, we cannot provide the requested data due to the methods we used to record. All cells were recorded under full anesthesia while the animals were fixed in a stereotactic apparatus. To induce anesthesia, we injected a mixture of ketamine/xylazine and maintained anesthesia using a laminar flow of an oxygen/isoflurane mixture. Under these conditions, animals have closed eyelids and it is difficult to include a valid visual stimulation. Another aspect that refrained us to perform these visual stimulation experiments is the effect of isoflurane anesthesia on the neural responses in the visual system. Isoflurane severely depresses cellular responses (Jehle et al. 2009). Given the methodical difficulties and to avoid any flawed cellular responses to the light stimulation, we decided to solely investigate the responsiveness of these cells upon optogenetic stimulation.

The contribution of the entopallium in visual processing is well documented on the electrophysiological level (e.g. Gu et al. (2002)) and was recently reviewed in Clark and Colombo (2020).

Clark, W. J., & Colombo, M. (2020), The functional architecture, receptive field characteristics, and representation of objects in the visual network of the pigeon brain, *Progress in Neurobiology*.

Gu, Y., Wang, Y., Zhang, T. et al. (2002) Stimulus size selectivity and receptive field organization of ectostriatal neurons in the pigeon. *J Comp Physiol A*, 188, 173–178.

Jehle T, Ehlken D, Wingert K, Feuerstein TJ, Bach M, Lagrèze WA. (2009), Influence of narcotics on luminance and frequency modulated visual evoked potentials in rats. *Doc Ophthalmol.*, 118, 217-224.

Comment 10

The third concern is that the entopallium is expected to have functional and anatomical topography, at least to my understanding. Given this, it was surprising that reporting of locations for recording, stimulating, and tracing was not more explicit. It would be helpful to know locations for the following elements of the manuscript: 1) Provide information about recording sites used to generate the data in figure 6. Ideally, this figure would include a panel for a map of the entopallium with all of the recording sites. A supplementary figure with the histology would also be useful. Together, these images would allow the reader to see if the three populations of cells are in different regions. I recognize that the data might not be available for all of this wish list, but presumably some version of coordinate data can be included.

Answer 10:

For the electrophysiological experiments, the whole entopallium was transfected with AAV1-CAG-CHR2 and the recording sites were located around $A 9.5 \pm 0.50$. We have included the microscopic images of the four recorded hemispheres from the electrophysiological experiments, which can be found in supplementary figure 19. As these pigeons were not chronically implanted, no clear cannula tracts are reconstructable. However, Chr2 was robustly expressed in the whole entopallium of all four recorded hemispheres. We did not intend to reconstruct the exact recording site, as we were interested in a proof of principle (light activates Chr2) instead of a physiological response profile of anterior vs. posterior entopallial cells (see previous comment).

We thus clarified the electrophysiological procedure in the manuscript as described in comment 5 and 9.

The new figure looks as follows:

Supplementary Figure 19. Chr2 expression in the entopallium of the four recorded hemispheres. All scale bars represent 1000 μ m. The entopallium is highlighted.

Comment 11

Similarly, it would be helpful to know locations for the canula implant sites in figure 8. After the behavioral tests, was histology performed to see exactly where the canula were implanted within the entopallium? Again, if histology images are available, these should be included in supplementary figures, ideally for each bird. Adding this information would allow readers to consider your data in the context of reported anatomical and functional topography of the region. If this is possible and proves interesting then it is also recommended that these associations are covered in the discussion.

Answer 11:

All behavioral animals underwent histology following the behavioral experiments. We validated whether ChR2 was successfully expressed within the entopallium and additionally assessed the cannula location sites. We have now included a supplementary figure 24 that outlines the position of the cannula tips for all experimental and control pigeons (color coding matches the colors in figure 7). We have added an example slice showing the position of a canula tip within the entopallium.

As is known from the literature, the entopallium is functionally segregated into at least an anterior/color and a posterior/motion processing area. Thus, we targeted the anterior entopallium for our study. As our findings are congruent to this previous knowledge and we did not explicitly test anterior vs. posterior entopallium, we did not discuss this in depth in the manuscript. However, in the revised manuscript we have now clarified that we targeted the anterior entopallium and that our findings are in line with previous functional data.

The new result section reads as follows:

At the end of experimental testing, for both the experimental and control group, coordinates of the cannula tips were assessed and these were mainly located in the medial to anterior entopallium (see Supplementary Fig. 24).

The new supplementary figure looks as follows:

Supplementary Figure 24. Schematic illustration of the cannula tips within the entopallium. Dots represent the cannula tips of the experimental group, squares represent cannula tips of the control group.

The new discussion part reads as follows:

Moreover, future optogenetic studies could investigate the effect of optogenetic stimulation on other entopallial functions such as motion, pattern, and colour discrimination. In these tasks, a functional segregation has been described as lesions within the anterior entopallium affect pattern, colour, and form discrimination, whereas lesions of the posterior entopallium impair motion processing^{18,21}. Based on these findings and comparable topographic projection patterns within the avian and mammalian tectofugal pathways, it has been proposed that the motion sensitive posterior entopallium is comparable to area MT, whereas the colour/form/pattern sensitive anterior entopallium is comparable to V2, V3 and IT in mammals²¹. As brightness sensitive neurons of the nucleus rotundus primarily project to the anterior entopallium, brightness processing has also been linked to this subdivision²¹. Although not showing a functional segregation, our study confirms that the anterior entopallium is involved in brightness perception as our neuronal manipulation was primarily focussed on this region. The finding that this area governs brightness perception furthermore suggests a functional similarity of the entopallium to mammalian V1⁶⁵.

Minor concerns:

Comment 12

Viral transduction: “morphological abnormalities” maybe worth mentioning possible neurotoxicity right away?

Answer 12

We now mention the possible toxic effect right away.

Comment 13

Could you provide additional justification for your technique of differentiating neurons from glia?

Answer 13

The NeuN staining was performed to determine the extent to which ChR2 expressing cells are also co-localized with NeuN. We performed this staining to be able to estimate the number of transfected neurons in comparison to other cells (which would not be relevant for our task). Our goal was however not to classify the other cells (non-neuronal cells).

However, what we observed in our NeuN stainings is that AAV9 injections resulted in a severe reduction of NeuN expression compared to AAV1 injections. This finding is now also reported quantitatively in figure 4 e and reaches significance.

The subsequent GFAP stainings were then performed, in order to see whether the reduction in NeuN for AAV9 was accompanied by an increase in GFAP signal around the injection site (which would indicate an inflammatory response). This finding has now also been quantified in the revised manuscript and can be seen in figure 4 i, m. The GFAP signal is larger for AAV9 than for AAV1.

Our GFAP staining was not performed to determine the overlap of the virus with GFAP (to classify the transduced cells as astrocytes), but rather to visualize that the neuronal loss observed for AAV9 is

associated with an inflammatory response within the injection site. We recognize that this was poorly worded in the initial manuscript and have rephrased this section accordingly.

We updated Figure 4 and 5 (see comment 3/4), included quantifications and also clarified the written result section which now reads as follows:

Since the CAG promoter can drive transgene expression in all cell types, the extent to which ChR2 expressing cells were also co-localized with a neuronal marker was investigated. Therefore, we performed a combined immunohistochemical staining against ChR2 and NeuN to visualize neurons (Fig. 4 a-d). We found that AAV1-CAG-ChR2 led to significantly more transgene expression in neurons than in glial cells (neurons: $92.19\% \pm 1.99$ SEM, $n = 5$; glial cells: $7.81\% \pm 1.99$ SEM, $n = 5$, $Z = -2.023$, $p = .043$, Fig. 4 a–d). This quantification could not be performed for AAV9-CAG-ChR2 injections, as this serotype resulted in a severe reduction of NeuN expression within the ChR2 expressing area (AAV1-CAG: 1871 NeuN+ cells per $\text{mm}^2 \pm 41$ SEM, $n = 6$; AAV9-CAG: 1102 NeuN+ cells per $\text{mm}^2 \pm 77$ SEM, $n = 4$; $Z = -2.558$, $p = .01$, Fig. 4 e-h), suggesting neurotoxicity of this serotype.

Comment 14

Why didn't you test an AAV2 serotype as well, as this is commonly used in the zebra finch song system?

Answer 14:

Here a misunderstanding may have occurred that we have now tried to clarify. The viral vectors we used are pseudotyped AAVs that contain the ITR sites of AAV2. This means that the genome of one serotype (in this case AAV2) was inserted into the capsid of a different serotype (in our case AAV1, AAV5 and AAV9). This is performed as viral capsids differ in their tropism and transduction efficiencies.

For more information on pseudotyping:

<https://pubmed.ncbi.nlm.nih.gov/15294177/>

Sometimes these constructs are then referred to as rAAV2/1, rAAV2/5 or rAAV2/9, but at the moment the convention is rather to simply refer to them as AAV1, AAV5 or AAV9 (as this is how they are commercially advertised by Addgene)

<https://www.addgene.org/100054/> (here you can also find information about the plasmids)

<https://www.addgene.org/26973/> (here you can also find information about the plasmids)

To make this clearer we have now added a sentence at the beginning of the results stating:

“For viral transfection studies we compared the efficiency of AAV serotype 2 pseudotyped with serotype 1 (here referred to as AAV1), pseudotyped with serotype 5 (here referred to as AAV5) and pseudotyped with AAV9 (here referred to as AAV9).”

Thus, the viral vectors that we have tested in our study are partly comparable to those that have been used in zebra finches so far. For example, Roberts et al. (2012) used scAAV2/9-CMV-hChR2–yellow fluorescent protein (YFP), while the paper from Xiao et al. (2018) also used AAV1-CAG among others.

<https://pubmed.ncbi.nlm.nih.gov/22983208/>

<https://pubmed.ncbi.nlm.nih.gov/29551492/>

Comment 15

Electrophysiology: It might be nice to report what type of recording you're doing in the manuscript as well as the optical power range. Would like to know the Hz of the LED light pulses, not just duration, since this could impact the biophysics of how channels open and close in the cells—unless it was always just a single pulse? Did you generate a dose-response curve of changes in optical power levels or just pick one level?

Answer 15

Reviewer 4 also raised questions about the nature of the recordings we used in our experiments. We added a section in the main text that clarifies our procedures as well as the pre-selection of cells that was only mentioned in the methods section (see comment 5).

Further, the requested values have been included in the supplementary table 3 as well in the new figure that highlights our optogenetic stimulation approach (Fig 5 a, supplementary table 3). In figure 5 a an overview of the employed stimulation is given. We used repeated pulses of different duration and intervals and repeated this stimulation sweep three times. In most cases the full stimulation protocol was recorded. However, in some cases only a short protocol could be recorded due to signal loss (see supplementary table 3).

We only used the highest power range (300 mW) and did not perform a dose-response curve. Electrophysiological experiments have shown an almost linear relationship for the light stimulation intensity. The higher the light intensity, the higher the evoked membrane current amplitudes (Erofeev et al. 2019). Thus, we used the highest available light intensity, because we were interested in the overall responsiveness of the transfected cells and we were not aiming to investigate individual cell characteristics and/or light stimulation parameters. The light intensity at the cannula tip was in a range between 3.5 – 4.5 mW.

Erofeev, A., Gerasimov, E., Lavrova, A., Bolshakova, A., Postnikov, E., Bezprozvanny, I., Vlasova, O.L. (2019) Light Stimulation Parameters Determine Neuron Dynamic Characteristics. *Appl. Sci.*, 9, 3673.

The new table reads as follows:

Supplementary Table 3. Stimulus durations, inter pulse-interval (ms) and rate (Hz) for the two electrophysiological protocols (short and long)

Short protocol				Long protocol			
Stimulation duration [ms]	Inter-pulse-interval [ms]	Rate [Hz]	Repeats per sweep	Stimulation duration [ms]	Inter-pulse-interval [ms]	Rate [Hz]	Repeats per sweep
1	399	2.5	10	1	199	5	20
10	390	2.5	10	10	190	5	20
100	900	1	10	100	400	2	20
200	1800	0.5	5	200	800	1	10
500	1500	0.5	3	500	500	1	5

Comment 16

IEGE: Additional justification for why you chose the 40Hz (15ms on, 10ms off) stimulus protocol would be helpful. After the pigeons were stimulated by the LED for 30 minutes, they were sensory deprived for 60 minutes afterwards—would you lose some of your IEGE signal by then? Why not perfuse them immediately after the LED stimulus?

Answer 16

We chose the 40 Hz (15ms on, 10 ms off) protocol, as we used this protocol in the behavioral experiments. The justification for why we have chosen this protocol for behavioral experiments can be read in comment 43.

We are sorry that the procedure of the IEGs was not adequately explained. Animals were first sensory deprived for 60 minutes to achieve low overall neuronal activity (background activation) before the stimulation started. Then, the animals were stimulated for a period of 30 minutes with alternating phases of 5 min 40 Hz stimulation and 5 min without stimulation. As c-Fos and other IEGs reach their peak expression after about 60 minutes, we did not perfuse them immediately after the stimulation. In these 60 minutes pigeons were further sensory deprived to reduce stimulation unrelated c-Fos expression. Thus, we did not lose any signal because of the delayed expression peak.

Below you can find some studies describing the expression time course of c-Fos.

<https://pubmed.ncbi.nlm.nih.gov/3037702/>

<https://pubmed.ncbi.nlm.nih.gov/11861125/>

We have clarified this in the manuscript as we have noted that this part was poorly worded. The new section in the results reads as follows:

“The physiological validity of the applied protocol was further verified in an additional experiment. Here, we investigated the functionality of ChR2 with immediate early gene expression. Following a sensory deprivation phase of one hour, pigeons were stimulated for a period of 30 minutes with alternating intervals of 5 min 40 Hz stimulation and 5 min no stimulation with orange light in one hemisphere and blue light in the other hemisphere (Fig. 6 a). After that, the pigeons were sensory deprived for a further 60 minutes to allow for adequate c-Fos expression and subsequently transcardially perfused with PFA. Sensory deprivation before and after the experiment was performed to reduce stimulation unrelated c-Fos expression. Subsequently, stainings against the immediate early gene c-Fos were performed. The cellular activation was assessed at two stimulation sites in each hemisphere of three pigeons within the stimulated entopallium and in a control area within the unstimulated visual wulst.

Comment 17 Behavior:

Would be helpful to report the bilateral implantation in the manuscript, and how long would the 40Hz stimulus last?

Answer 17

We described the bilateral implantation in the method section and though it may become clear based on the illustration showing the two light beams above the entopallium in figure 7 a. However, we have now made this clearer in the figure description of figure 7 as well as in the result section.

We described the duration of the stimulation in the method section where it can be easily overlooked.

We have now included a sentence in the result section.

To investigate the behavioral effect of optogenetic stimulation, an experimental group expressing ChR2 and a control group expressing tdTomato were bilaterally stimulated in the entopallium during the whole stimulus presentation phase or until the pigeon responded.

Comment 18

What does gray-scale different 10 vs. 50 mean? Were the luminance stimulus checked with a light meter? It would also be nice to see the resulting psychometric curve, perhaps in a supplemental figure.

Answer 18:

Yes, the stimuli were checked with a light meter and we have now included the resulting luminance differences values (cd/m^2) in the supplementary material (see supplementary fig 20). Greyscale difference 10 refers to the increments of RGB values that were between the two depicted stimuli. For example, in category difference 10 the animal would see one stimulus RGB 170 and one RGB 180, whereas in category difference 50 the animal would see one stimulus RGB 150 and one RGB 200. This can be seen in supplementary fig. 20, but we have added a further panel to this figure showing the cd/m^2 values. Psychometric curves of a similar experiment in pigeons have already been published in Lengersdorf et al. 2014.

The new supplementary figure looks as follows:

Supplementary Figure 20. Stimuli of the behavioral paradigm. (a) Stimuli of the five different stimulus classes (SC) with high RGB values (mean RGB value 215). **(b)** Stimuli of the five different SC with medium RGB values (mean RGB value 175). **(c)** Stimuli of the five different SC with low RGB values (mean RGB value 135). The RGB value difference between the two grey scale stimuli in SC1 was 10 (mean luminance difference $5.81 \text{ cd}/\text{m}^2$), in SC2 20 (mean luminance difference $12.56 \text{ cd}/\text{m}^2$), in SC3 30 (mean luminance difference $17.96 \text{ cd}/\text{m}^2$), in SC4 40 (mean luminance difference $23.91 \text{ cd}/\text{m}^2$) and in SC5 50 (mean luminance difference $30.39 \text{ cd}/\text{m}^2$). All RGB values are written in the upper right corner in all images. **(d)** Luminance differences were measured with a photometer. Mean luminance difference gradually increased from SC1 to SC5. The error bars represent the standard error of the mean.

Comment 19

Is it necessary to report the groups you used in your ANOVA in the manuscript rather than in the methods section?

Answer 19

We included the groups into the main text to facilitate the understanding of the underlying statistics. We believe this to be beneficial to the reader as we performed different statistical analyses for the different sections (transduction efficiencies, NeuN, electrophysiology, behaviour, ...) and the reader does not need to refer back to the method section for each analysis.

Comment 20

Discussion: I would be interested to hear any ideas the authors may have as to why they found 3 “classes” of responses during the electrophysiology as well as why the birds were more effected by the LED stimulus during the more difficult contrast perception tasks as opposed to being equally effected at all difficulty levels.

Answer 20:

As we stated above, we toned down the statement on the clear distinction between three classes of neurons. However, obvious differences in the response profiles might be the result of differences between the transfection qualities of different neurons, resulting in different numbers of expressed channels. Further, neurons are embedded in distributed neural networks and might be influenced by other cells in a recurrent network. Further, an indirect response can be the cause of the response. In addition the exact spot where the recordings are made (soma, dendrites, axons) cannot be disentangled and might also lead to different response characteristics.

The behavioral results might be explainable by adding noise to the system of integrating visual information. If the neuronal representation remains stable but noise is increased, the resulting signal to noise ratio is reduced. This might lead to a higher proportion of incorrect decisions following the stimulus presentation.

We have added a new section to the discussion that reads as follows:

Visual processing depends on such adaptive coding mechanisms, since neurons within the sensory system vary their responses dynamically according to the input⁶⁰. In visual discrimination tasks, where one stimulus needs to be selected over the other, cells in higher visual areas code for this discrimination by selectively increasing and decreasing their firing rates for the selected and non-selected stimuli respectively. The difference in firing rates of these cells represents the discriminative information, which is greater for simple compared to complicated discriminations⁶¹. Since the optogenetic manipulation within the entopallium was not cell type specific, the discriminative information of these cells was probably disrupted by elevating the firing rates of all cells, thereby reducing their contrasts. This reduced contrast in neural representations might have led to the impaired discrimination accuracy. Thus, for cell type unspecific optogenetics during visual discrimination, excitatory optogenetic tools might offer advantages over inhibitory optogenetic tools, as excitation creates a new signal that can disturb the population dynamics more intensely than inhibition, which might result in floor effects⁶². Although excitatory optogenetic tools offer the above mentioned advantages and might yield comparable effects to temporal lesions⁶³, it needs to be noted that these tools might have knock-on effects on connected circuit areas of the entopallium such as the NFL, NI and MVL⁶⁴. Thus, to further substantiate the role of contrast perception to the entopallium, similar experiments using inhibitory tools or pharmacology could be performed.

Comment 21 Methods:

During injections, did you always inject in a single track or did you have to do multiple tracks in the same area to fill the entire region?

Answer 21:

During the viral comparison study, we injected a single tract of 5 μ l which was actually enough for AAV1-hSyn and AAV1-CAG to fill the great majority of the entopallium. We have included supplementary figure 2 showing the spread in anterior/posterior direction for all serotypes (see comment 60).

However, we have noticed that the diffusion of the virus differs between structures. For example, in the NCL, striatum, HC and Wulst the expression was rather confined along the injection tract (roughly 200 μ m, as can be seen in Fig 3). Thus, in these areas I would recommend performing multiple injections when planning in vivo optogenetics.

Injecting a greater volume nevertheless increases the amount of transfection. Thus, for the behavioral experiments we injected the virus at two anterior/posterior locations (mirroring the placement of the dual fiber optic cannula), to make sure that Chr2 expression would be extensive.

Comment 22

How did you inject your constructs and how did you know the exact volume injected?

Answer 22

We injected the constructs with a pulled glass capillary and pressure injection system. We made sure that 5 μ l were injected as we pipetted 5 μ l onto parafilm, sucked the whole drop into the capillary, and injected the whole volume. Alternatively, a Hamilton syringe can be used which we are doing at the moment and works as well as the self-made capillary system.

We have included a sentence to the method section clarifying this point which reads as follows:

For the viral transfection study 5 μ l of each viral vector were pressure injected into at least five separate hemispheres of three pigeons. Therefore, 5 μ l were pipetted on parafilm and drawn into a glass pipette with a 25 μ m tip. The total volume was distributed over a range of 500 μ m dorsal/ventral.

Comment 23

Did you calibrate your fiber optic cannula before implanted to be sure of the optical power output? What is the high end of your power output? Reporting a range would be more helpful than just above 3.5 mW

Answer 23

Yes, we have checked our fiber optic cannula before implantation. Moreover, we have checked the output before the experiments. In the old version of the manuscript, we reported the lowest value measured in these tests. In the revised manuscript we made this point clearer.

The new section reads as follows:

Before cannulas were used for behavioral or electrophysiological experiments, the light output at the cannula tip was verified using a power meter. Before each experiment the light output of the patch chord was also checked (range of light intensity cannula: 3.5 mW – 4.5 mW, range of light intensity at patchchord: 4.5 mW – 6 mW, Thorlabs, Newton, USA).

Comment 24

Line 647: I found the language confusing —I did not realize immediately that “randomized stimulation” meant the LED light, not the discrimination stimulus.

Answer 24

We have now clarified this to avoid confusion.

Comment 25

Figure 2: d, e, h, k: would be nice to have label of retro vs. antero,

Answer 25

We have now labeled anterograde vs. retrograde transport.

Comment 26

e: does this mean no expression in MVL at 6 mo because it is no longer highlighted?,

Answer 26

There is extensive anterograde expression after 6 months. We hope this becomes clearer now as we included the new supplementary table 1 and supplementary figures 3-8 quantifying anterograde and retrograde transport as described in comment 2.

Comment 27

e, f, g: mirroring of schematic and image a bit confusing—a label of LSt in the schematic would help orient viewers

Answer 27

We arranged the pictures like this for aesthetic purposes (symmetry) and now included the label for orientation.

Comment 28

Figure 4/5 could be combined into a single figure exploring cellular tropism.

Answer 28

Thank you for this suggestion. We have now combined the figures, see comment 3/4.

Comment 29

Figure 8e: significance is from experimental group (ChR2 + 40Hz) compared to which control?

Answer 29

The significance value that was plotted in figure 8 e (now figure 7 b) depicts the significance value of the post hoc test comparing stim (ChR2 + 40 Hz) to stim (tdTomato + 40 Hz). We included this figure and analysis to show that not only a stimulation effect occurs within the ChR2 group, but that this group is also performing worse than the control group when stimulated.

Reviewer #3 (Remarks to the Author):

In my opinion this is a fine work reporting excellent news for the avian brain community. The paper is clearly written, and nicely illustrated. Results are convincing, and very promising. Methods, including statistical analysis, look sound and are adequately described. Discussion is also well written, and covers the main aspects of the work presented. I have only minor concerns I will like to point out:

Comment 30

The number of individuals forming the control and experimental groups in the behavioral experiment it is not said.

Answer 30

So far, the number of animals involved in the behavioral experiment was only stated in the table description of table 2 (n = 6 experimental group and n = 6 control group with 5 retests each), which may have easily been overlooked. We have now rearranged the old figure 8 (now figure 7) and furthermore stated the number of pigeons more clearly in the result section. Reviewer 1 had a similar concern, and a more detailed answer as well as the new figure 7 can be seen in comment 34.

Comment 31

Neither it is said the rate of success of the transfections. How many animals were transfected with the AAV1-CAG construct? How many of these attempts were successful? I think a table containing such info may be very valuable to assess the reliability of the method.

Answer 31

All injections that were made for AAV1-CAG during the viral comparison (n = 6) and during the behavior experiments (n = 12) were successful. The viral transfection can vary in its extent which is also apparent when looking at figure 1 (raw data points are plotted). However, there was no case where an AAV1-CAG injection did not result in any labeling. The only tested serotypes where there was no labeling to be detected following an injection were AAV9-hSyn (4 out of 5 injections) and AAV5-hSyn (1 out of 5 injections) for one time.

Comment 32

Legend of figure 7 describe panel 7g as "illustrating that no c-Fos expression occurred in the orange stimulated hemisphere". However, picture 7g indeed show some dark dots that looks very much alike c-Fos expression. I suggest rephrasing such description.

Answer 32

We now included a quantification of the c-Fos data based on a reviewer's suggestion. We have rephrased the figure description to include this quantification. A more detailed comment, the new result section and the revised figure (now figure 6) can be found in comment 7.

Reviewer #4 (Remarks to the Author):

Rook et al., provide the first example of optogenetic investigations in pigeons and evaluate a set of serotype and promoter combinations that are currently readily available to the scientific community - making for a valuable resource across labs. They quantify and highlight the usefulness of the AAV1-CAG construct for ChR2 expression in the pigeon brain, particularly, but not limited to, the entopallium. They go on to directly demonstrate the functionality of ChR2 expression to induce activity with light after transfection in the entopallium. Although the electrophysiological data presented is convincing, some additional controls and/or information about the recordings would be useful. Finally, they demonstrate the ability to directly manipulate behavior by disrupting activity in the entopallium with light stimulation of ChR2 in a context discrimination task. The results are novel in the respect that optogenetic manipulations have not been performed before in pigeons. Although this is not the first demonstration in birds, large differences can exist across avian species and given the import of pigeons as a model, this will be of interest to others in the field - particularly saving time/effort to optimize across various available AAV constructs. In general the paper is well written and includes high quality figures.

General comments:

Comment 33.

Previous work in finches has been done with the CMV promoter (e.g. Roberts et al., 2012). Is there a specific reason why hSyn and CAG were selected for these studies? It would seem to me that having a CMV comparison would be beneficial. In the least, the choice of promoters should be justified (e.g. lines 111) and comparison with CMV discussed (e.g. line 357).

Answer 33

When we started the project, our aim was that the whole community is able to use this method with commercially available constructs. Addgene currently offers CAG, CBA, CamkII, hSyn, EF1a, mDlx promoters. We decided for the CAG promoter because this promoter is known for strong and stable transgene expression in other species (Klein et al. 2002). As in optogenetics high transduction/expression rates are desired, the CAG promoter was our promoter of choice. To have a possible comparison and neuron selectivity the hSyn promoter (Klein et al. 1998) was chosen additionally.

<https://pubmed.ncbi.nlm.nih.gov/12093083/>

<https://pubmed.ncbi.nlm.nih.gov/9527887/>

We believe that our results cannot be compared to those of Roberts et al. (2012), as they used a custom built self-complementary AAV2/9-CMV which differs not only in the promoter but also in other aspects to our AAV2/9-CAG (for standard AAVs cell mediated second strand synthesis is necessary, whereas self-complementary AAVs are ready for immediate replication and transcription).

Furthermore, in Roberts et al. (2012), no quantification of the ChR2 expression was given. Moreover, more recent studies in zebra finches (Xiao et al. 2018) report the use of other constructs such as scAAV1-Cbh, scAAV9-Cbh and AAV1-CAG, or in Zhao et al. (2019) scAAV2/9-NX indicating a wide variety of promoter systems used.

The new section in the discussion reads as follows:

Thus, it is still possible that AAV9 combined with other promoters provides a useful optogenetic tool. Indeed, several studies conducted in zebra finches use scAAV9 in combination with the neurexin or CMV promoter in areas of the zebra finch song system^{7,44}. However, our results can not easily be compared to these studies, as they used custom built self-complementary AAVs (scAAVs), which differed not only in the integrated promoters but also in other aspects, as scAAVs do not require second strand synthesis but are immediately ready for replication and transcription.

Comment 34

Can the authors explain the choice to perform statistics across sessions and not animals For Figure 8 and related analysis. Specifically, since the injection site (and subsequent potential for a on/off stim ChR2 effect) for each animal does not change over sessions if there is no effect of learning across sessions, as stated.

Answer 34/Answer 36:

In our manuscript we calculated a repeated measures ANOVA with the within subject factor “session (session 1, session 2, session 3, session 4, session 5)”, “stimulus classes (SC1, SC2, SC3, SC4, SC5)” and “stimulation (on/off)” and the between subject factor “group (ChR2/tdTomato)” to include all available data into one analysis. The advantage of this analysis is that it offers the possibility to test for effects of the sessions (training effects) or whether there is an interaction of session and stim (to test whether the stimulation effect differed between sessions). Neither an effect of session, nor an interaction between session and stim was found.

As we tested 6 animals in 5 sessions we ended up with 30 individual data points plotted in old figure 8. However, those 30 sessions were not simply used as our N in the data analysis. When testing the effect of stim x condition x group, the repeated measures ANOVA considers that the 5 “session values” belong to specific pigeons and uses mean values of the 5 sessions for every pigeon.

In the revised manuscript, we have now moved the old figure 8 to the supplementary material (supplementary fig. 21) and updated it by connecting stim and no stim in all conditions, for all pigeons and sessions. Moreover, we plotted SC5 for all sessions and pigeons and connected stim and no stim values. In 4 out of 5 sessions a significant stimulation effect was found in SC 5. In session 4 no significant SC5 stimulation effect was found, however, in this session, a significant SC4 stimulation effect was seen. This indicates that the effect was robust across sessions.

In the main text we have now decided to plot the mean values of the 5 sessions for all 6 pigeons (as there was no session effect, and no interaction of session and stim) to clarify the number of pigeons used for the experiment (fig. 7). The new figure now includes the mean values of all pigeons plotted in different colours and connecting lines between stim and no stim. The colours are consistent across all analyses.

The new supplementary figure 21 looks as follows:

Supplementary Figure 21. Transient 40 Hz activation of ChR2 expressing cells in the avian entopallium reduces contrast sensitivity. (a) Schematic illustration of the experimental procedure. Pigeons were conditioned to discriminate grey-scalers of different stimulus classes (SC). SC1 consisted of grey-scale pictures that were difficult to discriminate and SC5 consisted of grey-scale pictures that were easy to discriminate. Pigeons were stimulated in half of the trials in a given session and stimulation took place during the whole stimulus presentation phase or until the animal responded. (b) Depicts the 40 Hz stimulation effect on the performance in SC5 over all five sessions for the ChR2 group (n = 6). Each data point represents the performance in one session of one pigeon. Stimulated and unstimulated performances within each session/pigeon have been connected with lines. Colors represent performances of individual animals (session 1 SC5 ($t_{(5)} = 4.126$, $p = .009$), session 2 SC5 ($t_{(5)} = 2.832$, $p = .037$), session 3 SC5 ($t_{(5)} = 2.935$, $p = .032$), session 4 SC5 ($t_{(5)} = 1.029$, $p = .351$), session 5 SC5 ($t_{(5)} = 5.616$, $p = .002$). However, even though session 4 SC5 was not significant, in this session a significant effect was seen in SC4 ($t_{(5)} = 2.661$, $p = .045$). (c) Depicts the 40 Hz stimulation effect on the performance of the ChR2 group (n = 6, the same data is shown in main text Fig. 7 c). To visualize performances of individual sessions/pigeons, the raw data of all sessions/pigeons is plotted. Furthermore, stimulated and unstimulated performances within each session/pigeon have been connected with lines. (d) Depicts the 40 Hz stimulation effect on the performance of the tdTomato group (n = 6, the same data is shown in main text Fig. 7 d). To visualize performances of individual sessions/pigeons, the raw data of all sessions/pigeons is plotted. Furthermore, stimulated and unstimulated performances within each session/pigeon have been connected with lines. Error bars represent the standard error of the mean (SEM). *** $p < .001$, ** $p < .01$, * $p < .05$.

The new main text figure 7 looks as follows:

Figure 7. Transient 40 Hz activation of ChR2 expressing cells in the avian entopallium reduces contrast sensitivity. (a) Schematic illustration of the experimental procedure. Pigeons were conditioned to discriminate grey-scalés of different stimulus classes (SC). SC1 consisted of grey-scale pictures that were difficult to discriminate and SC5 consisted of grey-scale pictures that were easy to discriminate. Pigeons were bilaterally stimulated in half of the trials in a given session and stimulation took place during the whole stimulus presentation phase or until the animal responded. (b) Visual discrimination performance of the control (n = 6) and experimental group (n=6) displayed in one graph. The control and experimental group had comparable performances and a significant drop in performance could only be seen for optogenetic stimulation in the experimental group in SC3 and SC5. (c) Visual discrimination performance of the experimental group expressing ChR2 in the entopallium.

Optogenetic stimulation reduced contrast sensitivity as indicated by a significant reduction of discrimination accuracy for stimuli in SC3, SC4 and SC5, but unimpaired discrimination performance in SC2 and SC1 **(d)** Visual discrimination performance of the control group expressing tdTomato in the entopallium. Optogenetic stimulation had no effect on contrast sensitivity as discrimination performance was equal between stimulated and unstimulated trials in all stimulus classes. **(e),(f)** Reaction times for stimulated and unstimulated trials of the experimental and control group. There was no difference in reaction times between the different stimuli classes or between stimulated and unstimulated trials for both groups. **(c-f)** Mean performances of all 5 sessions are plotted for all pigeons in individual colors. Stimulated and unstimulated performances within each pigeon have been connected with lines. Error bars represent the standard error of the mean (SEM). *** $p < .001$, ** $p < .01$, * $p < .05$.

Comment 35.

The choice to evaluate the functional success of the optogenetic stimulation with a task that the entopallium has not previously been determined to play a role in is somewhat confusing. Perhaps demonstrating both a novel role for the nucleus and evaluating the effects of optogenetic stimulation in the same nucleus is somewhat difficult to interpret. Can the authors comment on how they can assign the role of context discrimination directly to the entopallium when there are surely knock-on effects of the stimulation in connected circuit regions. There are some references to lesion studies in the discussion, but these should be elaborated if they want to strengthen the claim that they have established a new role for entopallium in context discrimination. Alternatively, I am not certain this is fundamental to the novelty of the paper and could also be toned down until further confirmatory experiments evaluating more circuit elements/blocking activity are performed.

Answer 35

It is correct that the specific task that we used to determine contrast perception has so far not been tested in pigeons (simultaneous contrast). However, the idea that the entopallium is involved in intensity discrimination/perception has already been proposed and investigated in two very similar behavioral lesion studies.

<https://pubmed.ncbi.nlm.nih.gov/5459212/>

<https://pubmed.ncbi.nlm.nih.gov/3166707/>

Our study offers methodological advancements and to our opinion provides stronger evidence for the role of the entopallium in contrast perception since in the old lesion studies, the lesions were not confined to the entopallium. Nevertheless, optogenetics can have knock-on effects in connected areas, therefore we discuss these opportunities more thoroughly in the revised manuscript and toned down the novelty of our finding but support our idea by including more information on prior lesion studies.

The new part in the discussion reads as follows:

We found that 20 Hz as well as 40 Hz blue light optical stimulation of ChR2 expressing neurons within the entopallium resulted in impaired grey-scale visual discrimination, while the same manipulations in control birds expressing tdTomato did not result in any behavioural effects. This indicates that the blue light stimulation itself had no effect on visual discrimination accuracy and that the discrimination deficit in the experimental group can be traced back to the changed physiology of cells within the entopallium. Furthermore, these findings suggest that the entopallium is involved in contrast perception/visual discrimination of luminance, which is well in line with lesion studies that have been performed in the entopallium and other areas of the avian collothalamoc pathway^{17,19,20,57}. For example, lesions of the entopallium have been shown to reduce the ability to categorize stimuli into bright and dim¹⁹ or to discriminate between pictures of varying patterns and luminance²⁰. However, in most cases, the lesions were not confined to the entopallium, complicating the attribution of the observed perceptual and behavioural deficits to the functionality of the entopallium alone. Using optogenetics, a stronger claim for the role of the entopallium in contrast perception can now be made, as we selectively increased the firing rates of entopallial cells. This procedure has been shown to be a suitable approach for disrupting behaviors that rely on heterogeneous as well as time-varying population coding^{7,58,59}. Visual processing depends on such adaptive coding mechanisms, since neurons within the sensory system vary their responses dynamically according to the input⁶⁰. In visual discrimination tasks, where one stimulus needs to be selected over the other, cells in higher visual areas code for this discrimination by selectively increasing and decreasing their firing rates for the selected and non-selected stimuli respectively. The difference in firing rates of these cells represents the discriminative information, which is greater for simple compared to complicated discriminations⁶¹. Since the optogenetic manipulation within the entopallium was not cell type specific, the discriminative information of these cells was probably disrupted by elevating the firing rates of all cells, thereby reducing their contrasts. This reduced contrast in neural representations might have led to the impaired discrimination accuracy. Thus, for cell type unspecific optogenetics during visual discrimination, excitatory optogenetic tools might offer advantages over inhibitory optogenetic tools, as excitation creates a new signal that can disturb the population dynamics more intensely than inhibition, which might result in floor effects⁶². Although excitatory optogenetic tools offer the above mentioned advantages and might yield comparable effects to temporal lesions⁶³, it needs to be noted that these tools might have knock-on effects on connected circuit areas of the entopallium such as the NFL, NI and MVL⁶⁴. Thus, to further substantiate the role of contrast perception to the entopallium, similar experiments using inhibitory tools or pharmacology could be performed.

Comment 36.

Nevertheless, the results that context discrimination is affected in some way by the optogenetic manipulation are convincing, although it is vital they present the data to be compared stim vs no stim directly following for each session/animal to evaluate the precise effect size of the stimulation itself, not just averaged across groups (as noted in detail below).

Answer 36

We have provided more information on this in comment 34. As recommended, we included statistics for SC5 in all sessions in the supplementary material (supplementary fig. 21) and illustrated this by plotting connecting lines between stim and no stim values for both each animal and session.

Comment 37.

In general, timelines of various experiments/expression could be clarified throughout. E.g.: Line 112: 6 weeks seems like a long time for AAV expression? Yet, experiments after 6 months would suggest expression continues to increase. What are the practical implications of this? How does this compare to AAV expression in other species?

Answer 37

Timelines for the expression experiments should be clearer now, as they were 6 weeks for the viral comparison study depicted in figure 1. We have included a supplementary table 1 quantifying the amount of anterograde and retrograde transgene expression which also includes the expression time for every single case. Indeed, expression times of 6 weeks are rather long compared to rodents. However, studies in zebra finches using AAV1-CAG report similar expression times (6-12 weeks, Xiao et al., 2018). The main difference from 6 weeks to longer expression times is the amount of anterograde and retrograde expression. The practical implications of 6 weeks expression times are not too bad as pigeons need post-surgery recovery times as well as time to get accustomed to the optogenetic set up and patch chords. Thus, the time of expression is not “lost” but can be used for these purposes. For anterograde and retrograde experiments, we would advise longer expression times similar to Xiao et al. (2018). However, expressions here might also depend on the specific projection (distance between injection and target area) and should therefore be adapted for the specific experiments.

<https://pubmed.ncbi.nlm.nih.gov/29551492/>

Comment 38

Line 498: the authors chose 6 weeks because this is deemed to be ‘stable’, but data from 6 months (anterograde and retrograde) would suggest otherwise(?). Could the authors please clarify/reconcile this. Additionally, what was the precise timeline of expression in pigeons used for behavioural tests. The general outline is indicated in the methods, but precise details are missing for the specific data analyzed.

Answer 38

The first part of the questions has been answered in the previous comment 37. For the behavioral experiment testing started 6 weeks after surgery. In the mean-time pigeons recovered from the surgery and were retrained/got accustomed to the patch chords.

We have clarified this in the manuscript and the new section reads as follows:

After 14 days and full recovery, pigeons were trained in the final paradigm again to get accustomed to the patch chords. Behavioral testing was started when 6 weeks since the surgery had elapsed, and behavioral performance was back to criteria. The 6 weeks between surgery and behavioral testing allowed for stable transgene expression.

Specific comments:

Comment 39.

Line 100: please include references for early lesion studies of note.

Answer 39

We have rephrased the section in question (see introduction) and have included more references for early lesion studies. For more information see comment 76.

Comment 40.

Line 110: ‘..different AAV serotypes...’

Answer 40:

This section has been rephrased in the updated manuscript (see comment 14).

Comment 41.

Line 202: a very brief indication here of the electrophysiological recording method would be useful. (i.e. extracellular single-unit recordings).

Answer 41:

We included the methods details in the main text. It reads:

The physiology of ChR2 was assessed in two experiments. In the first experiment, pigeons were anesthetized, and extracellular single unit recordings and optical stimulation were performed simultaneously within the entopallium of two pigeons in 4 hemispheres (Supplementary Fig. 19).

Further information can be found in comment 5.

Comment 42.

Line 203-204: It should be made clear that the goal of these specific experiments is to pre-select cells that are responsive to the light and assess their characteristics – not to do an unbiased assessment of what proportion of cells in the entopallium are then responsive to light after injection, for instance. How were cells chosen for recording is only indicated in the methods (this should be briefly mentioned in the results for clarity) as it includes pre-screening for optically-evoked responses. Therefore, to say that ‘In all recorded cells, a significant number of action potentials could be evoked by optical stimulation’ is a bit misleading since these were very specifically selected for it would seem. Also this should be addressed in some way, as Figure 4 shows many NeuN+ and ChR2- cells, so implications of low-transfection rates should be discussed in light of behavioural manipulations – as is a caveat in all optogenetic experiments. Do the authors have any data from cells that were not responsive to light, or any indication of how hard it is to find these responsive cells? Also, I did not see a place where it mentioned how many pigeons these cells were from. This should be included.

Answer 42:

We thank the reviewer for raising these issues and apologize for the unclear description of the methods. As also requested by reviewer 1, we included the coordinates of the respective recording sites of the two animals (four hemispheres) used in our experiments. We also included a brief section about the purpose of these experiments, the preselection of neurons and the search protocol to the main text (see comment 5).

The only information we have on how hard it is to find these cells is of anecdotal nature and based on our own electrophysiological experiments. The transfected areas were easy to identify due to evoked multiunit activity. We experienced no major problems in approaching single units. The only problem – but this is always the case using single electrode extracellular recordings – was to separate single units resulting in a sufficient signal-to-noise ratio.

The transfection rate will be discussed in more detail in comment 66.

Comment 43.

Line 220: I am not sure I understand the leap from Figure 6 results and the preceding two paragraphs to: ‘therefore we decided to use a pulsed optical stimulation protocol of 40 Hz in our behavioral experiments (pulse duration: 15 ms; inter-pulse interval: 10 ms).’ Just because 10ms was effective I assume? And what about the inter-pulse interval? Please clarify this.

Answer 43

Again we would like to apologize for the unclear description of the stimulation pattern used in our experiments. The 10 ms pulse duration resulted in a significant number of evoked spikes for all neurons tested. Further, when expressed in spikes per second, the number of evoked action potentials were found to be highest in the 10 ms duration for most of the recorded units (see supplementary figure 18 for raw and normalized data). For cultured hippocampal neurons it was reported, that 10 ms is the minimal time necessary to achieve a full neuronal response upon stimulation. The range of 10-30 ms was found to be optimal. Thus, we choose a stimulation time of 15 ms. For the inter-pulse interval we decided try two different conditions. We choose the 40 Hz stimulation protocol (15 ms stimulation and 10 ms inter-pulse interval) and in addition a 20 Hz stimulation protocol (15 ms stimulation and 35 ms inter-pulse interval) in our behavioral experiments. The inter-pulse interval was specified based on ChR2 off-kinetics, which are about 6 ms for 20 ms pulse duration and 15 ms for 1 s pulse duration.

DOI: [10.1016/j.neures.2005.10.009](https://doi.org/10.1016/j.neures.2005.10.009)

Comment 44.

Line 252: Can you give some specifics about ‘repeatedly tested’ and the sessions that were analyzed. How many sessions over how many days for learning for this experiment, range and variability, length of sessions, etc. I see from the methods that this was a very long training procedure, it would be useful to know the variability across learning for the pigeons and if this correlated with any measure in the experiments.

Answer 44:

Pigeons were tested until 5 sessions that matched the criteria were gathered (see more on this in comment 45). The learning for this experiment was rather variable and procedures were individually adapted to the animal’s needs, and can thus not objectively be compared. However, in the end, when the test sessions started, all pigeons performed equally well (we controlled for this by establishing criteria). We do not believe that learning history correlated with any measure in the test sessions, as during the test sessions animals showed highly comparable performances as well as similar effects of stimulation (see more on this in comment 46).

Comment 45.

Line 255: How were the 5 session chosen for analysis? The first 5 sessions to meet criteria? The last 5 sessions? Were there more than 5 to chose from for each pigeon? If so how was this done? In relation to this, are the results comparable if all sessions reaching criteria are used (i.e. how reliable are the results across all sessions)? If the pigeon is performing at 50% in SC5 in a session, does the

light have any additional effect? One may hypothesize that it would not if it was specifically behaviourally related, as in SC1.

Answer 45

Animals were tested in one condition (20 Hz or 40 Hz) until 5 sessions that reached the criteria (300 completed trials and 75% performance in at least one SC) were gathered. Thus, we have only 5 sessions in total from every pigeon and condition that reached the criteria. We have clarified this in the revised manuscript. It never occurred that a pigeon performed at chance level in SC5 during the test session phase, thus we cannot provide any data on the light effect in this case. A chance level performance in this category would not be a perceptual issue, but rather a cognitive issue (the pigeons have forgotten the rule of pecking the darker image). Thus, we can only speculate that no effect would have occurred as the entopallium is a rather perceptual brain area.

In general, only very few sessions did not meet our criteria. In these sessions, mainly the 50% participation criterion was not met. Calculating an analysis with these sessions would be prone to errors. First, one might ask whether the pigeon was not motivated enough to participate and to engage in the test. Such a reduced task motivation would especially affect sessions with fewer trials, since every individual stimulus class is presented less often. When considering that we have tested 10 different conditions in one session (SC1 - SC5, each stim and no stim), a sufficient task engagement is necessary for meaningful results. Therefore, we decided to set up a criterion of at least 300 completed trials, to end up with on average at least 30 trials for every tested condition.

Comment 46:

Related to above: While Figures 8c-f are informative, it will be important to show the results by animal, directly comparing each stimulus class Chr2 stim and no stim conditions – to see the within animal (& within session) effect of the optogenetics individually, rather than just the mean effect of the group, i.e. are the best performers in SC5 no stim also the best performers in SC5 stim. This is not a given and should be demonstrated. Is the effect size (absolute decrease in performance) consistent or variable across sessions/animals. Equally, does this effect size correlate to any property of the injection (e.g. size of injection site or density of Chr2 labelling in entopallium, etc).

Answer 46

In the revised manuscript we have updated the old figure 8 (new figure 7) and plotted connecting lines between the individual data points so that the absolute effect of the stimulation can be more easily seen. Furthermore, we calculated the effect of stimulation for every individual session (see comment 34). Both the figure and the statistics for this can now be found in the supplementary material in supplementary figure 21. As you can see, the effect of stimulation occurs in 4 out of 5 sessions in SC5.

Furthermore, there was no systematic difference between the animals as there was no animal where the effect occurred in less than 4 of the 5 sessions. Thus, the variations we see are daily fluctuations in animal's performances and this is the reason why we have tested them in five session instead of only one. Therefore, we believe that a correlation would not offer a meaningful outcome.

Comment 47.

How many times were stimuli classes presented in a session and were they randomized or did they proceed from SC1 to SC5 or vice versa?

Answer 47:

In a given test session all five stimulus classes appeared equally often in a randomized order. If a session was 600 trials, each stimulus class would occur 120 times (60 times stimulated and 60 times unstimulated).

Comment 48.

It is interesting that there is no effect of stimulus class on reaction time even though they are performing at near chance levels for SC1. The 'deciding' phase is not prolonged even if the task is difficult. Is this consistent with previous literature?

Answer 48

It has been shown that in sensory tasks there is a speed accuracy trade off. Accuracy sometimes decreases when decisions are performed quickly, especially when stimuli are difficult to identify. In these rather complicated cases, longer reaction times are often associated with better performances (Kay et al., 2006). However, this does not apply to all situations.

For example, one study from Rinberg et al. (2006) found that decision making times in mice in an olfactory two alternative discrimination task remained stable and were independent of task difficulty, when they were allowed to choose their own sampling rates. This strategy resulted in high performances/rewards in easy discriminations, but lower performances/rewards for complicated discriminations (in some cases even chance level). These mice were however able to perform better when they were forced to increase their sampling times. This indicates, that even though they could theoretically perform better, they preferred speed over accuracy, as it resulted in a good enough reward level with minimal effort.

We suspect that our pigeons used a similar strategy, as there was very little cost of errors (light turned off) and their response patterns resulted in enough rewards over the entire session (performed good enough in SC 4 and SC 5).

<https://pubmed.ncbi.nlm.nih.gov/16880120/>

<https://pubmed.ncbi.nlm.nih.gov/16880129/>

Comment 49

There is no comparison between the reaction time of the ChR2 injected group and the control group. The control group seems to have faster reaction times in general. Is this significant? If so, could the authors explain. This may be an important control to indicate that the ChR2 is not toxic or interfering with other elements of processing in the task.

Answer 49

For the revised manuscript, we have rearranged figure 8 and supplementary figure 4. Now both groups are plotted in one graph (supplementary fig. 22) so that they can be more easily compared. Furthermore, in the main text, the reaction times for the experimental and control group are now plotted in fig 7 e, f. We rearranged the statistics and included the between subject factor "group" to test for reaction time differences. It is correct that the control group displays slightly faster reaction times in general compared to the experimental group. However, this difference does not reach significance and is mainly driven by one animal, which is particularly slow (it is important to note that this animal is not an outlier and also shows the best performance (red dot)).

The new statistics read as follows:

Furthermore, the groups did not differ significantly in their overall reaction times (no main effect of group $F_{(1,10)} = 2.067$, $p = .181$, $\eta_p^2 = .171$, no interaction of “stimulation”, “stimulus class” and “group” $F_{(4,40)} = 2.139$, $p = .094$, $\eta_p^2 = .176$, Supplementary Fig. 22).

The supplementary figure looks as follows:

Supplementary Figure 22. Transient 40 Hz activation of ChR2 expressing cells in the avian entopallium does not influence reaction times. (a) Schematic illustration of the experimental procedure. **(b)** There is no difference between reaction times in stimulated and unstimulated trials. The control group ($n = 6$) was a bit faster than the experimental group ($n = 6$). The reaction time difference between the experimental and control groups did not reach significance ($F_{(1,10)} = 2.067$, $p = .181$, $\eta_p^2 = .171$) **(c)** Depicts the 40 Hz stimulation effect on the reaction times of the ChR2 group (the same data is shown in main text Fig. 7 e). To visualize performances of individual sessions/pigeons, the raw data of all sessions/pigeons is plotted. Furthermore, stimulated and unstimulated performances within each session/pigeon have been connected with lines. **(d)** Depicts the 40 Hz stimulation effect on the reaction times of the tdTomato group (the same data is shown in main text Fig. 7 f). To visualize performances of individual sessions/pigeons, the raw data of all sessions/pigeons is plotted. Furthermore, stimulated and unstimulated performances within each session/pigeon have been connected with lines. Error bars represent the standard error of the mean (SEM).

Comment 50:

Line 307: I think it would be prudent to note that the 20Hz protocol (please also specify stim time and inter-stim interval here) had comparable results only in the SC5 category – although the trend is clear, this stimulation protocol did not seem to be sufficient to elicit significant differences in SC3 and SC4.

Answer 50

Indeed, there is no significant difference of stimulation in SC3 and SC4. However, it needs to be noted that the 20 Hz protocol was only tested in 5 animals as in one animal the thread for attaching the cannula to the cable got lost. Thus, the effect might have been smaller due to the reduced power. However, as we cannot be sure whether 20 Hz is less efficient or whether the result occurred due to the reduced N we toned down the interpretation and only mention that 20 Hz stimulation also had an effect. Furthermore, we updated the figures in the supplementary material and now also show connecting lines between all individual data points for the 20 Hz stimulation protocol.

The new section in the results reads as follows:

A different pulse protocol with 20 Hz stimulation was tested in 5 pigeons and was also able to impair behavioral performance for stimuli in SC5 (see Supplementary Fig. 23)

The new supplementary figure looks as follows:

Supplementary Figure 23. Transient 20 Hz activation of ChR2 expressing cells in the avian entopallium reduces contrast sensitivity in SC 5. (a) Depicts the 20 Hz stimulation effect on the performance for the ChR2 group (n = 5). Each data point represents the performance in one session of one pigeon. Stimulated and unstimulated performances within each session/pigeon have been connected with lines. There was no main effect of session ($F_{(4,16)} = .469$, $p = .757$, $\eta_p^2 = .105$), 20 Hz stimulation ($F_{(1,4)} = 6.666$, $p = .061$, $\eta_p^2 = .625$) and stimulus classes ($F_{(4,16)} = 72.059$, $p < .001$, $\eta_p^2 = .947$) on the accuracy of grey-scale visual discrimination. However, there was an interaction between “20 Hz stimulation” and “stimulus classes” ($F_{(4,16)} = 6.810$, $p = .002$, $\eta_p^2 = .630$). Bonferroni corrected pairwise comparisons revealed that optogenetic stimulation significantly impaired the performance for stimuli of SC5. **(b)** Depicts the 20 Hz stimulation effect on the reaction times for the ChR2 group (n = 5). Each data point represents the performance in one session of one pigeon. Stimulated and unstimulated performances within each session/animal have been connected with lines. There was no main effect of session ($F_{(4,16)} = 1.167$, $p = .362$, $\eta_p^2 = .226$), 20 Hz stimulation ($F_{(1,4)} = .063$, $p = .814$, $\eta_p^2 = .016$) and stimulus classes ($F_{(4,16)} = 2.533$, $p = .081$, $\eta_p^2 = .388$) on the reaction times. Furthermore, there was no interaction between “20 Hz stimulation” and “stimulus classes” ($F_{(4,16)} = .717$, $p = .593$, $\eta_p^2 = .152$). **(c)** Depicts the same data as (a) but the session means are plotted for every pigeon in individual colors. **(d)** Depicts the same data as in (b), but the session means are plotted for every pigeon in individual colors. Error bars represent the standard error of the mean (SEM). ** $p < .01$. Dots represent raw data of all 5 pigeons tested in 5 sessions.

Comment 51.

Line 479: typo in second virus, presumable AAV5.hsyn...

Answer 51

We have corrected the typo in the revised manuscript.

Comment 52.

Line 482: how were injections made? Hamilton syringe (gauge)? Glass pipette (tip diameter)?

Answer 52

Injections were made with glass pipettes that had a tip diameter of 25 μm . We have described this in the method section.

Comment 53

Line 484: change ‘pigeons’ here to ‘experiments’; related,

Answer 53

We have changed this in the manuscript.

Comment 54

line 487: ‘The pigeons used in behavioural experiments...’

Answer 54

We have changed this in the manuscript.

Comment 55

Line 494: states recovery period was 14 days but link to timeline in behaviour is not made until further and virus expression was much longer than 14 days. It may be useful to specify recovery until what here.

Answer 55

The recovery period was 14 days. After that, pigeons were trained again and accustomed to the patch chords. The actual testing was then started 6 weeks after surgery (or later depending on the animals performance).

Comment 56

Line 512: The hSyn promoter virus was eYFP though correct? I assume part of the point of the DAB immune was also to create equal comparisons between these two AAVs (i.e. also different fluorophores)– if so, I would state this explicitly. Similarly though, was this eYFP signal sufficiently bright on it's own? Also, Figure 3 (mCherry) looks quite bright - was this sometimes variable then?

Answer 56

We realized that there was a huge discrepancy between the YFP and mCherry signal. While the native mCherry expression was quite bright, the native YFP expression was barely visible. Thus, to enable an equal comparison between all constructs, we conducted the antibody stainings and subsequent DAB reaction. We have made this point clearer in the method section in the revised manuscript.

The new section reads:

The counterstaining was performed to allow for an equal comparison between the serotypes, as serotypes with the hSyn promoter were tagged with eYFP, whereas serotypes with the CAG promoter were tagged with mCherry. Moreover, the amount of transgene expression can be underestimated when analyzing native fluorescence, as the signal increases with counterstainings (see Supplementary Fig. 1 and method section for more detail).

Comment 57

Line 531/533: In my experience, 'cells wells' are generally referred to as XX well plates (e.g. 24 well plates).

Answer 57

We have now changed this to 12 well plates in the revised manuscript.

Comment 58

Lines 552/554, etc. Do the authors mean AlexaFluor 594?

Answer 58

Yes, sorry, this was a typo. We have changed it in the revised manuscript.

Comment 59

Line 573: it would be useful to know the units of 'value'. Are these in um, pixels, arbitrary, etc.?

Answer 59

The values are arbitrary units except for area which refers to pixels. We have included more information on microscopic analysis, as for the revised manuscript we performed further similar analyses. We highlighted the specific section in the microscopic analysis section.

Comment 60.

Line 577: states that analysis for area was performed in 1/10th of the brain sections and then summed, but was the total number of sections kept constant? Was there a specified start and end point or was this determined by the injections size alone? (or atlas sections?) Taking every 10th section could theoretically give you a variable number of total sections to sum. Also, every 10th section seems quite low – this would be analysis only every 400um. Were there any differences in the rostral-caudal extent of labelling from injections that might indicate the injection size itself differed (which would clearly affect the amount of resulting labelling)?

Answer 60

Indeed, the whole analysis was performed in 1/10th of the total brain sections. Thus, the total amount of brain sections that were stained and regarded for analysis was kept constant. However, depending on the anterior posterior spread of the virus, the number of sections that contained signal could vary (thus the total amount of sections to sum was sometimes not the same). Nevertheless, this variance was not large and is now depicted in supplementary figure 2a in the revised manuscript. Although only every 10th section was taken for the analysis, labeling occurred in 4.5 – 7.5 slices in all serotypes indicating a rostro-caudal extent of 1800 μm even for the least efficient serotype. Thus, we believe that analyzing every 10th section still sufficiently depicts the full picture as the anterior posterior spread is large.

However, in the revised manuscript, we have included two new analyses based on your suggestions. In the new main text Fig 1 b we have now depicted the percentage of transduction in comparison to the whole size of the entopallium. This analysis takes into account that the serotypes have slightly different anterior posterior spreads and depicts the efficiency over larger distances.

In the second analysis, which can be found in supplementary fig. 2 b, we have depicted the ChR2 expressing area in relation to the size of the entopallium in slices with transduction only. This analysis compares the viral vectors independent of their anterior posterior spread.

The new section in the results reads as follows:

The efficiency of all six constructs was assessed based on the number of ChR2 expressing somata (Fig. 1 a) and the transfected area of ChR2 expressing somata, dendrites and axons in relation to the size of the whole entopallium (Fig. 1 b).

.... Furthermore, the serotype had a significant effect on the percentage of ChR2 expressing area within the entopallium (one-way ANOVA with Welch correction, $F_{(4,8.515)} = 12.791$, $p = .001$, Fig. 1 d). There was no significant difference in the percentage of transduced area between injections of AAV1-hSyn-ChR2 and AAV1-CAG-ChR2 (Bonferroni corrected pairwise comparisons, $p = .402$, Fig. 1 d, e, h). However, AAV1-CAG-ChR2 resulted in a significantly greater area expressing ChR2 than all other serotypes including AAV5-hSyn-ChR2 ($p < .001$, Fig. 1 d, f), AAV5-CAG-ChR2 ($p = .001$, Fig. 1 d, i) and AAV9-CAG-ChR2 ($p = .001$, Fig. 1 d, j, for mean values and SEM see table 1). Moreover, the ChR2 expressing area was significantly greater for AAV1-hSyn compared to AAV5-hSyn ($p = .039$, Fig. 1 d, e, f). Furthermore, transduction efficiencies of the serotypes followed a similar pattern, when the ChR2

expressing area was compared to the size of the entopallium only in slices with transduction (see supplementary Fig 2).

The new method section reads as follows:

For the comparative transduction analysis, the total amount of Chr2 expressing cells was determined for every injection in one brain series (one tenth of all brain sections) and then multiplied by 10 to estimate the actual amount of Chr2 expressing cells. For the analysis of transduced area, the total Chr2 expressing area within the entopallium was determined in the same brain series. Furthermore, the size of the entopallium was measured in all sections of the series. The transduced area was then put into relation with the size of the whole entopallium of that series and with the size of the entopallium in sections with Chr2 expression of that series. All serotypes were included in this analysis except for AAV9-hSyn-ChR2, as this serotype did not result in reliable Chr2 expression.

New main text figure 1.

Figure 1. Comparative transduction analysis of AAV1, AAV5 and AAV9 in combination with the hSyn and CAG promoters. (a) Schematic illustration of the injection area and analysis type. For the first analysis, all somata were counted that displayed Chr2 expression. (b) Schematic illustration of the injection area and the analysis type. For the second analysis, the area of Chr2 expressing somata, dendrites and axons was measured and compared to the total area of the entopallium. (c) Quantitative comparison of all tested constructs in their ability to drive transgene expression in somata of the entopallium. AAV1-hSyn-ChR2 as well as AAV1-CAG-ChR2 were significantly more efficient than all other tested constructs. (d) Percentage of transduced entopallia area for all tested constructs. AAV1-CAG was significantly more efficient than all other tested constructs. (e-j) Qualitative pictures of Chr2 expression following injections of (e) AAV1-hSyn-ChR2 (n = 5), (f) AAV5-hSyn-ChR2 (n = 5), (g) AAV9-hSyn-ChR2 (n = 5), (h) AAV1-CAG-ChR2 (n = 6), (i) AAV5-CAG-ChR2 (n = 5), and (j) AAV9-CAG-ChR2 (n = 5). All scale bars represent 500 μ m. Error bars represent the standard error of the mean (SEM), ***p < .001, **p < .01, *p < .05. Abbreviations: **AAV**: Adeno-associated viral vector, **hSyn**: human synapsin 1 gene promoter, **CAG**: chicken beta actin promoter.

Supplementary Figure 2. Transduction efficiencies of all tested AAVs. (a) Spread of all serotypes in the anterior posterior plane within the entopallium. (b) Percentage of Chr2 expressing area compared to the size of the entopallium in slices with transduction. The AAVs with the hSyn promoter are depicted in grey and with the CAG promoter in blue. Error bars depict the standard error of the mean (SEM).

Comment 61

Line 610: typo 'end'

Answer 61

We have corrected the typo in the revised manuscript.

Comments related to figures:

Figure 1:

Comment 62

1. D) To get a better idea of the density it would be helpful to know the total area of the entopallium for the measured sections (or simply to report density rather than summed area, which may be variable depending on the rostral-caudal extent of the injection itself).

Answer 62:

For the revised manuscript we have improved this analysis based on your suggestions. We now depict the transfected area in relation to the size of the whole entopallium and in relation to the size of the entopallium in sections with signal (see comment 60).

Comment 63

C-d) there seems to be a large variability in number of labelled soma (5-25000 range) – what are the implications for behaviour here? Can the authors speculate on the source of this variability (how is injection volume controlled, etc).

Answer 63

Indeed, there was a large variation in the amount of expression as can be seen in figure 1. This might result from various factors such as qualities of perfusions, minimal variations in titer, or individual difference between the animals. First of all, we do not believe that these issues skewed the viral comparison as they should occur for all serotypes. Moreover, we do not believe that this had an effect on the behavioral study. First of all, the behavioral effect was very comparable between all tested pigeons. Secondly, the area that can be reliably activated by light is limited to 200 μm to 500 μm . For this reason, we believe as long as there is sufficient transgene expression around the optic fiber, behavioral effects should be similar.

Figure 2:

Comment 64.

Are e to f & g oriented in the same way? F and g appear to be flipped mediolaterally to e? It is hard to tell if this is also the case for b and c.

Answer 64

Yes, we have flipped e and f for symmetry reasons. Thus, we would prefer to leave the figure like it is. However, we stated this more explicitly in the figure description and included the abbreviation for the highlighted area.

Comment 65.

G) There appears to be no labelling in LSt, but some in GP for hSyn?

Answer 65

Indeed, there is some labeling in GP, but this is not specific for hSyn and was also sometimes seen for CAG. We would not feel confident in describing this labeling as retrograde transport due to the proximity of GP and the entopallium and the possibility of injection leakage.

Figure 4:

Comment 66.

It would be helpful to add to b) a graph of NeuN+/ChR2- cells (% of NeuN cells labelled in entopallium – i.e. transfection rate)

Answer 66

We decided to analyze the transfected area in comparison to the whole entopallium in slices with DAB staining to provide information about the relative transduction efficiency. We believe that this information will be more accurate as we see an increase in signal when comparing fluorescent and DAB staining. Using fluorescent stainings, we might have underestimated the transfection rate.

Figure 6:

Comment 67

With only 9 cells, could the authors provide (at least as supplement) a figure showing responses in all these cells. This would be useful information to compare the evoked response patters across this population. This would also help to determine if dividing the 9 cells into three 'classes' is really appropriate? Are these three classes clear in a sample of only 9 cells, or are responses heterogeneous and on a continuum? (see also comment for Supp Table 1 below)

Answer 67

We provide figures of the neuronal responses for all cells and all conditions tested as supplementary material. We also agree with the reviewer that the statement of three distinct classes is not warranted with a population of only nine cells. Thus, we toned down our statement on the neuronal responses. We describe the found variability without qualifying these responses as distinct classes. More information on this can be found in comment 5.

Comment 68.

How robust are these responses across stim periods? Some measure of variability should be presented.

Answer 68

In our new figure 5 the stimulus protocol is depicted and explained in detail. We tested the neuronal responses using three consecutive sweeps. To assess the variability of the evoked neuronal responses, we compared the spikes evoked during these sweeps using a non-parametric analysis of variance (Kruskal-Wallis-Test). If we found significant differences between the sweeps a Bonferroni-corrected multiple comparison test was conducted.

In only two out of the nine cells, significant differences between sweeps in some conditions were detected. The differences were found in stimulation trials of longer duration (>100 ms). In the stimulation duration below 100 ms no significant differences could be detected. The significance is indicated in supplementary table 2 and in the single cell raster plots. Overall, the evoked responses are robust. More information can be found in comment 5 (supplementary table 2).

Comment 69

Reporting the spontaneous firing rate (in the off phase) would be helpful as it seems very low from what can be seen and this would be an important check of the overall health of the recorded/transfected neurons. Is it comparable to previous literature (i.e. ChR2 negative cells in entopallium)? Also, a stronger link could be made here in relation to the choice to use excitatory rather than inhibitory opsins.

Answer 69

We added the Baseline values for all cells in the supplementary table 2. Indeed the reported baseline values are low, but all the values are still in a range of previous reports (mean: 4.63 Hz, range: 0.2 – 27.3 Hz, Azizi et al. 2019; 108 cells recorded, 72 were spontaneously firing, range: 0.1 – 19 Hz, 36 cells showed no spontaneous activity, Gu et al. 2002). Further, the experiments were conducted under full anesthesia. To maintain a sufficient depth of anesthesia, we use a constant flow of isoflurane. In mammals, it has been shown that isoflurane has a severe depressive impact on neuronal responses (e.g. visual evoked potentials (visual system, rat), Jehle et al. 2009; extracellular recordings (auditory system, cat), Cheung et al. 2001). The depressive effects of isoflurane might have also lead to lower spontaneous activities in our recordings.

Based on reported neuronal data and the moderate spontaneous activity in the Entopallium, we decided to use excitatory opsins in our experimental approach.

Furthermore, we have included a section in the discussion describing the advantage of excitatory vs. inhibitory opsin for visual tasks and unspecific stimulation. See more on this in comment 35

Azizi, A. H., Pusch, R., Koenen, C., Klatt, S., Bröcker, F., Thiele, S., et al. (2019), Emerging category representation in the visual forebrain hierarchy of pigeons (*Columba livia*), Behavioural Brain Research, 356, 423–434.

Cheung SW, Nagarajan SS, Bedenbaugh PH, Schreiner CE, Wang X, Wong A. (2001), Auditory cortical neuron response differences under isoflurane versus pentobarbital anesthesia, Hear Res., 156, 115-127.

Jehle T, Ehlken D, Wingert K, Feuerstein TJ, Bach M, Lagrèze WA. (2009), Influence of narcotics on luminance and frequency modulated visual evoked potentials in rats. Doc Ophthalmol., 118, 217-224.

Gu, Y., Wang, Y., Zhang, T. et al. (2002) Stimulus size selectivity and receptive field organization of ectostriatal neurons in the pigeon. J Comp Physiol A, 188, 173–178.

Comment 70.

Additionally, is there any data from these cells for just visually-evoked responses to light (presented to the eye)? To demonstrate normal physiological responses?

Answer 70

This issue was also raised by reviewer 1 and our response is provided in comment 9.

Comment 71

Similarly, are there controls for these recordings showing that the cells do not respond to 620nm stimulation light?

Answer 71

Unfortunately, we cannot provide these control experiments.

Figure 7:

Comment 72

There seem to be many cells that are C-Fos positive but Chr2 negative – yet virtually no C-Fos expression in (g) (presumably reflecting a very low spontaneous activity rate?). In fact, from the presented image, the double labelling seems to be in the minority. Can the authors comment on this and potentially quantify and present the proportion of double labelling here.

Answer 72

It is correct that this could be related to a low spontaneous firing rate. Low basal expression levels have been reported for the entopallium in general, which makes this area especially suited to analyze stimulation dependent activity without having a lot of background noise (see studies attached). Especially because the animals were sensory deprived before, during and after the stimulation procedure the expression level was very low on the non-activated side.

<https://pubmed.ncbi.nlm.nih.gov/1906769/> (c-Fos chicken)

<https://pubmed.ncbi.nlm.nih.gov/16890299/> (low signal)

The exact expression level has now been quantified in our DAB stainings (see more in comment 7). Unfortunately, we have no more slices left to perform further fluorescent stainings to estimate the double labeling. The double labeling might be in the minority as for these sections no anti-ChR2 double staining has been performed. The native signal that we observed was much weaker than the DAB signal (this is shown in supplementary fig. 1). Thus, it is conceivable that some of the cells that showed c-Fos expression also expressed Chr2 but suffered from too little native fluorescent signal. Moreover, another possibility is that the c-Fos positive/Chr2 negative cells were activated by the stimulation of Chr2 positive cells.

However, the fluorescent picture was rather meant to show the specificity of the c-Fos staining for the cell nuclei, which is nicely visible from the overlay shown. Interpreting quantitative measures from this example picture would be misleading, as we only stained few sections for an example picture to illustrate this specificity.

Figure 8:

Comment 73

Although the mCherry only AAV control is convincing, it would be ideal to have a 620nm stimulation control as well for a Chr2 injected animal, even in small numbers as proof of principle. If the authors do not have this data, I would make a stronger but directly explicit argument in the text based on the mCherry controls that the stimulation light itself was not affecting the visual perception of the pigeons during the task. IF available, a video of the task during stimulation may indicate how bright the stimulation light itself is during the task etc.

Answer 73

Unfortunately, our behavioral animals were not tested with 620 nm stimulation before perfusion (this kind of experiment is not possible anymore). However, the main purpose of our control was to show that the light itself has no effect on the visual perception of the animals. This is the case, since the tdTomato blue light controls perform equally well with light on. We believe, that stimulating the Chr2 animals with 620 nm would have added confounding variables, which would have needed another control itself. This would be a new experiment of testing red vs. blue distraction during visual discrimination.

However, we made this point clearer in the manuscript based on your suggestions, highlighting that blue light has no distracting effects. We added also a picture, showing how much light can be seen from the outside, when a pigeon is stimulated.

Supp Table 1:

Comment 74.

It would be useful to see this table expanded with mean firing rates (baseline and stim +/- sem) for all cells, in addition to p values.

Answer 74

We added the requested values in the table (which is now supplementary table 2).

Comment 75.

What about the 20ms and 500ms durations? This data is missing (related, also line 602: why would these two stimulation categories not be analysed?)

Answer 75

We added the analysis of the 200 ms and 500 ms stimulation duration for all cells (see more in comment 5, fig. 5 and supplementary figures 10 – 18 and supplementary table 2)..

Reviewer #5 (Remarks to the Author):

This manuscript offers a proof of principle for the use of optogenetics in pigeons by examining the efficacy of several potential viral vectors. By examining the effectiveness of transfection at the injection site, anterograde and retrograde transport, they conclude that the AAV1-CAG is the construct of choice, whereas others are less effective, no effective, or even neurotoxic. Their conclusions are further supported with electrophysiological recordings from transfected neurons in the entopallium (E). Although their sample size is very (small), they show that all neurons can be effectively activated with pulses of light. Finally, in a very well controlled experiment they show that optogenetic activation of E can impair visual intensity discrimination in a behavioural task. Generally speaking this manuscript is well motivated, thorough, clearly written and well organized. With a few exceptions noted below, the data is subject to rigorous quantification and statistical analysis. This proof of principle is an important contribution as the pigeon is historically an important model for studying learning, memory and visual processing. These fields will benefit from the application of optogenetics. In addition, the demonstration that E is involved in intensity discrimination adds to our knowledge about visual processing in E. However, I believe the manuscript would benefit from with consideration of an E lesion study that was not cited by the authors.

Comment 76

The study not cited is by Nguyen et al. (2004), Journal of Neuroscience 24:4962-4970. In the discussion of the submitted manuscript the authors state that the previous studies of the entopallium is involved in pattern and motion processing and that the optogenetic stimulation results are in line with previous lesion studies of entopallium (lines 395-401). To the best of my knowledge, Nguyen et al. is the only study to show that lesion studies of the entopallium impair motion detection. Moreover, Nguyen et al. showed a clear double dissociation whereby the lesions to the caudal and rostral entopallium resulted, respectively, in motion and spatial vision deficits. (This finding was highly consistent with the known topographic projections from rotundas). This study could clearly inform the results of the present study, as it is likely that the optogenetic stimulation of the rostral entopallium is what impaired intensity discrimination. Moreover, their conclusion that their results lend support to the idea that the avian tectofugal pathway is similar to the mammalian geniculostriate pathway (lines 401-403 and 103-104) must be taken with caution. The pitfalls of such a conclusion is discussed in detail by Nguyen et al., who note that studies of the mammalian tectofugal pathway show that it is also involved in colour and form vision, but these studies tend to be overlooked.

Answer 76

We want to thank the reviewer for highlighting this important paper which we did not include in our original manuscript. As we have already described in comment 11, we mainly targeted the anterior entopallium, thus our data is well in line with Nguyen et al. and shows that the anterior entopallium is involved in contrast discrimination. In the revised manuscript we have stated more clearly that we investigated the role of the anterior entopallium and integrated the proposed paper into the discussion. Furthermore, we do not compare the avian tectofugal with the mammalian geniculostriate pathway anymore, based on your suggestion. Therefore, we rephrased parts of the introduction and parts of the discussion.

The new section in the introduction reads as follows:

Therefore, this study compared the efficiency of six viral constructs in their ability to transduce neurons in the pigeon forebrain, to determine the optimal viral construct for optogenetic experiments. As it has been complicated to induce behavioural effects with optogenetics in primates due to insufficient

protein expression^{25,27}, we furthermore wanted to confirm that stimulation of channelrhodopsin (ChR2) leads to physiological as well as behavioural effects in pigeons. In our study we have focused on the visual system, as birds are highly visual animals and recent studies have indicated that characteristic properties of sensory systems, such as a columnar and laminar organization, are conserved between birds and mammals¹³. We targeted the entopallium, which is the most important primary visual area in the pigeon telencephalon and which has been associated with discrimination of form, pattern, color, motion and luminance^{17,18,21}. We employed a grey-scale visual discrimination task and hypothesized that optogenetic stimulation within this structure would result in impaired contrast perception indicated by decreased discrimination accuracy. With this study, we aimed to provide the first proof of principle for optogenetics in pigeons as well as further insights into the function of the entopallium.

The new section in the discussion reads as follows:

Moreover, future optogenetic studies could investigate the effect of optogenetic stimulation on other entopallial functions such as motion, pattern, and colour discrimination. In these tasks, a functional segregation has been described as lesions within the anterior entopallium affect pattern, colour, and form discrimination, whereas lesions of the posterior entopallium impair motion processing^{18,21}. Based on these findings and comparable topographic projection patterns within the avian and mammalian tectofugal pathways, it has been proposed that the motion sensitive posterior entopallium is comparable to area MT, whereas the colour/form/pattern sensitive anterior entopallium is comparable to V2, V3 and IT in mammals²¹. As brightness sensitive neurons of the nucleus rotundus primarily project to the anterior entopallium, brightness processing has also been linked to this subdivision²¹. Although not showing a functional segregation, our study confirms that the anterior entopallium is involved in brightness perception as our neuronal manipulation was primarily focussed on this region. The finding that this area governs brightness perception furthermore suggests a functional similarity of the entopallium to mammalian V1⁶⁵.

Comment 77

In general there is a concern about how labelled neurons were counted and it was not clear in the methods and associated text. At first I thought it was an exhaustive count of all labelled cells, but at times it said that every tenth section was counted. If there is a lot of labelling, the authors should be following the principals of unbiased stereology, and clearly document these in the methods. Could a section be added to the methods clearly indicating the procedures for counting cells?

Answer 77

Analyzing every 10th section resulted in roughly 7 sections with signal per injection. In all sections with signal, the ChR2 labeling was counted exhaustively for the whole area of the entopallium, rather than estimating transfection in one subregion (counting square). To our knowledge the method of unbiased stereology is used to determine whether a cell should be counted or not, when it is located on a border of a counting square. This does not apply to our study. We have now clarified that we have counted within the whole extent of the entopallium.

Comment 78

Was the labelling distributed evenly throughout the rostro-caudal extent of the entopallium in each case? Was the extent of spread outside the entopallium quantified?

Answer 78

In the revised manuscript we have now included a figure depicting the anterior/posterior spread for every serotype. Even for the rather inefficient serotypes the spread was large. You can find more specific comment on this in comment 60/62 and the new supplementary figure 2. Furthermore, our analysis was limited to the entopallium, as signal outside the entopallium was not cellular but rather contained fibers indicating anterograde expression. We have quantified this signal, but in a separate analysis which was described more thoroughly in comment 2 and supplementary table 1 and supplementary figure 3-8. We have now also included an analysis which depicts the density of transfection in relation to the size of the whole entopallium (takes spread into account, Fig 1 b) and in relation to the size of the entopallium on sections with signal (this analysis is independent of the spread, Supplementary Fig 2b). A more detailed description of this can be found in comment 60/62 and figure 1b.

Comment 79

lines 179-180 – It is noted that there was a severe reduction of NeuN expression with AAV9 within the expression area. How was this quantified? To what areas were the comparison made? Is there any data in this regard?

Answer 79

For the revised manuscript we have quantified the NeuN levels in all sections displaying ChR2 signal, for four cases of AAV9 and five cases of AAV1. This quantification was performed automatically with ZEN 2.1 software. We have added a new method section on this procedure. The new figures and the new method section can be seen in comments 3/4/13.

Comment 80

lines 210-215 – It is stated that there were 3 classes of cells. Please indicate how many of the 9 total cells were in each class.

Answer 80

We provide figures of the neuronal responses for all cells and all conditions tested as supplementary material. In line with reviewer 1 and 4 we agree that the statement of three distinct classes is not warranted with a population of only nine cells. Thus, we toned down our statement on the neuronal responses. We describe the found variability without qualifying these responses as distinct classes (see more on this in comment 5, fig. 5, supplementary figure 10-18 and supplementary table 2).

The new section in the results reads as follows:

The recorded cells differed in their overall response properties. We found cells that responded throughout the entire optical stimulation period with a constant amount of spikes, albeit showing a pronounced peak of activation at the onset of light stimulation (Fig. 5, Supplementary Fig. 15, 17; Supplementary Fig. 18 cells 4, 7 and 9). Further we found cells that weakened their responses over the course of prolonged stimulation (Supplementary Fig. 10, 11 and 12; Supplementary Fig. 18 cells 1, 2 and 3). Another response pattern found, showed a sharp peak only during the onset of the stimulus (Supplementary Fig. 13, 14 and 16, Supplementary Fig. 18 cells 5, 6 and 8). After the electrophysiological experiments were finished, histology was performed to check for ChR2 expression in the entopallium (Supplementary Fig. 19)

Comment 81

lines 227-231 – It is indicated that optogenetic stimulation resulted in c-fos activation only in the blue-light hemisphere. Surely there must have been some basal level in the unstimulated hemisphere (and see fig 7g?). The number of c-fos-positive cells should be quantified in each E, and a alternate non-stimulated site in then telencephalon. I am also a little concerned with the section shown in 7h. The pale colour where the c-fos-positive cells are is concerning. Is this tissue damage? Is this simply c-fos activation due to injury?

Answer 81

We have now added a quantification on the c- Fos data and a more detailed comment on this can be found in answer 7. The rather weak c-Fos expression on the non-activated side can be explained with very low basal c-Fos expression levels of the entopallium. A more detailed comment on this can be found in comment 16. We have also updated the figure (now figure 6) and provided a new picture in better quality as we have stained a new series of brain sections. As you can see, the c-Fos expression also occurs in sides without injury, thus we do not believe that injury was the reason for the expression. Based on your suggestions we have also analyzed a control area in the telencephalon, the visual wulst (Fig. 6).

Comment 82

line 318 – replace “birds” with “pigeons”

Answer 82:

We have changed this in the revised manuscript.

Comment 83

line 465-466 – dosage of ketamine and xylazine should be given in mg/kg.

Answer 83

We have added this to the revised manuscript.

Comment 84

line 482 - how were the constructs injected? Hamilton syringes? Over what time course?

Answer 84

We injected the constructs with glass pipettes. They were pressure injected, with no specific time course. Typically, we inject small volumes over the course of 2 minutes. This has been described in more detail in the revised manuscript.

Comment 85

line 610 – replace “end” with “and”

Answer 85

We have corrected this in the revised manuscript.

Comment 86

line 667 – replace “with” with “by”

Answer 86:

We have replaced this in the revised manuscript.

REVIEWERS' COMMENTS:

Reviewer #1 (Remarks to the Author):

The authors did an outstanding job addressing our concerns, as well as those of the other reviewers. This is an impressive study that will hopefully be a major resource for those wishing to bring molecular tools to avian neuroscience. I do not have any further comments on the manuscript. Congratulations on an excellent piece of work.

Reviewer #2 (Remarks to the Author):

The revised manuscript addressed all of our major concerns. With the additional details and supplemental materials, the manuscript is an exciting addition to the field of avian neuroscience.

I had one question driven primarily by curiosity--it does not need to be addressed at this time. In figure 7, panel b, I'm wondering if you have any ideas as to why there are so many C-Fos+ cells in the Wulst. In an anesthetized prep with the eyes closed, I would not expect there to be as many C-Fos+ cells in the Wulst as in the blue light activated Entopallium.

Reviewer #4 (Remarks to the Author):

Rook et al., have significantly revised their manuscript and addressed all of my concerns and comments. They have made many clarifications in both the methodology as well as results and have also strengthened the discussion of the role of the entopallium in the manuscript. I have no further comments that need to be addressed and would highly recommend the manuscript for publication.

I would only like to provide a few further comments to some of the main revisions that were done. Specifically, I think the decision to rework the electrophysiological classifications of cells was very effective. The authors now describe the patterns without overstating the specific 'class'. Although this potential for discrete classification could be very interesting for the authors in a future, more dedicated, study.

Additionally, I greatly appreciate the effort of the authors to include primary data sources for the electrophysiology in the supplementary Figures (Supp. Fig 10-17) for each cell as well as the associated table (Supp Table 2). This is very well done and I think will be of great interest to others as they also establish these techniques across labs. It is sometimes a lot of effort but highly commendable to see this source data published and greatly strengthens the conclusions and transparency of the manuscript.

Similarly, the revised Figure 7 is improved and better demonstrates the robustness of the optogenetic stimulation on behaviour. The addition of the lines connecting the matching responses for each animal in Figure 7 and Supp Figures 21-23, also greatly strengthen the conclusions in my opinion. The effects are very clear and convincing in Figure 7c when one transitions from SC1 to SC5 and very reassuring to see on the level of each individual bird.

Reviewer #5 (Remarks to the Author):

The authors have taken on the mammoth task of answering 5 reviewers. They have done a truly outstanding job. I have no additional concerns.

General Reply:

We would like to express our gratitude to the editor and the five anonymous reviewers for their positive evaluation of our manuscript and their positive feedback to our initial responses. Attached we answered the remaining question by reviewer 2.

Reviewer #2 (Remarks to the Author):

The revised manuscript addressed all of our major concerns. With the additional details and supplemental materials, the manuscript is an exciting addition to the field of avian neuroscience. I had one question driven primarily by curiosity--it does not need to be addressed at this time. In figure 7, panel b, I'm wondering if you have any ideas as to why there are so many C-Fos+ cells in the Wulst. In an anesthetized prep with the eyes closed, I would not expect there to be as many C-Fos+ cells in the Wulst as in the blue light activated Entopallium.

Answer:

Here a misunderstanding may have occurred. The pigeons in the electrophysiological experiment were anesthetized. However, the pigeons that were part of the c-Fos experiment were not anesthetized. The c-Fos pigeons were awake and sensory deprived in a Skinner Box where they were stimulated with orange/blue light. Thus, the higher level of c-Fos expression in the Wulst compared to the entopallium is in line with relatively low baseline expression levels that are known for the entopallium in awake birds. Moreover, as the animals were not head fixated and it was not completely dark in the box, the c-Fos expression in the Wulst may be related to movements and minimal changes in visual perception. In the revised manuscript we now specifically highlight that the c-Fos animals were not anesthetized during stimulation.